# Cryo-EM structures of human Cx36/GJD2 neuronal gap junction channel

Seu-Na Lee [1,6], Hwa-Jin Cho[2,6], Hyeongseop Jeong [3], Bumhan Ryu[4], Hyuk-Joon Lee [1], Minsoo Kim[5], Jejoong Yoo[5], Jae-Sung Woo [1,7] ✉ & Hyung Ho Lee [2,7] ✉

Connexin 36 (Cx36) is responsible for signal transmission in electrical synapses by forming interneuronal gap junctions. Despite the critical role of Cx36 in normal brain function, the molecular architecture of the Cx36 gap junction channel (GJC) is unknown. Here, we determine cryo-electron microscopy structures of Cx36 GJC at 2.2–3.6 Å resolutions, revealing a dynamic equilibrium between its closed and open states. In the closed state, channel pores are obstructed by lipids, while N-terminal helices (NTHs) are excluded from the pore. In the open state with pore-lining NTHs, the pore is more acidic than those in Cx26 and Cx46/50 GJCs, explaining its strong cation selectivity. The conformational change during channel opening also includes the α-to-π-helix transition of the first transmembrane helix, which weakens the protomer-protomer interaction. Our structural analyses provide high resolution information on the conformational flexibility of Cx36 GJC and suggest a potential role of lipids in the channel gating.

Intercellular signaling is an essential function of multicellular organisms and involves directly connecting two adjacent cells for cell-to-cell communication. The direct connection between adjacent cells is made by end-to-end docking of two hemichannels (termed connexons) from each cell, each consisting of six protomers, thereby forming a dodecameric gap junction channel (GJC)[1]. The GJCs are a family of integral membrane proteins, enabling direct exchange of electrical and small molecule signals such as ions, second messengers, hormones, and metabolites[2]. As a result, GJCs play key roles in numerous cellular processes, including synaptic electrical transmission, cardiac contraction, development, and differentiation. Vertebrate GJCs are formed by connexins, while invertebrate GJCs consist of innexins with no sequence identity to connexins. Vertebrate Pannexins have detectable sequence identity and structural similarity with innexins, but function as hemichannels connecting cytoplasmic and extracellular space[3].

Twenty-one human connexin genes, except for the highly diversified connexin 23 (Cx23)/GJE1, share high sequence identity (50–80%) in the region spanning four transmembrane (TM) helices and two extracellular loops (ECLs). However, cytoplasmic regions, including the N-terminal helix (NTH), cytoplasmic loop (CL), and C-terminal tail (CT), are quite diverse, suggesting that each GJC has its own specific functions mediated by these regions. Notably, it has been suggested that conformational changes in NTHs are critical for channel gating[4,5]. The gating and permeability of GJCs are regulated by voltage, pH, divalent ions, and membrane lipids[6,7]. Furthermore, GJCs can be dynamically regulated by isoform composition, assembly, disassembly, or post-translational modifications such as phosphorylation[8,9]. Therefore, it is crucial to understand the structural features of each GJC, to determine how they are differentially regulated in various cell types.

Cx36 is mainly expressed in neurons and forms the major component of GJCs in electrical synapses, playing important roles in

[1]Department of Life Sciences, Korea University, Seoul 02841, Republic of Korea. [2]Department of Chemistry, College of Natural Sciences, Seoul National University, Seoul 08826, Korea. [3]Center for Research Equipment, Korea Basic Science Institute, Chungcheongbuk-do 28119, Korea. [4]Research Solution Center, Institute for Basic Science, Daejeon 34126, Republic of Korea. [5]Department of Physics, Sungkyunkwan University, Suwon 16419, South Korea. [6]These authors contributed equally: Seu-Na Lee, Hwa-Jin Cho. [7]These authors jointly supervised this work: Jae-Sung Woo, Hyung Ho Lee. ✉e-mail: jaesungwoo@korea.ac.kr; hyungholee@snu.ac.kr

cognitive functions, including memory consolidation and epileptogenesis[10]. It is also expressed in pancreatic β-cells, mediating insulin secretion[9,11]. The abolition of Cx36 gene in mice largely disrupted the synchrony of agonist-induced supra- or subthreshold oscillations[12,13], suggesting its pivotal role in brain function. The dysfunction of Cx36 is closely related to the acquired central nervous system (CNS) diseases, amyotrophic lateral sclerosis (ALS), and diabetes[14–17]. Therefore, understanding the molecular structure of Cx36 GJC is of biological and medical interest. Notably, Cx36 contributes to neuronal death following a range of acute brain insults such as ischemia, traumatic brain injury, and epilepsy, suggesting that specific blockers of Cx36 GJC might be useful for treating these pathological situations[17]. For example, in ALS, secondary neuronal death is extended by neuronal GJCs, and progressive neuronal death can be mitigated by blocking these channels[17]. However, since Cx36 is widely expressed in the nervous system and plays a role in the regulation of neuronal activity, inhibiting Cx36 GJCs could disrupt normal brain function and potentially worsen the symptoms of neurodegenerative disorders.

Although various structural and physiological studies of connexin GJCs have been performed, it remains unclear how the large pores of GJC are completely closed, and whether lipids are directly involved in the closing process. Analysis of the undocked hemichannel structure of *Caenorhabditis elegans* innexin-6 in lipid nanodiscs showed that flat double-layer densities obstruct the channel pore, suggesting that lipids can completely close the hemichannel[18]. A recently solved structure of the human pannexin 1 channel, which shares high structural homology with innexin hemichannels, showed pore-occlusion by phospholipids in the presence of a chemical inhibitor, probenecid[19]. However, there is no experimental evidence for a lipid-mediated closing model of connexin GJCs.

In this study, we determine eight structures of human Cx36 GJC in its pore-occluded and open states using single-particle cryo-electron microscopy (cryo-EM). In the pore-occluded state, the channel pores are filled with two layers of lipids, and the NTHs of Cx36 are dissociated from the pore. In comparison, the channel pore is completely open without any obstruction in the pore-lining NTH (PLN) state, suggesting that the binding of NTHs to TM1 and TM2 of the channel pore through the hydrophobic interaction is an essential step for channel opening. Extensive single-particle analyses and molecular dynamics (MD) simulations are used to investigate the structural dynamics and functional properties of Cx36 as a neuronal gap junction channel.

## Results

### Structure determination of Cx36 GJC at 2.2 Å resolution using the BRIL-fusion method

To understand the function of Cx36 GJC in electrical synapses, we conducted a structural study of Cx36 using cryo-EM. Wild-type Cx36 (Cx36-WT) proteins were solubilized in lauryl maltose neopentyl glycol (LMNG) and cholesterol hemisuccinate (CHS) and purified as dodecameric GJCs (Supplementary Fig. 1b). In the cryo-EM images, Cx36-WT GJC particles showed a highly preferred orientation (top view) (Supplementary Fig. 1c). This problem was difficult to solve by extensive screening of grid types, glow-discharge protocols, grid preparation methods, and sample buffer conditions. We attributed this to 12 long cytoplasmic loops (CLs) at both the top and bottom of Cx36 GJC. Approximately 41% of the total residues in these CLs are hydrophobic (Supplementary Fig. 1e). We therefore reasoned that the CLs might prefer the hydrophobic air-water interface, facilitating the preferred orientation of the particles[20].

Based on this hypothesis, we designed three Cx36 constructs by removing the CLs (residues 109–187) or their replacement with cytochrome b562RIL (BRIL; residues 21–128, Supplementary Fig. 1a) or T4 lysozyme (residues 21–128). Since BRIL and T4 lysozyme are highly soluble and their N- and C-termini are close to each other (~8 Å

between the two ends), they could replace the CLs without disturbing the structural integrity of the TM helices, but significantly decrease the hydrophobicity of the cytosolic regions of Cx36 GJC. The three constructs were individually produced in insect cells, but only the BRIL-fused Cx36 (Cx36-BRIL) could be purified with sufficient yield for cryo-EM single-particle analysis. BRIL fusion indeed changed the behavior of Cx36 GJC particles in thin vitrified ice, and we obtained cryo-EM images with various particle orientations (Supplementary Fig. 1c). Using Cx36-BRIL, we determined the high-resolution cryo-EM structure of Cx36 GJC solubilized in LMNG (hereafter referred to as Cx36$_{LMNG}$-BRIL) with D6 symmetry at 2.2 Å (Fig. 1a and Supplementary Tables 1 and 2). The cryo-EM structure revealed a dodecameric architecture of Cx36 subunits with overall dimensions of 90 Å × 90 Å × 140 Å (Fig. 1a). The overall structure and dodecameric interactions in Cx36 GJC were similar to those of Cx26 homomeric and Cx46/50 heteromeric GJCs, as expected from the high sequence identity (52–54%) between these connexins[21,22]. The two hemichannels docked with each other through intermolecular interaction of ECLs, and water molecules were highly concentrated at the boundary between the TM helices and the ECLs (Fig. 1a, middle, red spheres). At the corresponding boundary of Cx31.3 hemichannel, a solvent tunnel was observed[23]. However, similar to available Cx26 and Cx46/50 GJC structures, the solvent tunnel was closed in Cx36 GJC by the interactions of Glu49, Arg77, Arg246, and Glu249 residues (Supplementary Fig. 2a).

### Unique features of the hemichannel-hemichannel docking interface

Each protomer of Cx36 contains two ECLs that are responsible for hemichannel-hemichannel docking. Both ECLs (ECL1 and ECL2) are interconnected by three disulfide bonds (Cys55-Cys242, Cys62-Cys236, and Cys66-Cys231, Supplementary Fig. 2b). While the residues of ECL1 that participate in docking interactions are conserved in the connexin family, those of ECL2 includes two variable residues. These variable residues are together called the compatibility motif because two connexons with the same motif can dock together not only between identical but also different connexons[24,25] (Fig. 1c). As depicted in Fig. 1c, the ECL2 of Cx26 docks through the interactions mediated by Lys168, Asn176, Thr177, and Asp179; however, Cx36 contains Lys238 and Glu239 in ECL2, instead of the K/R-N motif seen in Cx26 and Cx46/50 GJCs[21,22] and the H-type of recently reported Cx43 GJC[26]. This motif is therefore Cx36-specific and leads to the formation of alternative intermolecular salt bridges in it. We reasoned that the salt bridges between Lys238 and Glu239 contributed to the tight docking of the Cx36 hemichannels. Indeed, the K238E mutation caused the dissociation of the Cx36 gap junction into the two hemichannels (Supplementary Fig. 1d).

### Cx36$_{LMNG}$-BRIL has flexible NTHs and pore-bound detergents

We observed no Coulomb potential map for NTHs in Cx36$_{LMNG}$-BRIL, which was also the case with the X-ray structure of Cx26 GJC in n-Decyl-β-D-Maltopyranoside (DM) detergents[21] (Fig. 1a). Although another structure of Cx26 GJC in n-Undecyl-β-D-Maltopyranoside (UDM) detergents contains pore-lining NTHs, their weak electron densities indicate that they are mostly disordered or not strongly bound to transmembrane domains (TMDs)[21]. It is plausible that, when the amphipathic NTHs do not line the pore vestibule, a large hydrophobic surface is exposed to the solvent and masked by detergents present in the sample solution. Indeed, we observed substantial densities presumed to be CHS molecules (two CHS molecules per protomer) and acyl chains of LMNG or cellular lipids, which sufficiently covered the cylindrical hydrophobic surface of the pores (Fig. 1b). This suggests that detergents or lipids may contribute to the flexible NTH (hereafter referred to as FN) state of Cx36.

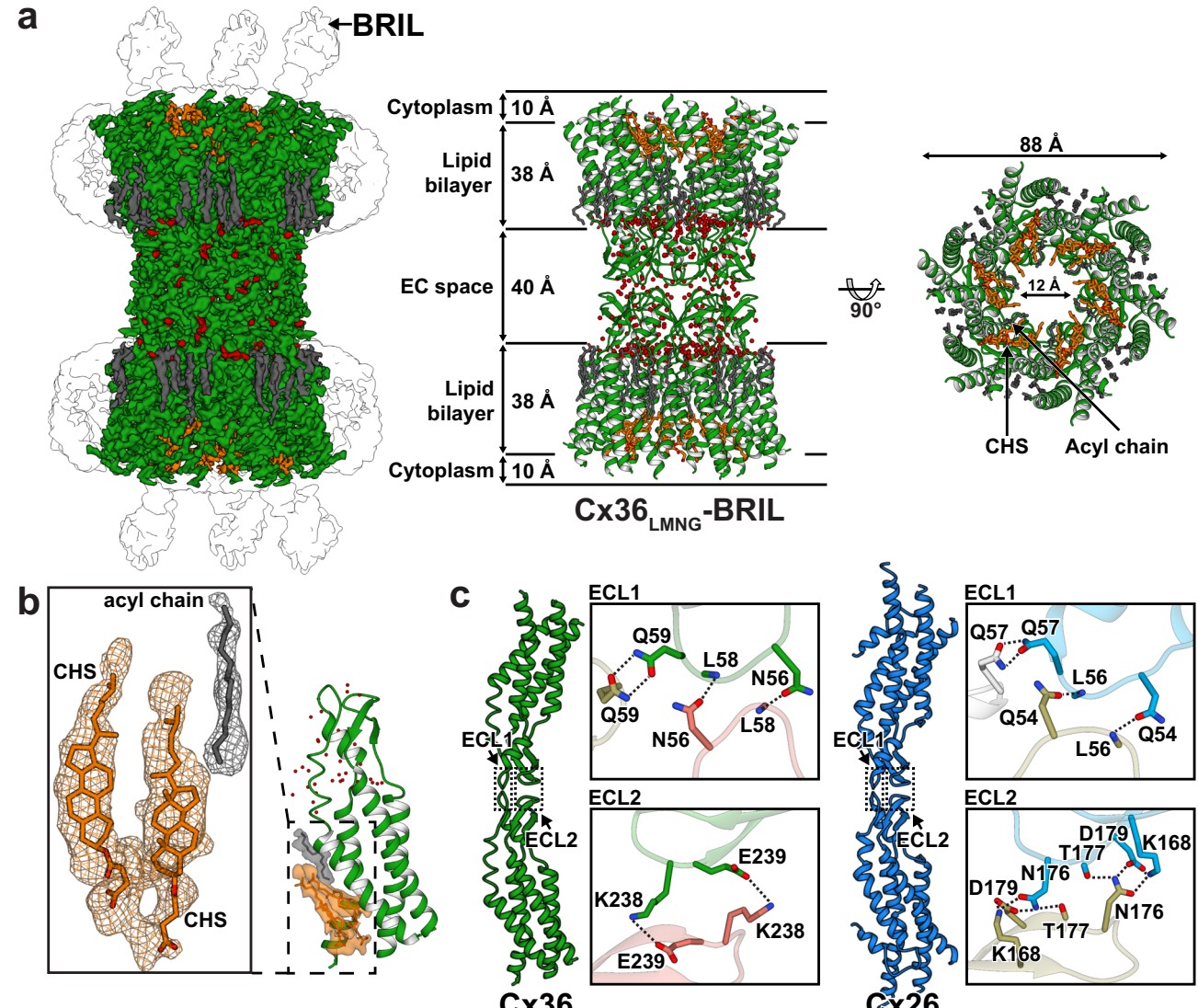

**Fig. 1 | Overall structure and inter-hemichannel docking interface of human Cx36. a** Cryo-EM reconstruction map and ribbon representation of Cx36$_{LMNG}$-BRIL. The density and atomic model of Cx36, CHS, acyl chain, and water molecules are colored green, orange, gray, and red, respectively. The ambiguous densities of detergent micelles and BRIL are colored white. The size of each part of the protein (middle), the channel diameter, and the solvent-accessible pore diameter (right) are represented. **b** Two CHS densities and one acyl chain density covering the surface-exposed NTH-binding site at the channel entrance. **c** Comparison of the junctional docking regions of Cx36 and Cx26 (PDB code 2ZW3). EC extracellular space, CHS cholesteryl hemisuccinate, ECL extracellular loop, NTH N-terminal helix.

## Structures of Cx36 GJC in soybean lipids in two different conformations

To investigate the structure of Cx36 GJC in a lipid bilayer, we reconstituted wild-type Cx36 GJCs in lipid nanodiscs (hereafter referred to as Cx36$_{Nano}$-WT) using the membrane scaffold protein 1 E1 (MSP1E1) and soybean polar lipid extract. The preferred orientation problem was partly solved by adding 50 mM phenylalanine to the GJC-nanodisc sample, and we obtained a cryo-EM consensus map of Cx36$_{Nano}$-WT with D6 symmetry (Supplementary Fig. 3). In this map, we observed clear map densities of the NTHs lining the channel pores. However, the map densities of the NTH-TM1 linkers were not observed, and those of the cytoplasmic halves of the TM helices were very poor.

Because locally poor densities are usually caused by varying conformations of the corresponding region, we performed further 3D classification using the initial consensus map and solved two GJC structures refined with D6 symmetry in PLN and FN conformations at 3.05 and 3.16 Å resolutions, respectively (Supplementary Fig. 3). In the structure of the PLN state, the map densities of NTHs and TM helices

were much clearer than those in the initial consensus map, and those of NTH-TM1 loops were also clearly observed, allowing us to build a reliable atomic model for all regions except those of CL and CT (Fig. 2b). In the FN state, the protein model including TM helices and ECLs was almost identical to those in the LMNG/CHS environment (Figs. 1a and 2a). However, unlike the structure in detergents, it showed no clear map densities of CHS molecules in the interior of the pore (Supplementary Fig. 6a), probably because the resolution was much lower and/or CHSs were mostly removed during nanodisc reconstitution.

## Unique structural features and hydrophobic pockets of Cx36 GJC in the PLN state

Compared with other human connexins, Cx36 has a longer NTH-TM1 loop (residues Ala13-Ser19) owing to the insertion of Ala14 (Supplementary Fig. 4). Although the amino acid sequence of the NTH-TM1 loop is not highly conserved in the connexin family, its length is strictly conserved with the exception of Cx36. In the available Cx43

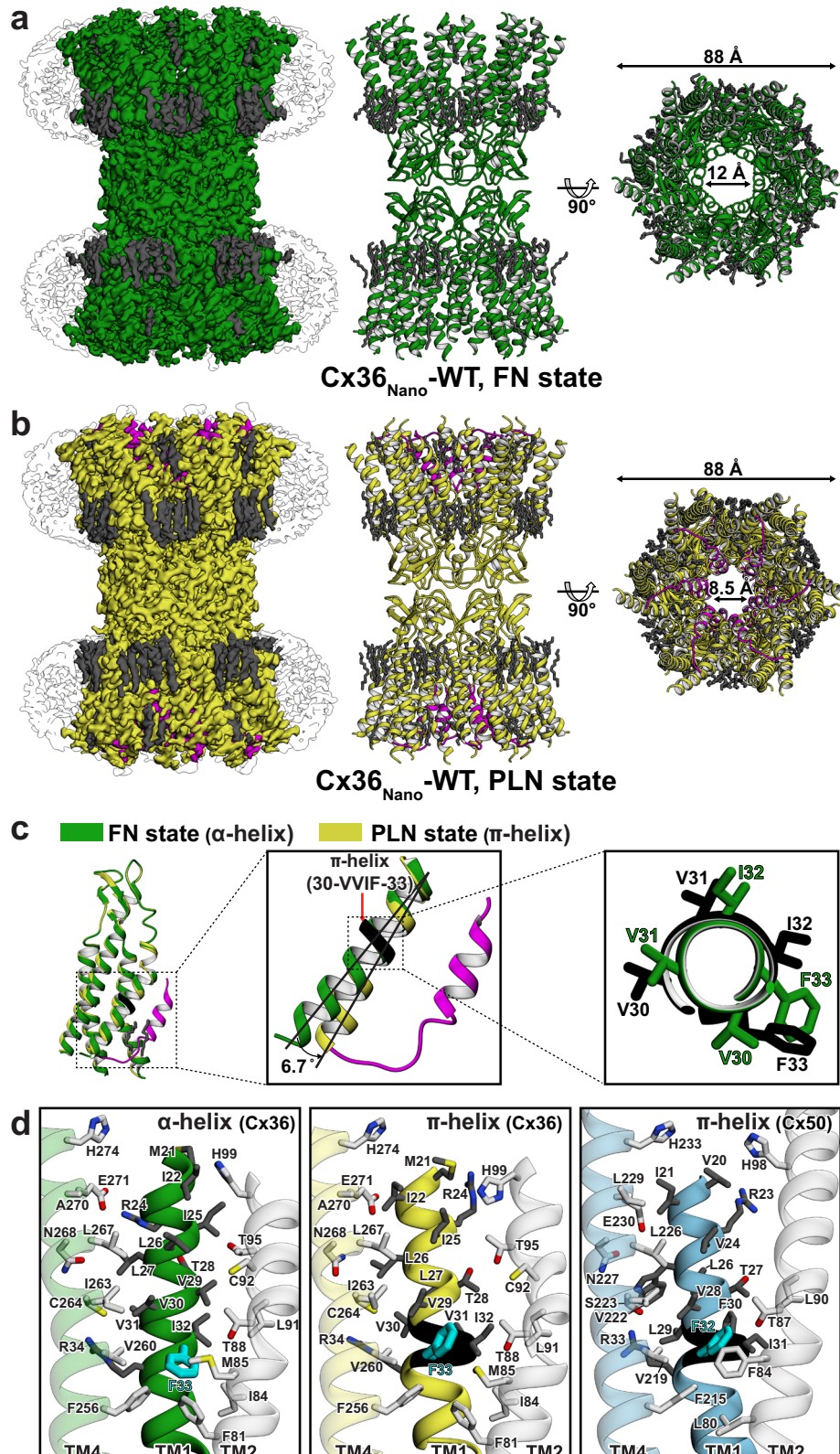

**Fig. 2 | Structural comparison of Cx36 FN and PLN states in lipid nanodiscs.**
**a**, **b** Cryo-EM reconstruction map and ribbon representation of Cx36$_{Nano}$-WT in the FN and PLN states, respectively. The acyl chains, nanodiscs, and NTHs, are colored in deem gray, white, and magenta, respectively. **c** Structure alignment of protomers of the FN and PLN states, colored in green and yellow, respectively (left). TM1 of the PLN state is bent towards the channel pore at -6.7°, and NTH binds to the hydrophobic surface of TM1 (middle), mediated by the α-to-π structural transition (right). The π-helix is colored black. **d** Detailed interactions around TM1 of Cx36 in FN (green) and PLN states (yellow), and Cx50 in the PLN state (sky blue). The TM2 of the neighboring protomer is represented as a white ribbon. Phe33 of Cx36 and its corresponding residue in Cx50 (Phe32) are represented as cyan sticks.

and Cx46/50 structures, the corresponding loop is not flexible, suggesting that it may play an important role in maintaining the PLN conformation. Therefore, we investigated the structure of the NTH-TM1 loop of Cx36 to determine whether it affects the overall NTH conformation and, thus, the funnel-shaped vestibule structure formed by the six NTHs.

In the structure of the Cx36 PLN state (Cx36$_{Nano}$-WT), the NTH-TM1 loop was likely stabilized by its close intramolecular interactions with TM2 and TM3. In particular, all carbon atoms of Gln17 interact with the phenyl ring of Phe93 from TM2 at a distance of 3.7–4.4 Å, while the amide group of Gln17 interacts with Gln202 from TM3 at a distance of ~3.8 Å in the hydrophobic environment formed by Phe93, Phe198, and Tyr199 (Fig. 3a, bottom box). Because Gln17 is a residue unique to Cx36, these interactions and the consequent loop structure are likely Cx36-specific (Supplementary Fig. 4).

In the structural comparison of the region from NTH to the following loop in Cx36, Cx43, and Cx46/50, we found that the Ala14 insertion in Cx36 not only slightly pushed the C-terminal region of NTH toward the pore center (Fig. 3e), but also caused ~90° rotation of the highly conserved Val15 (Fig. 3a–d, bottom box). This caused the C-terminal regions of NTHs to be loosely packed on TMDs and opened a large space surrounded by hydrophobic residues from TM1 and two adjacent NTHs (Fig. 3a, f, box 1). Notably, we identified two acyl chain densities that filled this space (Fig. 3f, box 1, 2), suggesting that a lipid molecule may contribute to the PLN conformation of Cx36. Val15 directly interacts with both acyl chains, whereas the corresponding residues of Cx43 (Val14), Cx46 (Ala14), and Cx50 (Val14) are involved in the tight intramolecular interaction with TM1 and TM2, and Gln15 (Cx43 and Cx46) and Asn15 (Cx50) are placed at the position corresponding to Val15 in Cx36 (Fig. 3f–i, box 1). Therefore, Cx43, Cx46, and Cx50 do not form a hydrophobic pocket at this location, suggesting that the lipid binding between NTHs in the PLN conformation may be a feature unique to Cx36.

Trp4, which is conserved in 18 human connexins, plays a major role in NTH-TMD interaction. In the Cx43, Cx46, and Cx50 GJC structures, the indole ring of Trp4 is stuck in a deep groove between two adjacent TM1s. Although Cx36 GJC has a similar hydrophobic groove for Trp4 binding, the exact binding site in the groove differs between these GJCs. Since Cx36 NTH is located ~2 Å further from the channel entrance than Cx43, Cx46, and Cx50 NTHs, Trp4 binds more closely to the residues (Ala39 and Ile40) at the 6th helical turn of TM1 than the 5th helical turn (Fig. 3e). Since Cx36 NTH is located slightly further from TM1 and closer to the pore center than Cx43, Cx46, and Cx50 NTHs in the PLN state, the overall volume of the pore lumen in Cx36 GJC is much smaller than those of Cx43 and Cx46/50 GJCs[22,26] (Supplementary Fig. 5a, b). We also compared the NTH structure of Cx36 with that of Cx26 where the conserved tryptophan (Trp3) is exposed to the pore lumen[21] (Supplementary Fig. 9g). Since the high-resolution structures of Cx36, Cx43, Cx46, and Cx50 commonly showed the binding of the conserved tryptophan to the groove between two adjacent TM1s, Cx26 would have a similar binding mode of Trp3 in its PLN conformation. Because the density map of Cx26 NTH was very weak, the structure of the N-terminus including Trp3 in Cx26 would have been difficult to be correctly modeled.

On the pore surface formed by six NTHs in a Cx36 connexon, each Glu8 residue forms a salt bridge with Arg9 on a neighboring NTH (Fig. 3f, box 2), thereby contributing to the stability of the PLN conformation. Although these residues are not conserved in most human connexins, Cx25 and Cx37 have Arg8/Asp9 and Glu8/Lys9 pairs, respectively, suggesting that their homomeric GJCs may have similar salt-bridge networks in the PLN state (Supplementary Fig. 4). Taken together, our structural analysis suggests that the longer NTH-TM1 loop affects the overall NTH conformation, reflecting Cx36-specific structural features.

## The structurally hetero-junctional Cx36$_{Nano}$-WT GJC

The 3D classification of Cx36$_{Nano}$-WT GJCs indicated only two different GJC conformations, in which both hemichannels were in the same state (PLN or FN) (Fig. 2). However, the conformations of the two opposing hemichannel regions are not strongly dependent on each other because the structures of the extracellular loops are nearly identical between the PLN and FN states. Therefore, we reasoned that the particles in a single Cx36$_{Nano}$-WT sample might include conformationally hetero-junctional GJCs that contain two opposing hemichannel regions in different conformations.

To determine the structure of hetero-junctional GJC, we first processed the particle images with a focus on the hemichannel region. In the results of the 3D classification (Supplementary Fig. 3), three classes (classes 2, 6, and 7; referred to as PLN group) showed clear map densities for PLNs and an empty hole through the hemichannel region, whereas the other five classes (referred to as the FN group) exhibited neither PLN densities nor a central hole. We collected hemichannel particles (~13% of the total particles) in the FN group, traced them back to the original GJCs, and removed duplicates. This process led to the selection of 14,155 GJC particles with two hemichannel regions each belonging to the PLN and FN groups, 3D reconstruction of which with C6 symmetry produced a conformationally hetero-junctional GJC structure at 3.34 Å resolution. Compared with the GJC structures in the full PLN and FN states, the map density of the hemichannel region in the PLN state was weaker because of the smaller number of particles used for reconstruction, but that in the FN state was significantly improved in the cytoplasmic half of the TMD, as we expected. These analyses suggest that two opposing hemichannel regions are not structurally interdependent and may have different conformations. More importantly, the hetero-junctional GJC structure provides an avenue for analyzing the distinct conformations of NTHs in the same GJC.

It should be noted that the hetero-junctional GJC structure is not the main conformation in the Cx36$_{Nano}$-WT GJC sample, but is the average of only 13% selected particles, which are still conformationally heterogeneous. In addition, it is currently unclear which structure of Cx36 GJC represents the fully open state. Although the purified GJC sample was in the condition without transjunctional voltage where this channel showed the maximum conductance[27], other conditions such as physiological membrane lipids may be needed to induce the fully open channel.

## The pore in the FN state is completely obstructed by lipids

In the hetero-junctional structure of Cx36$_{Nano}$-WT GJC, the hemichannel in the FN state had two flat layers of density blobs blocking the pores around the cytoplasmic and extracellular ends of TM1 or TM2 (Fig. 4a and Supplementary Fig. 6a). These densities inside the pore were as strong as those of the lipids filling the nanodisc and were not observed in the other hemichannels in the PLN state. To avoid artifacts from C6 symmetry imposed during 3D reconstruction, we confirmed that the pore-obstructing densities were also present in the density map reconstructed without imposed symmetry. The features of the pore-obstructing densities, including double layers, flatness, and a thickness of ~4 nm, are highly consistent with those observed in the pore-occluded structures of the Innexin-6 hemichannel and pannexin 1 channel[18,19], suggesting that Cx36 GJC in the FN state may also be obstructed by lipids. We also investigated the Cx36$_{LMNG}$-BRIL structure and confirmed that the double layers of pore-occluding densities are consistent in the detergent environment (Supplementary Fig. 2f).

To investigate the possibility that the pore-obstructing densities correspond to flexible NTHs and CLs, we performed a structural study on two Cx36 mutants: Cx36$_{Nano}$-ΔN8, in which 7 N-terminal residues (residues 2–8) are deleted, and Cx36$_{Nano}$-BRIL-ΔN16, in which 15 N-terminal residues (residues 2–16) are deleted and the CL is replaced with BRIL. The cryo-EM maps of mutant Cx36 GJCs

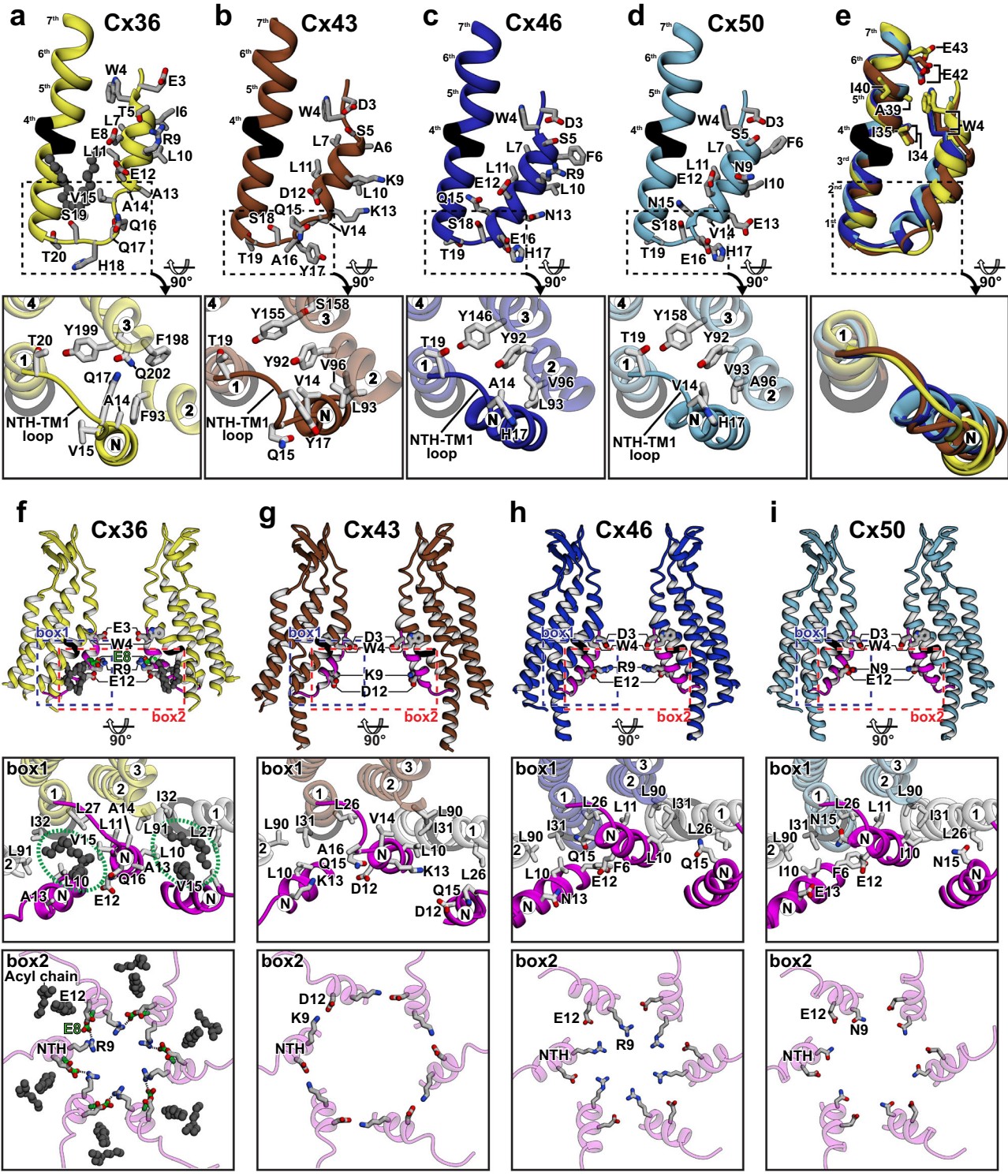

**Fig. 3 | Unique structural features of the Cx36 PLN state. a–e** Detailed structures of NTHs and the NTH-TM1 loops in Cx36 (**a**), Cx43 (**b**), Cx46 (**c**), and Cx50 (**d**), and their structural alignment (**e**). **f–i** Ribbon representation of two facing protomers in Cx36 (**f**), Cx43 (**g**), Cx46 (**h**), and Cx50 (**i**) hemichannel regions. NTHs and acyl chains in the lipid-binding pockets are represented as magenta ribbons and dark gray ball-and-chain models, respectively. The detailed interactions in box 1 (blue dashed line) and box 2 (red dashed line) are represented in the bottom panels.

Compared with Cx43, Cx46, and Cx50, Cx36 has two unique structural features: hydrophobic pockets between two NTHs (**f**, box 1, green dotted circles) and an intermolecular salt bridge between two neighboring NTHs (**f**, box 2, black dotted line). The Cx36 Glu8, which forms an intermolecular salt bridge by interacting with Arg9, is colored green. Other neighboring protomers and the π-helix are shown as white and black ribbons, respectively.

reconstructed with and without symmetry imposition showed similar pore-occluding densities (Fig. 4b, c and Supplementary Fig. 6b, c), indicating that the channel pore is obstructed in the absence of NTHs and/or CLs, likely by lipids. However, in Cx36-WT GJC, flexible protein domains such as NTH, CL, or CT may partly contribute to the cytoplasmic layer of the pore-occluding densities. Especially, amphipathic NTHs are highly possible to interact with the surface of the cytoplasmic lipid layer.

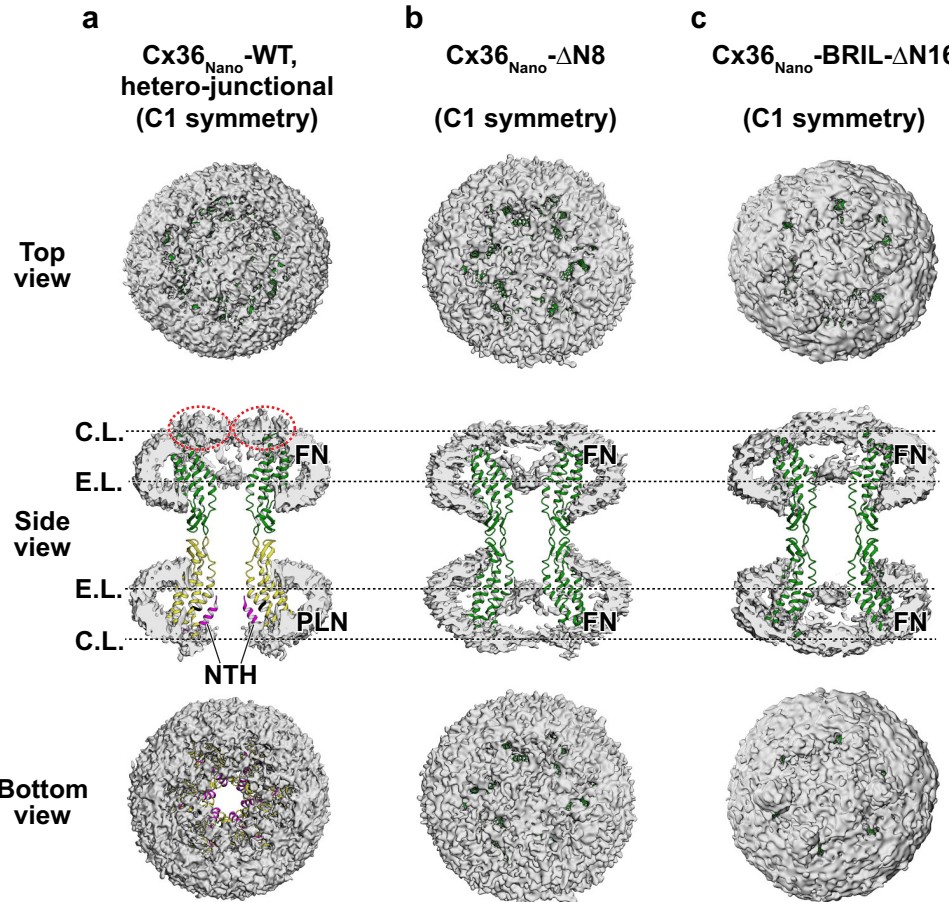

**Fig. 4 | Structural characterization of the pore-occluded state in lipid nanodiscs. a–c** Top, cross-sectioned side, and bottom views of the cryo-EM reconstruction map with C1 symmetry imposition. Ribbon representations of the hetero-junctional Cx36$_{Nano}$-WT (**a**), Cx36$_{Nano}$-ΔN8 (**b**), and Cx36$_{Nano}$-BRIL-ΔN16 (**c**) GJCs are also shown. C.L. and E.L. denote cytoplasmic layer and extracellular layer, respectively. The lipid nanodiscs and pore-occluding lipids are displayed as white densities. Cx36 proteins in the FN and PLN states are colored green and yellow, respectively. The NTHs and the π-helix spanning residues 30–33 are colored magenta and black, respectively. Red dotted circles indicate possible locations of flexible NTHs.

## The acidic pore surface of Cx36 GJC confers strong cation selectivity

Previous electrophysiological studies have shown that Cx36 GJC preferentially transfers small cationic molecules with a diameter of <10 Å, such as ethidium bromide (net charge +1), EAM-1 (net charge +1), and EAM-2 (net charge +1) fluorescence dyes[28,29]. We showed that Cx36 GJC in the PLN state has a pore with a solvent-accessible diameter of ~8.5 Å, which is sufficient for the passage of these dye molecules as well as hydrated cations such as K⁺ (~6.6 Å), Na⁺ (~7.2 Å), and Ca²⁺ (~8.2 Å)[30]. Therefore, the PLN conformation likely represents the open state of the channel.

To understand the mechanism underlying cation selectivity, we first analyzed the surface charge distribution of Cx36 GJC. We found two acidic surface bands formed by six NTHs (the cytoplasmic acidic band) and the C-terminal regions of six TM1s (the extracellular acidic band), respectively. The cytoplasmic acidic band is composed of Glu3, Glu8, and Glu12, but the acidity of Glu8 is compromised by Arg9 (Fig. 3f and Supplementary Fig. 5c). While Glu3 and Glu12 are conserved in the majority of human connexins including Cx46 and Cx50, Glu8 is not conserved at all (Supplementary Fig. 4). The extracellular acidic band is composed of Asp47, Asp48, and Glu49, and its acidity is stronger than that of the cytoplasmic band because of the absence of surface-exposed basic residues in this region (Supplementary Fig. 5c). Although Asp48 and Glu49 are strictly conserved in the human connexin family,

Asp47 is conserved only in Cx31.9, Cx62, and Cx30 (Supplementary Fig. 4).

Next, to determine how the pore surface properties affect ion selectivity, we performed MD simulations of this channel in two lipid bilayers containing 1-palmitoyl-2-oleoylphosphatidylcholine (POPC) and 150 mM NaCl. Since 12 CLs are considerably long and located close to the channel entrance and exit, they may prevent ions from accessing the entrance and escaping at the exit, resulting in a substantially reduced ion-transfer rate. Therefore, we tested two models of Cx36 GJC in the PLN state with and without the CL, respectively. We used the CL model predicted by Alphafold, which mostly consists of unstructured loop regions. Six CLs in each hemichannel region were highly flexible during the simulation and formed large pores/gaps for ions to freely diffuse through.

In the 1.2 μs simulation without transmembrane potential for the channel with the CL, Na⁺ ions gathered at the two extracellular acidic bands, increasing the local concentration to >2 M (Supplementary Fig. 7a), while few Cl⁻ ions (~0.03 M) were found inside the pore (Supplementary Fig. 7b). This suggests that the diffusion of anions through these pores is very limited because of the acidic bands. When we applied a transjunctional potential of 200 mV, the Na⁺ current was 0.007 nA with maximum flux at the extracellular acidic band on the cathode side (Supplementary Fig. 7c), whereas the Cl⁻ current was undetectable (Supplementary Fig. 7d). Therefore, the single-channel conductance in this simulation was 35 pS, which was higher than the

experimental conductance of 5–15 pS[31,32]. This discrepancy may be because the viscosity of the standard water model used in this simulation is approximately 30% of the experimental viscosity[33–35], resulting in diffusion coefficients and currents overestimated by a factor of three. For example, the current values computed using similar MD simulation methods showed a deviation from the experimental value by a factor of two to three[34,36,37]. If this also applies to the above simulation of Cx36 GJC, the calculated conductance changes to ~12 pS, which is within the range of the experimental one.

In contrast, the 0.6 μs simulation for Cx36 GJC without the CL showed the $Na^+$- and $Cl^-$ currents of 0.023 nA and 0.001 nA, respectively, resulting in the conductance of 120 pS (Supplementary Fig. 7i, j). This data confirms that the strong cation selectivity is caused by the pore surface property, not by the CL, and suggests that the CL might be able to regulate ion permeability independently of the gating regulation by NTHs and lipids. However, these data need to be carefully interpreted because our MD simulations were not based on the exact information of the structural dynamics and intermolecular/intramolecular interactions of six CLs in each hemichannel region.

Taken together, these data confirm the strong cation selectivity of Cx36 GJC shown in previous electrophysiological studies and provide mechanistic insights into the process of ion transfer through the channel. However, the experimentally determined conductance (5–15 pS) could not be well explained by our structures and MD simulations in the current state where the CL structure and dynamics are unknown.

## Dynamic conformational changes in individual protomers in a single Cx36 GJC

Recently solved structures of Cx43 GJC showed that 12 protomers in a single channel underwent independent conformational changes, resulting in structurally diverse GJC particles in the protein sample[26]. To determine whether a single Cx36 GJC also contains conformationally heterogeneous protomers, we performed single-subunit-focused 3D classification (see "Methods" for details) using the initial consensus map from the Cx36Nano-WT dataset (Supplementary Fig. 8a)[38,39]. The results showed that ~43% of the protomers were in the FN conformation (FN protomers) (Supplementary Fig. 8a; classes 5, 6, and 8) and ~57% were in the PLN conformation (PLN protomers) (Supplementary Fig. 8a; class 1–4 and 7). Next, we traced back the protomers in the PLN conformation (referred to as PLN protomers) to the GJC particles to which they originally belonged, and investigated the distribution of PLN protomers in each GJC particle. We found that Cx36 GJCs (85,080 particles) had various FN:PLN compositions ranging from 0:12 (the full PLN state, ~0.4%) to 12:0 (the full FN state, ~0.05%) (Supplementary Fig. 8b). Consistent with the previous Cx43 GJC structures, PLN protomers in the Cx36 GJC structure showed a normal distribution, suggesting that they are randomly distributed throughout GJC particles (Supplementary Fig. 8b). We also analyzed the positional distribution of the PLN protomers in a hemichannel region and compared the percentages of all possible PLN/FN compositions with the predicted values when they were randomly distributed (Supplementary Fig. 8c). The result showed that the experimental and predicted values were not significantly different. Thus, we concluded that the conformational changes in individual protomers are not strongly affected by the conformations of neighboring protomers in a hemichannel region. However, we cannot exclude the possibility of the structural interdependency between neighboring PLNs caused by the intermolecular salt bridges between neighboring Glu8 and Arg9 residues, which may be too weak to be identified by our analyses with the current dataset.

It should be noted that one FN class showed a putative NTH density of which position and orientation are similar to those of gate-covering NTHs (GCNs) shown in Cx31.3 and Cx43 structures (Supplementary Fig. 8d). This was unexpected because the hydrophobic residues of TM2 to maintain the GCN conformation in Cx43 (Tyr92, Leu93, Val96, Phe97, and Met100) and Cx31.3 (Thr95, Leu96, Val99, Ile100, and Trp103) were not conserved in Cx36 (Supplementary Fig. 4). We performed hemichannel-focused 3D classification with C6 symmetry imposition and identified one hemichannel class with the GCN-like densities. However, the refined 3D map did not show the map densities of NTHs that are sufficiently clear to build the structural model (Supplementary Fig. 8d). The unclear NTH densities would be primarily because substantial noises from PLN and FN protomers were included in the final 3D map. In addition, NTHs in this conformation might be still flexible due to weak interaction with TM2 and lie on the intracellular layer of the pore-occluding lipids.

While the protomers in Cx36Nano-WT GJCs mainly showed the PLN conformation (57%), those in Cx36LMNG-BRIL GJCs were completely in the FN conformation (Supplementary Fig. 10). We were curious whether the shift of the structural equilibrium is caused by the change in lipid/detergent environments or the replacement of CL with BRIL. Thus, we performed cryo-EM experiments of Cx36-WT in LMNG/CHS (Cx36LMNG-WT) and Cx36-BRIL in soybean lipid nanodiscs (Cx36Nano-BRIL) and obtained 3.2 Å and 3.4 Å consensus maps, respectively. Next, we conducted protomer-focused 3D classification to understand the structural equilibrium in each GJC sample and found that Cx36LMNG-WT and Cx36Nano-BRIL GJCs showed the FN/PLN ratios of 71:29 and 12:88, respectively.

While Cx36-WT and Cx36-BRIL in soybean lipids were mainly in the PLN conformation (Supplementary Fig. 10), those in detergents were mainly or completely in the FN conformation, suggesting that detergents contributed to the PLN-to-FN transition more than lipids. Since the GJC pore in the full FN conformation is funnel-shaped, detergents with large head groups may be more tightly packed at the intracellular layer in the pore, resulting in increased FN protomer populations. However, although BRIL fusion had a considerable effect on the PLN-FN equilibrium in GJC, the role of CL or BRIL is still unclear because the replacement of CL with BRIL had different effects on the structural equilibrium of GJC in detergents and lipids: increasing and decreasing the FN/PLN ratio, respectively.

## Structural comparison of Cx36 FN and PLN protomers reveals α-to-π helix transition in TM1

In the structure of Cx36Nano-WT GJC in the full PLN state, we identified two π-helices in the TM1 of each protomer, consisting of highly conserved residues (Supplementary Fig. 9b). Although they have been previously observed in the Cx43 and Cx46/50 GJC (Fig. 2d and Supplementary Fig. 9d, f), their functions are not yet clearly understood. While the typical α-helix is characterized by main chain hydrogen bonds between residues that are set four residues apart in the sequence (i + 4), the π-helix has an additional amino acid per helical turn (i + 5), resulting in a "bulge" structure[40]. When we compared the structures of TM1s in Cx36 PLN and FN protomers, one π-helix at the C-terminus of TM1 39–42 was observed in both protomer states (Supplementary Fig. 9a, b). This π-helix is also found in currently available connexin structures such as Cx26, Cx31.3, Cx43, and Cx46/50 (Supplementary Fig. 9a–f, cyan)[4,22,23,26], suggesting that it may play an important structural role in maintaining a 30.8° kink in TM1. However, the other π-helix in the middle of TM1 (residues 30–33) was only observed in the PLN protomer, and the corresponding region was α-helical in the FN protomer, indicating that this region undergoes an α-to-π-helix transition during the conformational change of NTH from FN to PLN state (Fig. 2c and Supplementary Fig. 9a–d).

The α-to-π helix transition in the middle of TM1 causes a slight bending (~6.7°) of TM1 toward the channel pore, and a large helical rotation (~65°) in the cytoplasmic half of TM1, including residues Thr20 to Phe33 (Fig. 2c and Supplementary Fig. 9a, b). This results in significant changes in the interactions between TM1 and other TM helices. First, the tight intermolecular interaction of Phe33 in TM1 with Phe81

and Ile84 in TM2 is loosened, while new intramolecular interactions with Val29, Ile32, and Val260 are established. Second, Arg34 moves slightly towards Glu210 in TM3 (Supplementary Fig. 9a, b, cross-section 1). The ionic interaction between these two residues in the hydrophobic core of TMD is the only strong interaction between TM1 and TM3, and is thus thought to be crucial for the structural integrity of TMD. Therefore, the dislocation of Arg34 during the α-to-π-helix transition may increase structural stability via a closer interaction with Glu210 of TM3. Third, Ile32, which is completely exposed to the channel pore in the FN state, moves to interact with Phe33 of TM1 and Leu91 of TM2 in the adjacent protomer (Supplementary Fig. 9a, b, cross-section 1). Fourth, the intermolecular salt bridge between Arg24 of TM1 and Glu271 of TM4 is broken (Supplementary Fig. 9a, cross-section 2) and an intermolecular hydrogen bond between Thr28 of TM1 and Thr95 of TM2 is formed (Supplementary Fig. 9b, cross-section 1). Residues 30–33 of Cx36 are highly conserved in the connexin family, and the corresponding residues of Cx43 have also been found to undergo an α-to-π-helix transition (Supplementary Fig. 9c, d, cross-section 1)[26]. Therefore, we concluded that the transition and consequent structural changes in TMD might be a common feature of all connexin homologs.

## Discussion

Cx36 GJC mediates ionic transmission through vertebrate electrical synapses, which collaborate with chemical synapses to dynamically shape brain function[41]. Like chemical synapses, electrical synapses are plastic, and their modifications reconfigure the neural circuits[42]. The strength of the electrical connection varies between different neurons and on distinct timescales from milliseconds to days[43]. In a single synapse, the strength can be regulated not only by changing the total number of GJCs with a quick turnover rate of ~3 h, but also by modulating the ratio of open and closed GJCs or the probability of each GJC being open[44]. Known regulatory mechanisms include inhibition by $Mg^{2+}$ and phosphorylation, and the combined role of calcium, calmodulin, and $Ca^{2+}$/calmodulin-dependent protein kinase II[8,14,25]. In this study, the structural analysis of $Cx36_{Nano}$-WT GJC indicated a distinct gating mechanism that involved the possible role of membrane lipids.

Membrane lipids are currently not considered direct regulators of GJC gating, probably because previous structural studies on GJCs have not clearly identified lipids inside the channel pore. In addition, the direct role of a specific lipid is difficult to study at the cell or tissue level, since it requires the delivery of insoluble molecules into the cell membrane or the channel without using detergents. Based on previous studies[22,45], it can be concluded that NTHs strongly bind to TMDs, forming a hydrophilic pore; thus, membrane lipids cannot enter the pore. However, the structures of Cx36 GJC in lipid nanodiscs in this study revealed two different conformations (PLN and FN) of this channel (Fig. 2), corresponding to the open and closed states, respectively. Because these structures were produced from a single grid sample, we believe that we captured the equilibrium between the two states. This was further supported by the identification of intermediate states with various combinations of the two different protomer conformations in a single channel (Supplementary Fig. 8). Importantly, in the FN hemichannel of the hetero-junctional GJC structure (Fig. 4a), the channel pore was completely obstructed by the cytoplasmic and extracellular layers of lipids, whereas the channel pore was open in the PLN hemichannel region. These structural data suggest that, at least in soybean lipids, Cx36 is conformationally flexible, and lipids can dynamically move into and out of the channel. However, we cannot completely exclude the possibility that the pore-occluding lipids in the FN conformation were artificially introduced during the cryo-EM sample preparation,

While we consistently observed the conformational equilibrium of NTHs in the structures of $Cx36_{LMNG}$-WT, $Cx36_{Nano}$-WT, and $Cx36_{Nano}$-BRIL GJCs, only the FN conformation was observed in $Cx36_{LMNG}$-BRIL GJC, indicating that the equilibrium completely shifted to FN. When we examined the inner surface of the channel pore of $Cx36_{LMNG}$-BRIL GJC, strong map densities were observed for the acyl chains of LMNG, but not the head group, and for the sterol ring of CHS, but not the succinyl group. These data suggest that a specific lipid environment with high cholesterol content might induce the FN conformation of the channel, leading to the complete obstruction of the pore. A previous electrophysiological study using HeLa cells showed that most Cx36 GJCs (>99%) were closed[46]. In this context, judging by our structural data, the closed channels might be mostly in the FN conformation, obstructed by membrane lipids. However, this does not exclude the possibility that the channels are in the PLN state and mostly plugged by CL or other regulatory proteins. For example, the channel closing by $Ca^{2+}$-loaded calmodulin has been extensively studied, and the calmodulin-cork model has been proposed[47], although this model needs to be confirmed by structural studies in the future. In addition, direct pore-plugging by NTHs was recently observed in Cx26 GJC at acidic pH[5]. Since Cx26 GJC was reconstituted in amphipol A8-35, the role of detergents or lipids was likely excluded.

This study raises several interesting questions. First, it is still unclear whether the Cx36 gap junctions in physiological cell membranes (e.g., cryo-frozen electrical synapses) contain GJCs in both PLN and FN conformations, and whether the ratio of the two conformations is related to the strength of the electrical synapse. Investigating this would require the advancements in in situ electron cryotomography (cryo-ET) and sub-tomogram averaging technologies.

Second, it is unclear how lipids diffuse into and out of channel pores. The Innexin-6 hemichannel structure shows a large gap between the protomers through which the lipids in the inner leaflet can pass[18]. Consistent with this, the recently solved structures of Cx43 GJC showed that a small but sufficiently large gap for lateral lipid transfer is created during conformational change[26]. Although the corresponding gap is much smaller in Cx36 GJC, we observed much weaker intermolecular interaction at the gap in the PLN state than in the FN state (Fig. 2d). Since the independent conformational transition of each protomer in Cx36 GJC is possible, there may be an intermediate conformation with a larger gap, which we have not been able to classify from our data yet. Interestingly, we have observed top views of either tetradecameric GJCs or heptameric hemichannels in all the datasets collected in this study (Supplementary Fig. 1f). Although we cannot exclude the possibility that these channels are artificially created during protein purification or nanodisc reconstitution, the fact that they were commonly observed in both detergent and lipid environments suggests that they might be formed in various detergent/lipid environments including cell membranes. This observation also suggests that Cx36 GJC or hemichannel might be interchangeable between two different assemblies with six- and sevenfold symmetries. The interchange would require a temporary disruption of the channel assemblies, which may allow the diffusion of lipids into and out of channel pores (Supplementary Fig. 11). This idea needs to be validated by cryo-ET or single-molecule experiments.

Third, we identified a unique hydrophobic pocket between the NTHs in Cx36 GJC in the PLN state (Fig. 3f, box 1). Although two acyl chains bound to the pocket likely contribute to the structural stability of the PLN conformation, it is unclear whether this interaction is crucial for channel opening. There is also a strong possibility that other hydrophobic signaling molecules may strongly bind to this pocket to increase the probability of Cx36 GJC opening (Fig. 5).

Fourth, it is unclear whether the conformational change of Cx36 GJC discovered in this study is involved in its voltage-gating mechanism. Since the charged residues in NTHs function as voltage sensors[44,48–50], the transjunctional voltage likely induces the conformational transition of NTHs. Because two hemichannel regions in a GJC face each other, their responses to the applied transjunctional voltage would be opposite to each other. In the case of Cx36 GJC, the

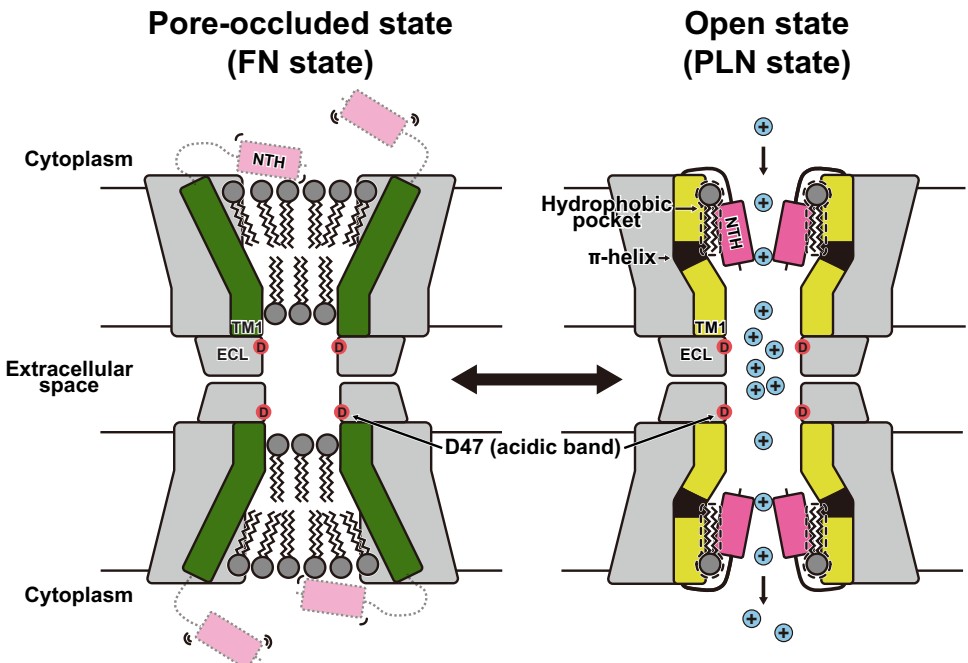

**Fig. 5 | Overall model and unique structural features of Cx36.** Schematic representation of Cx36 in the FN and PLN states. TM1 in the FN and PLN states, π-helix spanning residues 30–33, and NTH, are colored green, yellow, black, and magenta, respectively. The phospholipids, cations, and surface-exposed Asp47 at the ECL1 of the acidic band, are represented by the tailed gray, blue, and red circles, respectively. The flexible NTH of the FN state is outlined by a light-gray dashed line. The channel pore of Cx36 in the FN state is occluded by membrane components, whereas that of Cx36 in the PLN state is open. Some lipid molecules are bound to the hydrophobic pockets (dashed line) around the NTHs in the PLN state. Cx36 undergoes a conformational change from the FN to PLN state via the α-to-π transition in TM1.

full PLN conformation might be formed in one hemichannel region and the full FN in the other, and vice versa, resulting in the channel closing in both cases. Amphipathic NTHs in the FN state might not be totally out of the pore pathway but lie on the intracellular layer of the pore-occluding lipids so that they can sense the voltage field. GJCs generally show the maximum conductance when the transjunctional voltage difference tends to zero, and our experimental conditions for the structural study have no transjunctional voltage. Therefore, although Cx36 GJC in the full PLN conformation showed substantial ionic current and cation selectivity in our MD simulation, the conformationally dynamic state of Cx36 GJC might form a larger pore and show maximum conductance. For example, a hemichannel region with four PLN protomers and two consecutive FN protomers without lipids or with highly mobile lipids has a pore substantially bigger than that with only PLN protomers. Therefore, the frequent conformational change of one or two NTHs in the full PLN conformation could result in an ion-transfer rate higher than no conformational change. Alternatively, the full PLN state might be the maximum conductance state, and its maintenance might require other factors that are abundant in cells but not included in our experimental system. A specific membrane lipid or amphipathic molecule might strongly bind to the pocket between neighboring NTHs in the PLN conformation and stabilize the conformation.

The dynamic conformational change observed in Cx36 GJC, as previously shown in Cx43 GJC, may help large molecules pass through the channels. Since a pore with a diameter of ~12 Å can be formed during the dynamic conformation change (Supplementary Fig. 12), large cellular metabolites such as ATP or dinucleotides might pass through the channel more efficiently in the structural equilibrium state than in the full PLN state. The binding affinity of NTH for TMD to form the PLN conformation may have been optimized to maintain the structural equilibrium in the cell membrane for ease of the gating regulation. Therefore, the binding of specific signal molecules could shift the equilibrium to the full PLN or FN state for the channel opening or closing. However, the hypothesis of thermodynamic equilibrium interconnecting different states of Cx36 GJC, particularly considering the role of lipids as structural mediators of channel blocking, remains to be validated by functional experiments. Unfortunately, there is no method to measure the conformational state of GJCs during a patch clamp recording. In addition, it is difficult to design mutational studies for the measurement of ionic current in a specific channel conformation. Any mutation to fix NTH or TM1 in a specific conformation would inhibit the conformational dynamics of the channel, which may be crucial for lipid exclusion at its initial assembly. To prove this hypothesis, further structural and electrophysiological studies are needed to find molecules or conditions that greatly shift the conformational equilibrium of the channel to the FN and PLN states, respectively, and investigate their effects on the ionic current through the channel.

## Methods

### Expression and purification of Cx36-WT and Cx36-ΔN8 in the human cell expression system

A synthetic gene fragment encoding the full-length human Cx36 (*GJD2*) was purchased from Integrated DNA Technologies and inserted into pX plasmid vector as previously reported[51]. The Cx36-WT gene was expressed in transiently transfected HEK293E cells as a fusion protein C-terminally connected with a human rhinovirus (HRV) 3 C cleavage site, an enhanced yellow fluorescence protein (eYFP) tag, 10×His-tag, and Rho-1D4 epitope tag (8 amino acids of TETSQVAPA). The nucleotide sequences of the primers used for molecular cloning are listed in Supplementary Table 3. The pXY-Cx36-WT plasmid was transfected using 25-kDa linear polyethylenimine (Polysciences) into HEK293E cells, which were grown at 37 °C in suspension in Dulbecco's modified Eagle's medium (DMEM) with glucose (4500 mg/l) without calcium (WELGENE) supplemented with 5% fetal bovine serum (FBS). Dimethyl

sulfoxide (Amresco) was added immediately after the transfection to a final concentration of 1%, and the temperature was lowered to 33 °C. At 48 h after transfection, tryptone (Amresco) was added to a final concentration of 0.5%. At 96 hours after transfection, the cells were centrifugated at 500×$g$ for 20 min.

The harvested cells were resuspended in a buffer A [20 mM CAPS pH 10.5, 500 mM KCl, and 2 mM $\beta$-mercaptoethanol] supplemented with 10% glycerol and 1 mM phenylmethylsulfonyl fluoride (PMSF) and lysed using a Dounce homogenizer (Bellco) with a tight (B) pestle (25–30 strokes). The membrane fraction was isolated by high-speed centrifugation at 42,600×$g$ for 1 h. The membrane pellet was resuspended using a WiseTis homogenizer (Daihan Scientific Co., Ltd.) in 50 ml buffer A supplemented with 1 mM PMSF and 0.5/0.05% (w/v) LMNG/CHS (Anatrace). After incubation for 2 h with slow rotation, the sample was mixed with 2.5 ml neutralization buffer containing 1 M Tris (pH 7.0) to lower the sample pH to ~7.5 and centrifugated at 42,600×$g$ for 1 h. The supernatant was mixed with adipic acid dihydrazide-agarose resin (Sigma) conjugated with Rho-1D4 antibody (University of British Columbia) in an open column (Bio-Rad) and incubated with slow rotation at 4 °C for 1 h. The resins were settled down in the column and washed twice with 10 column volumes (CVs) of buffer B [20 mM HEPES pH 7.5, 500 mM KCl, 2 mM $\beta$-mercaptoethanol] supplemented with 0.005/0.0005% (w/v) LMNG/CHS. The bound proteins were incubated at 4 °C overnight with the addition of excess HRV 3 C protease (~0.25 mg) to remove the C-terminal eYFP-Rho-1D4 tag from Cx36 and eluted from the resin. The eluted Cx36 proteins were concentrated and further purified using Superose 6 Increase 10/300 column (Cytiva) equilibrated with a buffer B. Peak fractions were pooled, concentrated to ~2 mg/ml, flash-frozen in liquid nitrogen, and stored at −80 °C for nanodisc reconstitution and EM grid preparation. Protein purity and quality were assessed by SDS-polyacrylamide gel electrophoresis (SDS-PAGE).

The Cx36-ΔN8 construct was created by polymerase chain reaction (PCR) using pXY-Cx36-WT as a template. The resulting plasmid pXY-Cx36-ΔN8 was transfected into HEK293E cells, and the mutant Cx36 was expressed and purified using the same protocol as for Cx36-WT.

### Expression and purification of Cx36-BRIL and Cx36-BRIL-ΔN16 in the baculovirus expression system

The full-length Cx36 gene was subcloned into pEG BacMam expression vector to produce pEG-Cx36[52], which was designed to express Cx36 as a fusion protein with the C-terminal eYFP and FLAG tags (8 amino acids of DYKDDDDK) or only with the FLAG tag. This plasmid was further engineered to create pEG-Cx36-BRIL through the deletion of the CL region (residues 109–187) by PCR and the insertion of the cytochrome b562RIL gene fragment (BRIL; residues 21–128) by the conventional enzymatic DNA assembly method[53]. *E. coli* DH10Bac strain (Gibco, cat #10361012) was transformed with pEG-Cx36-BRIL to produce Cx36-BRIL Bacmid, which was transfected into *Spodoptera frugiperda* (Sf9) to produce baculovirus containing the Cx36-BRIL expression cassette, according to manufacturer's instructions. Human embryonic kidney (HEK) 293E cells and *Spodoptera frugiperda* (Sf9) cells were obtained from ATCC (CRL10852 and CRL-1711).

Sf9 cells were grown at 27 °C in suspension in ESF293 medium (Expression Systems) supplemented with 0.06 mg/ml penicillin G and 0.1 mg/ml streptomycin (Sigma). At 72 h after infection, the cells were centrifugated at 500×$g$ for 10 min. The membrane was solubilized with buffer C [20 mM HEPES pH 7.5 and 200 mM NaCl] supplemented with 5 mM ethylenediaminetetraacetic acid (EDTA), protease inhibitors (1 mM PMSF, 2 μg/ml leupeptin, 2 μM pepstatin A, and 2 μM aprotinin), and 1% (w/v) LMNG for 2 h at 4 °C. The insoluble fraction was removed by high-speed centrifugation at 100,000×$g$ for 1 h. The soluble fraction was twofold diluted with buffer C supplemented with 0.01% LMNG and

mixed with monoclonal anti-FLAG antibody agarose beads (Wako chemicals, cat #016-22784). The mixture was incubated with slow rotation at 4 °C for 5 h. The resins were settled down in the column and washed three times with 10 CVs of buffer D [20 mM HEPES pH 7.5, 200 mM NaCl, 0.01% LMNG, and 0.001% CHS]. The bound proteins were eluted with buffer D supplemented with 450 μg/ml FLAG peptide (sigma) at 4 °C overnight. The eluates were concentrated and further purified using Superose 6 Increase 10/300 column equilibrated with buffer D. Peak fractions were pooled, concentrated to ~2 mg/ml, flash-frozen in liquid nitrogen, and stored at −80 °C for EM grid preparation.

The Cx36-BRIL-ΔN16 construct was created by PCR using pEG-Cx36-BRIL as a template. The resulting plasmid pEG-Cx36-BRIL-ΔN16 was used for Bacmid and baculovirus production. The mutant Cx36 was expressed and purified, using the same protocol as for Cx36-BRIL.

### Reconstitution of Cx36-WT, Cx36-ΔN8, and Cx36-BRIL-ΔN16 in lipid nanodiscs

Purified Cx36-WT, Cx36-ΔN8, and Cx36-BRIL-ΔN16 proteins were reconstituted into membrane scaffold protein (MSP1E1) nanodiscs containing soybean polar lipids extract. Soybean lipid extract powder (Avanti), mainly composed of phosphatidylcholine, phosphatidylethanolamine, phosphatidylinositol, and phosphatidic acid was solubilized in 5/0.5% (w/v) LMNG/CHS and incubated at room temperature for overnight to make a clear lipid stock solution in ~10 mg/ml. The pET28a plasmid containing MSP1E1 gene was obtained from Addgene (plasmid #20062). The membrane scaffold protein (MSP1E1) was expressed and purified as previously described[54]. The purified Cx36 sample was mixed with the soybean polar lipid extract stock (~10 mg/ml) at the molar ratio of Cx36 to lipids of 1:100 and incubated at 4 °C for 1 h. Then, the purified MSP1E1 protein was added to the final molar ratio of Cx36:MSP1E1:lipids of 1:0.5:100. The mixture was incubated with slow rotation at 4 °C for 30 min. For the removal of detergents and the protein-nanodisc reconstruction, Bio-Beads SM2 resin (Bio-Rad, resin 100 mg) was pre-washed with buffer E [20 mM HEPES pH 7.5, 250 mM NaCl, and 2 mM $\beta$-mercaptoethanol] and added to the Cx36-lipid-MSP mixture. After 4 h with gentle rotation, the supernatant was collected, and another round of detergent removal was done with the pre-washed 100 mg Bio-Beads SM2 resin. The mixture was incubated overnight at 4 °C with gentle rotation. To remove insoluble particles, the supernatant was filtered through a membrane with a pore diameter of 0.22 μm (Millipore). The filtered sample was further purified by size-exclusion chromatography using Superose 6 Increase 10/300 column equilibrated with buffer E for Cx36-WT and Cx36-ΔN8 or buffer C for Cx36-BRIL-ΔN16. Fractions containing both Cx36 and MSP1E1 were pooled, concentrated to ~2 mg/ml, flash-frozen in liquid nitrogen, and stored at −80 °C for EM grid preparation. Protein purity and quality were assessed by SDS-PAGE.

### Fluorescence-detection size-exclusion chromatography

The oligomerization state of Cx36 was evaluated by fluorescence-detection size-exclusion chromatography (FSEC)[55]. C-terminally eYFP-tagged Cx36-WT and Cx36-K238E were expressed as described above and solubilized with buffer C supplemented with 5 mM EDTA, protease inhibitors, and 1% (w/v) LMNG for 2 h at 4 °C. The insoluble fraction was removed by high-speed centrifugation at 100,000×$g$ for 1 h. The soluble fraction was loaded onto Superose 6 increase 5/150 GL size-exclusion column, and the fluorescence signal was monitored by a fluorescence detector (Agilent Technologies, cat #DEAEJ00102).

### Cryo-EM specimen preparation and data collection

Three microliters of purified Cx36 proteins (1–2 mg/ml) in detergents or nanodiscs were applied onto a negatively glow-discharged (15 mA current, 60 s) holey carbon grid (Quantifoil R1.2/1.3 Cu 200 mesh). In case of Cx36$_{LMNG}$-WT and Cx36$_{Nano}$-WT, to improve orientation

diversity, phenylalanine was added at the final concentration of 50 mM. The grid was blotted and plunge-frozen in liquid ethane using Vitrobot Mark IV (ThermoFisher Scientific, USA) at 4 °C and 100% humidity. Cryo-EM images were collected at Institute for Basic Science (IBS) and Korea Basic Science Institute (KBSI), using Krios G4 (ThermoFisher Scientific;TFS, USA) equipped with BioQuantum K3 detector (Gatan Inc, USA) and Titan Krios G2 (FEI, USA) with Falcon 3EC detector, respectively. Automated data acquisition was performed in electron counting mode using EPU software (TFS, USA). More details are described in Supplementary Table 2.

### Image processing and reconstruction

The cryo-EM image processing was performed with cryoSPARC version 3.1[56] or Relion 3.1 softwares[57] (Supplementary Fig. 13). The Cx36$_{LMNG}$-WT dataset was processed using cryoSPARC (v.3.1.0). Patch-based pre-processing (Patch motion correction & Patch CTF estimation) was performed for the dataset containing 3780 movies. Next, 609,292 particles picked by reference-based auto-picking were extracted into 360-pixel boxes. After three rounds of 2D classification, good particles were re-extracted into 512-pixel boxes and subjected to two rounds of 2D classification. Finally, 51,480 particles were used for 3D refinement with D6 symmetry, which yielded an EM density map at a resolution of 3.2 Å.

The Cx36$_{LMNG}$-BRIL dataset was processed using cryoSPARC (v.3.1.0). Patch-based pre-processing (Patch motion correction & Patch CTF estimation) was performed on the dataset containing 2228 movies. Next, 435,817 particles picked by reference-based auto-picking were extracted into 340-pixel boxes. After three rounds of 2D classification, good particles were re-extracted into 540-pixel boxes and subjected to three rounds of 2D classification. Finally, 70,095 particles were used for 3D refinement with D6 symmetry, which yielded an EM density map at a resolution of 2.2 Å.

The Cx36$_{Nano}$-BRIL dataset was processed using cryoSPARC (v.3.1.0). Patch-based pre-processing (Patch motion correction & Patch CTF estimation) was performed on the dataset containing 499 movies. Next, 71,238 particles picked by reference-based auto-picking were extracted into 400-pixel boxes. After three rounds of 2D classification, good particles were re-extracted into 512-pixel boxes and subjected to three rounds of 2D classification. Finally, 10,611 particles were used for 3D refinement with D6 symmetry, which yielded an EM density map at a resolution of 3.4 Å.

The Cx36$_{Nano}$-WT-ΔN8 dataset was processed using cryoSPARC (v.3.1.0). Patch-based pre-processing (Patch motion correction & Patch CTF estimation) was performed on the dataset containing 3463 movies. Next, 766,823 particles picked by reference-based auto-picking were extracted into 360-pixel boxes. After six rounds of 2D classification, good particles were re-extracted into 540-pixel boxes and subjected to three rounds of 2D classification. Finally, 460,806 particles were used for 3D refinement with D6 symmetry, which yielded an EM density map at a resolution of 3.2 Å.

The Cx36$_{Nano}$-BRIL-ΔN16 dataset was processed using cryoSPARC (v.3.2.0). Patch-based pre-processing (Patch motion correction & Patch CTF estimation) was performed on the dataset containing 2402 movies. Next, 426,090 particles picked by reference-based auto-picking were extracted into 512-pixel boxes. After five rounds of 2D classification, good particles were used for the generation of 3D initial model. Finally, 39,444 particles were used for 3D refinement with D6 symmetry, which yielded an EM density map at a resolution of 3.4 Å.

### Focused 3D classification for the Cx36$_{Nano}$-WT dataset and structure determination of full PLN, full FN, and structurally hetero-junctional Cx36 GJCs

The Cx36$_{Nano}$-WT dataset was processed using Relion (v.3.1). Beam-induced motion correction and CTF estimation of 7250 movies was performed using MotionCor2 version 1.2.6 and Gctf version 1.18,

respectively. Next, 689,081 particles picked by reference-based auto-picking were extracted into 360-pixel boxes. After six rounds of 2D classification, good particles were re-extracted into 540-pixel boxes and subjected to additional three rounds of 2D classification. After five rounds of 3D classification, final 85,080 particles were used for 3D refinement with D6 symmetry imposition, which yielded a 3.19 Å consensus map. The consensus map was used for further processing focused on GJC or hemichannel. The overall workflow is presented in Supplementary Fig. 3.

In the approach focused on GJC, 85,080 particles were subjected to 3D skip-alignment classification with D6 symmetry imposition. To increase the accuracy of classification, we applied a mask covering GJC for a focused 3D classification. The first 25 iterations with Regularization parameter (T) of 20 and the second 10 iterations with increasing $T$ value ($T = 40$) and GJC mask were performed. In the resulting three classes, Class 1 and 2, respectively, including ~50% and ~15% GJC particles showed clear map densities of pore-lining NTHs (PLNs), whereas the NTH densities were unclear in Class 3 with ~35% particles. To improve the map quality, particles in two GJC classes were used for 3D refinement with D6 symmetry imposition and local angular searches, respectively. After sharpening, the result showed a 3.05 Å GJC map with full PLN state and a 3.16 Å GJC map with full FN state.

In the approach focused on hemichannel, 85,080 particles were used for 3D skip-alignment classification, as previously described in ref. [38]. Cx36 hemichannel in the PLN state and Cx43 hemichannel in the GCN state (PDB 7F92) were combined and used to create a hemichannel mask. Two masks covering each hemichannel of the consensus GJC map were used to generate the subtracted particles in two opposite orientations. After re-centering and re-extraction into 300-pixel boxes, the subtracted particles were subjected to 3D classification ($K = 6$; $K$ is the number of classes to classify) with C1 symmetry imposition to align into one orientation. Three hemichannel classes (Class 2, 4, and 6) with good quality side views were subjected to focused 3D refinement (C1 symmetry) with a soft mask covering hemichannel to obtain a 3.65 Å map reconstructed from 140,255 particles. With this new consensus map, we performed 25 iterations of 3D classification ($T = 10$). Then, the additional 10 iterations were performed with increasing $T$ value ($T = 20$) and a soft mask covering only the cytoplasmic half of hemichannel to classify more accurately for NTH conformations. To obtain a structurally hetero-junctional GJC map, hemichannel particles in five classes (Class 1, 3, 4, 5, and 8) in the FN state were chosen and traced back to their original GJC particles, and the redundant particles were removed. Finally, 14,155 GJC particles were subjected to 3D refinement with 1.8° angular sampling, producing a 4.9 Å unsharpened map refined with C1 symmetry imposition and a 3.3 Å sharpened map refined with C6 symmetry imposition. The GJC structures clearly showed PLN hemichannel on one side and FN hemichannel on the other side.

### Protomer-focused classification for the Cx36$_{Nano}$-WT dataset

D6 symmetry expansion was performed with the 3.19 Å consensus map from the Cx36$_{Nano}$-WT dataset. All protomer particles were subtracted using a mask covering a single protomer. The subtracted protomer particles were subjected to focused 3D classification ($K = 8$, $T = 20$) with the protomer mask and without orientation search. In the resulting eight classes, three classes showed the flexible NTH conformation (FN protomer), and five classes showed the pore-lining NTH conformation (PLN protomer) (Supplementary Fig. 8a).

### Analysis of the distribution of PLN protomers in GJC particles

To investigate how many PLN protomers were included in each GJC particle, we analyzed the metadata file generated by protomer-focused 3D classification as previously reported[39]. The metadata includes the class number of each protomer particle and the identification number (ID) of the original GJC particle. Each GJC ID should be found 12 times in

the metadata due to D6 symmetry expansion. The metadata was sorted by protomer class number, and GJC IDs were collected only for the two classes with the PLN conformation. The number of repetitions of each GJC ID in the collection indicates the number of PLN protomers in the GJC particle. We counted the number of GJCs at each of 13 different ratios of FN to PLN protomers (0:12 to 12:0) to produce the final distribution graph in Supplementary Fig. 8b. To analyze relative protomer positions in each hemichannel represented in Supplementary Fig. 8c, we used the rotation angle information for the rotation of each GJC particle during the symmetry expansion, which is recorded in the metadata file for each PLN protomer. More detailed information is described in the previous report in ref. [26].

## Model building and refinement

All structural models with acyl chains and/or CHS molecules were built in Coot program[58,59]. All models do not include CL (Ala102-Glu193), and CT (Trp277-Val321), and all models except the PLN state of Cx36$_{LMNG}$-WT, Cx36$_{Nano}$-WT and Cx36$_{Nano}$-BRIL do not include NTH (Met1-His18), due to weak map density. The structure of Cx36$_{LMNG}$-BRIL GJC was initially solved and used as a reference for modeling other structures in the FN state. The head groups of lipids and detergents could not be modeled due to weak map density. All structures were refined using phenix.real_space_refine[60] in PHENIX software and visualized using UCSF Chimera[61].

## MD simulation protocol

We performed all MD simulations using the GROMACS package[62]. We used the CHARMM36 force field in the Gromacs format downloaded from the website of the Mackerell group: the CHARMM36m for proteins[63], the CHARMM36 for lipid molecules[64], the CHARMM-modified TIP3P model for water molecules, and the CHARMM36 standard ion parameters. In combination with the CHARMM force fields, we employed the CUFIX corrections to improve the charge-charge interactions among ions and charged side chains[65]. Van der Waals forces were computed using a 10- to 12-Å switching scheme. Long-range electrostatic forces were calculated using the particle-mesh Ewald summation scheme[66] of a 1.2-Å grid spacing and 12-Å real-space cutoff. The time step was two femtoseconds by constraining covalent bonds to hydrogen in non-water using the LINCS[67] and water molecules using the SETTLE algorithm[68].

## MD preparation of Cx36 embedded in a lipid bilayer

We used the structural model of Cx36$_{Nano}$-WT in the PLN state for MD simulation, in which the CL and CT regions are missing. To prepare the GJC model composed of Cx36 with the CL, we manually reconstructed the unstructured CL of each Cx36 chain (residues 102 to 196) using the corresponding region in the predicted Cx36 model (Q9UKL4) from the Alphafold Protein Structure Database[69]. Near the transmembrane domain of each hemichannel of Cx36, we placed a lipid bilayer of a 1-palmitoyl-2-oleoyl-*sn*-glycero-3-phosphatidylcholine (POPC), followed by the removal of lipid molecules overlapping with the channel. We immersed the resulting complex of Cx36 and the double bilayer systems in an explicit solution of 150 mM NaCl. The final system contained a channel, 734 POPC lipids, 135,625 water molecules, 450 Na ions, and 462 Cl ions in a periodic hexagonal box ($a = b \approx 14$ nm, $c \approx 34$ nm, $\alpha = \beta = 90°$, $\gamma = 60°$). We energy-minimized each system for 5000 steps and equilibrated it for 70 ns under a constant surface tension–constant temperature (NP$\gamma$T) ensemble at zero surface tension ($\gamma = 0$)[70] and 300 K temperature[71]. We simulated each of the assembled systems for 400 ns in total (200 ns at a voltage bias of 0 mV with position restraints on the experimentally determined heavy atoms followed by 200 ns at a voltage bias of 200 mV). For the measurements of ionic currents through the two channels composed of Cx36 with and without the CL, we performed the simulation under 200 mV for 1.2 and 0.6 μs, respectively, starting from the final structure of the equilibration.

During MD simulations, we saved atomic coordinates every 20 ps. Using the saved trajectory, we computed the three-dimensional density-flux map and visualized the results as described in a previous report by Yoo and Aksimentiev[72].

The Cα root mean square deviations (RMSDs) of Cx36 GJC with and without the CL converged to about ~4 Å and ~3.5 Å, respectively, suggesting that these channels remained structurally stable under a thermal fluctuation (Supplementary Fig. 7f, k). The root mean square fluctuation (RMSF) using the trajectory more than 200-ns (Supplementary Fig. 7f, i) was ~2 Å for all residues except those in six CLs and several terminal residues, suggesting that this channel does not have particularly dynamic structural motifs and are stable in lipid bilayers.

## Reporting summary

Further information on research design is available in the Nature Portfolio Reporting Summary linked to this article.

## Data availability

The data that support this study are available from the corresponding authors upon request. Cryo-EM density maps have been deposited in Electron Microscopy Data Bank (EMDB) under accession codes EMD-33270 (Cx36$_{LMNG}$-BRIL), EMD-34822 (Cx36$_{Nano}$-BRIL), EMD-33256 (Cx36$_{LMNG}$-WT), EMD-34856 (Cx36$_{LMNG}$-WT (C6 symmetry)), EMD-34857 (Cx36$_{LMNG}$-WT (C1 symmetry)), EMD-33315 (Cx36$_{Nano}$-WT in PLN state), EMD-33327 (Cx36$_{Nano}$-WT (C6 symmetry)), EMD-33328 (Cx36$_{Nano}$-WT (C1 symmetry)), EMD-33274 (Cx36$_{Nano}$-ΔN8 (D6 symmetry)), EMD-33275 (Cx36$_{Nano}$-ΔN8 (C1 symmetry)), EMD-33254 (Cx36$_{Nano}$-BRIL-ΔN16 (D6 symmetry)) and EMD-33255 (Cx36$_{Nano}$-BRIL-ΔN16 (C1 symmetry)). Atomic coordinates for models have been deposited at the Protein Data Bank (PDB) under accession codes 7XKT (Cx36$_{LMNG}$-BRIL), 7XKK (Cx36$_{LMNG}$-WT), 8HKP (Cx36$_{LMNG}$-WT (C6 symmetry)), 7XNH (Cx36$_{Nano}$-WT in PLN state), 7XNV (Cx36$_{Nano}$-WT (C6 symmetry)), 7XL8 (Cx36$_{Nano}$-ΔN8 (D6 symmetry)) and 7XKI (Cx36$_{Nano}$-BRIL-ΔN16 (D6 symmetry)). The previously published structures used for structural comparison were obtained from the Protein Data Bank under accession codes 2ZW3, 6L3T, 7F94, 7JKC, and 7JJP. The raw data of MD simulation generated in this study have been deposited in the Zenodo OpenAIRE database and is available under accession code 7608483. Source data are provided with this paper.

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

## Acknowledgements
This work was supported by the Suh Kyungbae Foundation (SUHF-18010097 to J.-S.W.), the National Research Foundation (NRF) grants funded by the Ministry of Science and ICT (NRF-2018R1C1B6004447 to J.-S.W. and NRF-2020R1A2C1101424 to J.Y.), and by the National Super-computing Center (KSC-2020-CRE-0080 to J.Y.). This work was also supported by Samsung Science & Technology Foundation and Research (SSTF-BA2101-13) and the NRF grants (2015M3D3A1A01064919, 2022R1A2B5B02002529, 2022R1A5A6000760) to H.H.L.

## Author contributions
J.-S.W. and H.H.L. conceived this project; H.J.C. purified BRIL-fused Cx36 and solved the initial structure; S.-N.L. purified Cx36-WT and solved the structures in lipids; H.J. and B.R. performed electron microscopy; H.J.C., S.-N.L., H.J., B.R., and H.-J.L. performed EM image processing; S.-N.L., H.J.C., and J.-S.W. performed model building; M.K. and J.Y. performed MD simulations; S.-N.L., H.J.C., J.-S.W., and H.H.L. wrote the manuscript.

## Competing interests
The authors declare no competing interests.
