## [Peer Review File · Nature Communications]

Cryo-EM structures of human Cx36/GJD2 neuronal gap junction channelReviewers' Comments:

Reviewer #1:

Remarks to the Author:

NCOMMS-22-19702-T

Cryo-EM structures of human Cx36/GJD2 neuronal gap junction channel

Lee et al.

Cx36 is known to express in mammalian brain neurons, and it is thought to be one of the main components of electrical synapses. It was first identified as a neuron-specific gap junction protein in mammals, but now other connexins are reported to express in neurons along with Cx36. The signal transduction by electrical synapses in mammals has long remained in mystery as chemical synapses are the predominant way of neurotransmission. Since the discovery of Cx36, its importance has been recognized in mammals, and the structural study of Cx36 is essential to understanding the electrical coupling of cells.

Lee et al. analyzed the eight structures of Cx36 using single-particle cryo-EM at 2.2-3.4 Å resolution and constructed an atomic model. BRIL-fusion improved the particle orientation in the hole of cryo-grids and enabled high-resolution structure determination. The structures of Cx36 have been determined both in detergent and in nanodiscs. In nanodiscs, the pore of Cx36 was occupied by densities supposed to be lipids, which is similar to other large pore channels like CALHM2, CALHM4, and innexin-6. The authors interpret that the PLN conformation represents an open channel, and the FN with the pore lipids represents a closed-form. The structures are well determined with great caution and reliability, providing essential insights into the gating mechanism of Cx36 gap junction channels. The discussion has been provided with caution.

On the other hand, the novelty of the connexin channel provided by the Cx36 structure is somewhat weak, and there are some limitations of this study. Specifically, there are concerns regarding the map interpretation, and the functionality mainly relies on the MD simulations. The authors should address these concerns, and substantial revision is appreciated.

Major concerns:

1.

The FN conformation has been determined in both detergent and nanodiscs, while the pore obstruction density was found only in nanodisc. The atomic structures corresponding to the protein parts are very similar to each other. The authors' interpretation is that FN represents closed because of the occluded pore in nanodisc. However, the map of Cx36LMNG-WT looks like a PLN conformation while the N-terminal helix densities are weak. This appears obvious when the Gaussian filter of 1.5~2 sigma is applied to the map as in the attachment. The authors claim that the FN state of Cx36LMNG-BRIL was not caused by the replacement of CLs with BRIL, but it looks like the effect of BRIL itself. If so, the interpretation is different, and this should be clarified. This is related to the following comments regarding the functionality, and please see below.

The other thing is that the map of Cx36LMNG-BRIL looks to have the double layer obstruction as pointed out in the structure in nanodisc when the Gaussian filter is applied. This should be mentioned in the text, or one may think the pore of Cx36LMNG-BRIL is open. Could they be carried-over lipids? Also, EMD-33315 (Cx36Nano-WT (PLN state)) and EMD-33327 (Cx36Nano-WT (heterotypic)) are not original but masked maps. These have to be replaced with unmasked maps like others.

2.

What are the functionalities of the Cx36 containing BRIL, ΔN8, and BRIL-ΔN16? Related to the previous concerns, this is important for map interpretation. Given the authors' interpretation, the functionalities of ΔN8 and BRIL-ΔN16 are probably non-functional because only FN conformation was found. A plausible possibility is that BRIL eliminates the Cx36 function resulting in the pore occlusion in both LMNG and nanodisc. Once the channel function of the constructs used for cryo-EM is evaluated, these concerns will likely be addressed, and the misinterpretation of the map with

ambiguity will be reduced.

Do the authors have a cryo-EM map of Cx36nano-BRIL-WT in nanodiscs that should be compared with the structure in detergent (Cx36LMNG-BRIL)? Depending on the functionality of Cx36-BRIL, the map in nanodisc would be consistent with the Cx36nano-WT (PLN + FN mixture) or Cx36nano-deltaN8 (only FN).

3.

The α -to-n-helix transition of TM1 and the formation of a unique NTH-TM1 loop structure are the characteristic features specific to Cx36. However, it is unclear how these lead to the Cx36 specific functionality, such as the cation selectivity and small conductance. The Cx36 specific functional properties seem to be accounted for only by the electrostatic potentials and MD simulation. The comparison and consistency with other connexin channels such as Cx43 (ref. 26) and Cx31.1 (ref.23) should be shown to avoid an intentional interpretation.

4.

The structure of this cytoplasmic loop used for the MD simulation is forming an assembly on the cytoplasmic entrance and is possibly not well-founded, which is different from the Cx31.1 model used for MD before by the same authors. Alphafold has generated this portion, but one would expect it to interfere with ion permeation with this CL arrangement. In fact, the single-channel conductance of Cx36 estimated by MD was 3.5 pS, which is, as the authors mentioned, small as ref. 31 shows 5-10 pS, and another study shows 10-15 pS (Srinivas et al. Neurosci. 1999 19(22):9848-55). Is this MD reasonable to estimate the conductance? Also, is this analysis consistent with Cx43, Cx31.1 or other connexins?

5.

The interpretation of lipid diffusion in and out of the pore is similar to those proposed for Innexin-6, Pannexin-1 and CALHMs. As mentioned by the authors, the inter-subunit space of Cx36 is small and insufficient for lipid migration as far as the solved structures are considered. The dynamism of the Cx36 channels (Line 396) is a bit weak to assert the lipid diffusion. Did the author ever find any top-view 2D class averages that lose 6-fold symmetry? Or were there classes showing symmetry breaking during 3D classification even if it did not reach high-resolution?

6.

Line 321,
"PLN: 17%, FN: 0.5%"

This means that ~82% of junction channels are a mixture of PLN and FN. How were most of these particles (~82%) processed in the structure analysis? Are the features of mixed PLN and FN not visible in the image processing? For example, one can expect that the volume of the pore occluding densities increases as the number of FN subunits increases.

In the individual subunit model (Oh et al. JGP 116, 13-32, 2000), Oh et al. point out that only one protomer is sufficient to initiate the voltage gating going toward closure. Is it possible to tell how many PLNs would cause the hemichannel to be open?

7.

The authors have shown multiple structures of nanodisc reconstituted Cx36, including Δ N8, BRIL- Δ N16, but the density in the pore seen in FN seems to vary. Some of those look like the pore aggregate densities where the ball-and-chain mechanism proposed for Cx26 pH gating (Khan et al. 2020, Cell Rep. 31,107482), and in that model, those were interpreted as the N-termini occluding the pore. Are these contour level issues, or do they all have bilayer properties?

8.

Line 429,
The final paragraph of the Discussion is quite speculative. The authors mention that the structure of Cx36 with Mg²⁺ did not show a significant difference. For the discussion about the contribution of

Mg²⁺ to channel activity inhibition, additional positive and significant data are required. I recommend that this paragraph is deleted.

Minor points:

Abstract, last line

"plasticity of electrical synapse."

This work does not account for the synaptic plasticity directly. This expression seems misleading.

Line 77,

"the channel pores were blocked by lipid bilayers"

It is likely, but not conclusive. I recommend milder expression like pore occlusion.

Line 81,

"...determine the distinct structural features"

This phrase is uncomfortable because MD in this work does not show distinct structural features but supports the functional properties of Cx36, for instance, cation selectivity and small conductance. This should be revised.

Line 103,

"Notably,..."

It is misleading if this sentence means that the Cx36-BRIL function is normal or comparable to that of WT Cx36. Ref. 20 demonstrates that the deletion of the cytoplasmic loop of Cx36 significantly reduced the junctional conductance (WT: 2.54 μ S to mutant: 0.39 μ S) while the response to acidosis is shifted. When discussing open or closed, the absolute value of conductance is more important. As described above, the structures of Cx36LMNG-WT and Cx36LMNG-BRIL look different, and the conductance values in Ref. 20 seem to account for the structural differences. This should be addressed.

Line 118,

The term of "solvent channel" is unclear. Does this mean the pathway of water molecules forming a hydrogen bond network between inside and outside the pore?

Line 140,

Ref. 27 should read ref. 21.

Line 145,

"electron density" should be revised because visible densities in cryo-EM maps are not electron density but potential.

Line 146,

"acyl chains of LMNG" What if carried-over lipids? The head group of LMNG is large. Something should be mentioned about the position, orientation, and how it binds.

Line 147,

"This suggests that CHS and/or LMNG may stabilize the free NTH..." sounds strange. FN (free N-terminus) is named after being invisible and not stabilized. "stabilize" should be revised by something like induce or cause.

Line 151,

The authors added phenylalanine to the sample grid to solve the orientation preference of the Cx36 channels. What kind of effect do the authors expect? Is there any precedent example of using phenylalanine for this problem before?

Line 160,

Why are soybean polar lipids used for making nanodiscs while Cx36 is derived from human? POPC was used for the MD simulation and is there a possibility that the different lipid components could lead to alternative conclusions?

Line 173,

"almost identical" is unclear. Is this regarding the built models? The maps look different as one has BRIL.

Line 174,

"it showed no clear map densities of CHS...."

Does this mean CHS was pushed out during nanodisc reconstitution? Could it be that the binding of CHS in the pore observed in the structure in detergents might be artificial? Because the authors mention the contribution of CHS before, this sounds not consistent. I am wondering what occupies this space of Cx36 in nanodiscs. In line 176, "data not shown" should be avoided.

Line 221,

"due to the insertion of Ala14,....."

This reads that only Ala14 causes the N-terminal portion specific to Cx36, but in fact, other factors like Glu17 and lipid acyl chains may contribute to that conformation. This can be deleted.

Line 259,

"...N state may also be obstructed by lipids"

Is there a possibility that anything other than lipids may comprise the pore density? For example, invisible C-terminus or CL?

Line 261,

"Cx36Nano- Δ N8"

If my understanding is correct, this construct as well as Cx36Nano-WT does not have BRIL. The authors described that BRIL has changed the behavior in terms of the orientation in the hole of grids (Line 107). Is the effect of BRIL not essential for the structure determination or solving the particle orientation problem? Does it depend on nanodisc or detergent? The modification in CL is expected to affect the functionality of the Cx36 channels seriously, and if BRIL is not needed for structure determination, that would be better.

Line 289,

In MD simulations, can the movement of the lipids be traced? What if the FN structure is used as an initial model for MD?

Line 344,

" α -helix is also found in all solved"

Because this may not apply to all connexins in the future, please revise it to the moderate expression.

Line 387,

Do the N-termini of Cx36 have a voltage sensor? It is interesting if Cx36 can exhibit the gating depending on the transjunctional voltage mediated by the N-terminus. It has been postulated that the N-terminus, including the voltage sensor, should be located near the entrance of the pore to feel the electric field to respond to the transjunctional voltage (Verselis et al. Nature, 1994). If the N-terminal portion of FN is totally out of the pore pathway, it would be impossible for the N-terminus to sense the voltage field. The description of the relationship between the voltage gating property of Cx36 and the structure is preferable.

Line 503,

The information of ref. 26 should be written appropriately.

Fig.3 legend,

The sub-pannel labels should be written in the legend like "Cx36 (a) and Cx50 (b), and superposition of Cx36 and Cx50 (c)".

Reviewer #2:

Remarks to the Author:

Review Comments:

The authors present a well-written and comprehensive analysis of the previously unexplored molecular architecture of human Cx36 gap junction intercellular channels via the clever application of Cryo-EM techniques. They explore, compare, and contrast Cx36 structures with previously solved gap junction channel structures such as Cx50 and Cx26 and find features that are previously undescribed in Cx channels such as the occlusion of the pore by lipids in what they refer to as the FN state. This is significant as Cx36 is the main neuronal gap junction protein and the authors' findings illuminate new questions about the regulation of electrical synapse conductance and the potential for new studies of Cx36 structure/function.

While the study is well written and comprehensive, several minor revisions are suggested to increase clarity of the manuscript.

Minor revisions:

Overall, the paper suffers from an overuse of initialisms that are not always explained in the text. It would be beneficial, especially in the figure legends, to clearly state all initialism definitions at the end of the figure legend. For example, in Figure 1 it would be helpful to define in the figure legend: CHS, ECL, NTH, and EC.

Main text:

Line 43: The 21 connexin genes referenced need to be clarified as 21 human connexin genes.

Lines 69-71: Innexin and pannexins are discussed without referencing their relationship to Connexins. It would be helpful for the reader to explain these relationships.

Line 90: In Extended Data Figure 1a, several forms of Cx36 are compared, yet we have not yet had an introduction to them in the text. This figure was then quite confusing to parse upon first view. We suggest moving 1e to become 1a and offering a short introduction to these forms before moving on to the current 1a. In addition, the SDS-PAGE gel is completely unexplained in the figure legend. The relevance of the bands and the identity of MSP1E1 is unknown until much later in the text. Explanation is needed.

Line 95: The figure legend for Extended Data Figure 1d does not state what is highlighted in red. We assume these to be hydrophobic residues. Please correct the figure legend to explain the red color designation.

Line 172: The acronym CTL is unexplained.

Line 214: Figures are referred to out of order.

Line 290: The acronym POPC is unexplained.

Line 615: The description of cells "harvested at 500 g for 20 minutes" is an odd phrasing and confusing. Please reword. Same for line 656.

Line 975: We believe "meshes" is incorrectly represented as "mashes" here.

Line 995: The blue colors are difficult to distinguish/understand. We suggest diversifying the colors here.

Extended Data Fig 10 is not referenced in the manuscript.

Reviewer #3:

Remarks to the Author:

Review for manuscript Cryo-EM structures of human Cx36/GJD2 neuronal gap junction channel

Summary of findings:

The authors of this work aimed to derive the structure of the human gap junction channel (GJC) formed by the human Cx36. To do so, they obtained eight structures of human Cx36 GJC in its pore-occluded and open states using single-particle cryo-electron microscopy (cryo-EM). In the pore-occluded state, the channel pores were blocked by lipid bilayers and the NTHs of Cx36 were excluded from the pore. In comparison, the channel pore was completely open without any obstruction in the pore-lining NTH (PLN) state, suggesting that the binding of NTHs to the channel pore is an essential step for channel opening. By analyzing the set of determined structures, the authors of this work proposed that a dynamic (thermodynamic) equilibrium between the closed and open conformation is present for the case of the Cx39 GJC. Moreover, in the open state of the GJC, the lumen pore is narrower and more acidic than those of other connexins, suggesting an explanation for its low conductance and strong cation selectivity. On top of that, the authors describe a conformational change of the NTH in the open state of the GJC that is accompanied by an alpha-to-pi-helix transition of TM1. As a whole, the work proposed by Lee et al provides high resolution information on the conformational flexibility of the Cx36 GJC, suggesting a novel and potential role of lipids in the regulation of the channel activity that could be consistent with similar mechanism reported for both innexins and pannexins channels as well.

Manuscript assessment:

In general terms, is a very well written manuscript, using a consistent and sounding language but relying on unorthodox definitions (see below) that may be confusing for the connexin-aware academic community. This is the case of using the GJIC acronyms (i.e. gap junction intercellular channel) instead of the usual GJC (i.e. gap junction channel) and the term heterotypic channel to define a GJC whose composing hemichannels appear in different structural conformation. The experimental design is straightforward and similar to that of previous works describing the structure of innexins and pannexins channels (see manuscript) solved by single-particle cryo-electron microscopy using lipid nano-discs. Following this approach, the authors of this work produced WT and several mutant Cx39 GJC (see extended data figure 1) which were structurally analyzed. Among other interesting proposals arising from the authors work, they found that the closed structure of the channel present lipids in the lumen pore (referred as FN). This occurs simultaneously with an NTH which is absent. Therefore, the authors propose a novel regulatory role for lipids in the case of Cx39 GJC. The authors complemented their structural analyses by molecular dynamics simulation of the WT GJC to determine the conductance of the channel under a transjunctional potential difference. In doing so, they propose that the low conductance of this channel is due to both a narrower pore and an acidic charged patch that may in turn hinder ion transport. Of note, despite the relevance for their results, the authors didn't discuss that the pore lumen of the FN state, the one classified as closed, is larger than that of the closed state of the channel. Probably the most appealing but also controversial finding of this work is the existence of a dynamic equilibrium between the closed and open conformation of the Cx39 GJC. If

this proposal is to remain valid, it will certainly reconfigure the biophysical landscape of the GJC world.

After careful evaluation of this manuscript, a series of issues arise that should be thoroughly accessed before publication (see annotated manuscript):

Line 21 and 38: The authors should consider using a more standard nomenclature denoting this structure as a Gap Junction Channel (GJC). As the GJC is in fact, an intercellular molecular structure, the nomenclature suggested by the authors seems redundant.

Line 28: Which is the actual characteristic making this loop unique?

Line 44: consider replacing homology with identity

Line 65: As GJCs perform a series of fundamental biological functions, such as electrical synapse, a clear pharmaceutical strategy that can be followed to use GJC blockers as drugs is currently missing.

Lines 70 – 76: The authors should consider including a clear connection, either structural or phylogenetical, between innexins, pannexins and connexins to better support this paragraph.

Line 81: consider adding the electrostatic characteristic to the NTH binding

Line 82: consider adding TM1, or the corresponding TMHs to better define the binding region

Line 117: consider replacing homology with identity

Line 147: consider adding the electrostatic characteristic to the NTH binding

Line 237: Despite the authors use the term heterotypic to refer to a GJC where its constituents hemichannels are in different conformations, this usage could be confusing as in the field of connexins, heterotypic usually refers to channels formed by protomers belonging to different connexins. This is the case of the Cx46/50 GJC structure.

Line 251 – 252: It is well known in the field that for the case of GJC, maximum conductance arises when the transjunctional voltage difference tends to zero. This evidence seems to be contradictory to the authors statement and it should be considered for further discussion.

Line 307: Do the authors refer to C-alpha RMSD? If this the case, as the C-alpha RMSD obtained is close to the crystal structure resolution, the maintenance of the overall structure can be ascertained.

Line 382: In strict sense, Connexin based-channels are passive transporters. Therefore, they shouldn't be considered as "facilitators". The authors should consider reconfiguring this sentence to avoid misleading

Lines 386 – 389: A proper set of references is needed here.

Line 391 – 392: Despite that in ion channels there is plenty of evidence supporting the notion that a certain thermodynamic equilibrium interconnect different states of the channel, to ensure that this phenomenon is also valid for connexin-based channels, in particularly gap junction channels, a larger body of evidence is required. The authors should consider further electrophysiological experiments such as dual-cell voltage clamp protocols.

Lines 394 – 395: How the authors can be sure that the lipids blocking the pore lumen are actually a biologically relevant function and not an artefact of the applied protocol. This is particularly appealing when considering that previous works in connexins didn't find these lipids. On the other hand, if these

finding are to remain valid, the authors should discuss the role of transjunctional voltage difference regulating the presence of lipids in the channel pore.

Line 401 – 402: Despite a very appealing and highly interesting suggestion, again, a larger body of evidence is required to establish an equilibrium between the open and closed state of the channel. This is particularly relevant when the available evidence suggest that the lumen pore is highly hydrophilic. In such case, a strong free energy gradient should be compensated so to be able to change the conformation of the NTH to allow the insertion of membrane lipids. Up to the same extent, the authors should discuss the accessing pathway that membrane lipids should follow to block the lumen pore. Should they arrive throughout the transmembrane helices or following another pathway?

Lines 402 – 404: How the authors can prove that the equilibrium described is in fact a biologically relevant mechanism and not an artefact of the applied methodology. Independently of the nature of this phenomenon, either artefactual or biologically relevant, the inclusion of lipids in the lumen pore should be a continuous mechanism guided by a thermodynamic equilibrium. Therefore, the finding of intermediate states is consistent with both scenarios.

Lines 409 - 410: The authors should consider further discussion on the role of detergents altering the lipid equilibrium at the micelles. Up to want extent the alteration of the lipid equilibrium may disrupt important biophysical properties of the membrane such as area per lipid, curvature and viscosity?

Lines 414 – 415: This is a very interesting observation. The inclusion of cholesterol will certainly change several properties of the membrane. I wonder why the authors didn't included cholesterol in their protocol? Further exploration in this direction could provide insights on the role of cholesterol during connexin-based channel gating and will certainly contribute to the current knowledge.

Line 419: The authors should include a discussion comparing their results with that of Khan and collaborators (<https://doi.org/10.1016/j.celrep.2020.03.046>) where they found a ball-and-chain mechanism mediated by the NTH for pH-mediated regulation of Cx26 hemichannels. If the ball and chain mechanism will remain valid for GJCs, could it be relevant in dragging lipids into the lumen pore?

Lines 445 – 447: As the authors suggested, in the FN state, the lumen pore becomes obstructed by the presence of lipids. Therefore, it may be the case that Mg²⁺ binding is an intermediate state of the closing process. If this is the case, could the authors find these bindings in the intermediate states of the protomers?

Line 453: consider replacing inhibitory by regulatory

Lines 560 – 561: The authors should consider including a brief discussion comparing the position and orientation of the NTHs of both structures. This could be particularly relevant to shed lights on the controversy arising from the rotation of the Cx26 NTH exposing hydrophobic residues to the lumen pore.

Lines 575 – 577: For clarity, the authors should consider removing the electron density map from panel a.

Lines 596 – 597: The position of the NTH in the FN state of the channel is not clearly represented. The authors should consider identifying both the position and orientation of the NTH in this state. This is particularly relevant to obtain a better interpretation of this figure compared to figure 5.

Line 607: The authors should consider rephrasing this sentence because, in its current form, could be misleading. Despite the lumen pore in the FLN state could be empty of lipids, it is certainly filled with other molecules such as water, ions and gases.

Line 803: Do the authors refer to heterogenous instead of heterotypic? In the usual language of the field, heterotypic refers to channels composed of protomers belonging to different connexins.

Lines 855 – 857: This is certainly an unorthodox protocol. I wonder why the authors decided not to use a comparative modelling approach such as modeler to generate the coordinates of the CL, using as template the CL region predicted by AlphaFold.

AUTHORS' RESPONSE TO REVIEWERS

Reviewer #1

NCOMMS-22-19702-T

Cryo-EM structures of human Cx36/GJD2 neuronal gap junction channel
Lee et al.

Cx36 is known to express in mammalian brain neurons, and it is thought to be one of the main components of electrical synapses. It was first identified as a neuron-specific gap junction protein in mammals, but now other connexins are reported to express in neurons along with Cx36. The signal transduction by electrical synapses in mammals has long remained in mystery as chemical synapses are the predominant way of neurotransmission. Since the discovery of Cx36, its importance has been recognized in mammals, and the structural study of Cx36 is essential to understanding the electrical coupling of cells.

Lee et al. analyzed the eight structures of Cx36 using single-particle cryo-EM at 2.2-3.4 Å resolution and constructed an atomic model. BRIL-fusion improved the particle orientation in the hole of cryo-grids and enabled high-resolution structure determination. The structures of Cx36 have been determined both in detergent and in nanodiscs. In nanodiscs, the pore of Cx36 was occupied by densities supposed to be lipids, which is similar to other large pore channels like CALHM2, CALHM4, and innexin-6. The authors interpret that the PLN conformation represents an open channel, and the FN with the pore lipids represents a closed-form. The structures are well determined with great caution and reliability, providing essential insights into the gating mechanism of Cx36 gap junction channels. The discussion has been provided with caution.

On the other hand, the novelty of the connexin channel provided by the Cx36 structure is somewhat weak, and there are some limitations of this study. Specifically, there are concerns regarding the map interpretation, and the functionality mainly relies on the MD simulations. The authors should address these concerns, and substantial revision is appreciated.

RESPONSE

We greatly appreciate the reviewer for thoroughly reading our manuscript and providing valuable suggestions, which helped us improve the quality of the manuscript. To address the major concerns of the reviewer, we performed substantial amounts of structural analyses. Following the reviewer's detailed and inspiring suggestions for the proper interpretation of cryo-EM maps, we revisited all data in the previous manuscript, added the structural analysis of Cx36-BRIL in nanodiscs, and included more proper interpretation in the revised manuscript.

One of the major concerns was the potential effect of the replacement of CL with BRIL on the function of Cx36. Based on the new analysis of the Cx36^{LMNG}-WT dataset and the new structure of Cx36^{Nano}-BRIL, we recognized the possibility that the BRIL fusion partially affected the conformational population of Cx36 NTHs, probably as a consequence of removing

CLs. We carefully interpreted these data and removed any descriptions that might mislead the readers into believing that wild-type and BRIL-fusion structures are identical. In addition, we solved the structure of Cx36-BRIL in nanodiscs, which addressed several concerns of the reviewer related to the potential effect of BRIL fusion on the channel structure and function.

Again, we thank the reviewer for many constructive suggestions. During the revision process, we have tried our best to address every issue that the reviewer commented below, and we hope that our responses are clear and to the point and satisfy the reviewer.

Major concerns:

Reviewer1-Q1

The FN conformation has been determined in both detergent and nanodiscs, while the pore obstruction density was found only in nanodisc. The atomic structures corresponding to the protein parts are very similar to each other. The authors' interpretation is that FN represents closed because of the occluded pore in nanodisc. However, the map of Cx36LMNG-WT looks like a PLN conformation while the N-terminal helix densities are weak. This appears obvious when the Gaussian filter of 1.5~2 sigma is applied to the map as in the attachment. The authors claim that the FN state of Cx36LMNG-BRIL was not caused by the replacement of CLs with BRIL, but it looks like the effect of BRIL itself. If so, the interpretation is different, and this should be clarified. This is related to the following comments regarding the functionality, and please see below.

RESPONSE

We thank the reviewer for the finding of NTH maps in Cx36_{LMNG}-WT that we overlooked. Since the NTH density observed in the consensus map was not strong, we performed protomer-focused 3D classification (see Methods for details) to find how many portions of PLN protomers are included in the map. As the reviewer expected, the result indeed showed a large proportion (~29%) of PLN protomers (Supplementary Fig. 10), which was unexpectedly high because the 3D classification of the dodecameric channel showed only a single class in the full FN conformation. Since we realized that such a high portion of PLN protomer could result in a weak and disconnected map density of NTH in the 3D reconstruction map of the dodecameric channel, we thoroughly revisited all cryo-EM maps presented in the manuscript; we performed protomer-focused 3D classification for all datasets and included the result and new interpretation in the revised manuscript (Supplementary Fig. 10, Lines 364-381). In Cx36_{LMNG}-BRIL, we observed FN conformation only, suggesting that the BRIL-fusion likely affected the NTH conformation in detergents by promoting FN conformation as the reviewer expected. However, the analysis of Cx36-BRIL in nanodiscs, newly included in the revised manuscript, showed the majority of protomers were in the PLN conformation similar to that of Cx36_{Nano}-WT showed. Therefore, the effect of BRIL is not conclusive and may differ dependent on the detergent/lipid environments.

Overall, variables such as detergent/nanodiscs and WT/BRIL-fusion seem to affect the populations of FN and PLN states. Nevertheless, we consistently observed the dynamic nature of NTHs in Cx36_{LMNG}-WT, Cx36_{Nano}-WT, and Cx36_{Nano}-BRIL datasets as well as the NTH

conformation directly correlated with the pore obstruction, i.e., FN state with pore-obstruction and PLN state with pore-opening. The followings are the result of protomer-focused 3D classifications of four different datasets included in Supplementary Fig. 10 in the revised manuscript.

Supplementary Fig. 10. Protomer-focused 3D classification of Cx36_{LMNG}-WT, Cx36_{LMNG}-BRIL, Cx36_{Nano}-WT, and Cx36_{Nano}-BRIL. All protomer classes that do not show clear map density of pore-lining NTH were classified as the FN conformation.

The other thing is that the map of Cx36LMNG-BRIL looks to have the double layer obstruction as pointed out in the structure in nanodisc when the Gaussian filter is applied. This should be mentioned in the text, or one may think the pore of Cx36LMNG-BRIL is open. Could they be carried-over lipids?
Also, EMD-33315 (Cx36Nano-WT (PLN state)) and EMD-33327 (Cx36Nano-WT (heterotypic)) are not original but masked maps. These have to be replaced with unmasked maps like others.

RESPONSE

As the reviewer suggested, we analyzed Cx36 GJCs in LMNG/CHS (Cx36_{LMNG}-BRIL) with C1 symmetry imposition and added the description (Lines 259-261) that the structure appeared to have the double layer obstruction (Supplementary Fig. 2f). Since we could not resolve the identity of the pore-occluding densities, it is unclear whether they are LMNG/CHS or cellular lipids. As the reviewer suggested, we deposited both masked and unmasked maps of EMD-33315 (Cx36_{Nano}-WT (PLN state)) and EMD-33327 (Cx36_{Nano}-WT (heterotypic)).

Lines 259-261:

We also investigated the Cx36_{LMNG}-BRIL structure and confirmed that the double layers of pore-occluding densities are consistent in the detergent environment (Supplementary Fig. 2f).

[Reviewer1-Q2]

What are the functionalities of the Cx36 containing BRIL, Δ N8, and BRIL- Δ N16? Related to the previous concerns, this is important for map interpretation. Given the authors' interpretation, the functionalities of Δ N8 and BRIL- Δ N16 are probably non-functional because only FN conformation was found. A plausible possibility is that BRIL eliminates the Cx36 function resulting in the pore occlusion in both LMNG and nanodisc. Once the channel function of the constructs used for cryo-EM is evaluated, these concerns will likely be addressed, and the misinterpretation of the map with ambiguity will be reduced.

Do the authors have a cryo-EM map of Cx36_{nano}-BRIL-WT in nanodiscs that should be compared with the structure in detergent (Cx36_{LMNG}-BRIL)? Depending on the functionality of Cx36-BRIL, the map in nanodisc would be consistent with the Cx36_{nano}-WT (PLN + FN mixture) or Cx36_{nano}- Δ N8 (only FN).

RESPONSE

As the reviewer suggested, we have determined the structure of Cx36-BRIL in lipid nanodiscs (Authors' Response Fig. 1) and performed the protomer-focused 3D classification (Supplementary Fig. 10). When we analyzed the FN/PLN population of Cx36_{Nano}-BRIL-WT (~20% FN and ~80% PLN, Supplementary Fig. 10), it was similar to Cx36_{Nano}-WT (~30% FN and ~70% PLN, Supplementary Fig. 10) rather than Cx36_{Nano}- Δ N8 (only FN). This suggests that the BRIL-fusion did not largely affect the NTH conformation at least in nanodiscs containing soybean lipids. Since Cx36-BRIL and Cx36-WT GJCs showed similar dynamic conformational equilibria of individual protomers largely shifted to the PLN conformation,

the BRIL-fusion would not completely inhibit the channel activity. This is consistent with the previous report showing that Cx43 GJC with the deletion of half of CL was still partially functional.

The structures of Cx36-BRIL and Cx36-WT in detergents showed a substantial difference in the structural equilibrium, suggesting a potential effect of the BRIL fusion on the channel activity. However, neither BRIL nor CL is clearly seen in our structures, and therefore the role of CL cannot be explained by our structures. So, we believe we should not expand the scope of our manuscript to the function of CL.

We have performed electrophysiological experiments to measure the currents through transiently expressed Cx36-WT GJCs in HELA and N2A cells, but have failed to obtain intercellular currents significantly different from those of untransfected cells so far. The current through Cx36 GJC seems too small to be detected in our experimental condition. Although we do not have the functional data for the mutant Cx36 GJCs, it could be generally accepted that the deletion of NTH (Δ N16) would result in nonfunctional GJCs because the hydrophilic pore cannot be formed without NTHs. In addition, Cx36- Δ N8 should be also non-functional because it does not contain a functionally crucial residue W4. W4 in NTH is completely conserved in the connexin family. The deep insertion of W4 into the hydrophobic pocket between two TM2 helices mainly accounts for the PLN conformation. The mutation of the corresponding residue in Cx26, Cx32, Cx36, Cx37, Cx40, and Cx43 resulted in non-functional GJCs/hemichannels¹⁻⁵. These evolutionary, structural, and functional evidences strongly suggest that the tryptophan in NTH, and thus the N-terminal half of NTH, is absolutely necessary for the PLN conformation. Therefore, we believe that all specialists in this field including the reviewer would agree that both Cx36- Δ N16 and Δ N8 should be nonfunctional.

References

1. Oshima, A. *et al.* Asymmetric configurations and N-terminal rearrangements in connexin26 gap junction channels. *Journal of molecular biology* **405**, 724-735 (2011).
2. Xu, Q. *et al.* Specificity of the connexin W3/4 locus for functional gap junction formation. *Channels* **10**, 453-465 (2016).
3. Zoidl, G. *et al.* Evidence for a role of the N-terminal domain in subcellular localization of the neuronal connexin36 (Cx36). *Journal of neuroscience research* **69**, 448-465 (2002).
4. Kyle, J. W. *et al.* An intact connexin N-terminus is required for function but not gap junction formation. *Journal of cell science* **121**, 2744-2750 (2008).
5. Shao, Q. *et al.* Structure and functional studies of N-terminal Cx43 mutants linked to oculodentodigital dysplasia. *Molecular biology of the cell* **23**, 3312-3321 (2012).

Authors' Response Fig. 1. Overall structure of Cx36_{Nano}-BRIL. **a** Cryo-EM reconstruction map and ribbon representation of Cx36_{Nano}-BRIL. The density and atomic model of Cx36_{Nano}-BRIL, NTH, and acyl chain are colored green, cyan and deem gray, respectively. The ambiguous densities of lipid nanodisc and BRIL are colored white. **b** Structure superimposition of Cx36_{Nano}-BRIL (green) and Cx36_{Nano}-WT in PLN state (yellow). The NTH of Cx36_{Nano}-BRIL and Cx36_{Nano}-WT in PLN state, and π -helix of both atomic models are colored cyan, magenta and black, respectively. The structure of Cx36_{Nano}-BRIL shows PLN state and very similar to the structure of Cx36_{Nano}-WT in PLN state (RMSD of 0.22 Å between 181 C-alpha atoms).

[Reviewer1-Q3]

The α -to- π -helix transition of TM1 and the formation of a unique NTH-TM1 loop structure are the characteristic features specific to Cx36. However, it is unclear how these lead to the Cx36 specific functionality, such as the cation selectivity and small conductance. The Cx36 specific functional properties seem to be accounted for only by the electrostatic potentials and MD simulation. The comparison and consistency with other connexin channels such as Cx43 (ref. 26) and Cx31.1 (ref.23) should be shown to avoid an intentional interpretation.

RESPONSE

We agree with the reviewer, and we have added more detailed structural comparisons with Cx43, Cx31.3, and Cx46 in the revised manuscript (Lines 183-195 and 385-399, Fig. 3 and Supplementary Fig. 9). The α -to- π -helix transition of TM1 is not the unique feature of Cx36, since it is shown in Cx43 GJC. In contrast, a narrow vestibule, the NTH-TM1 loop structure, and the hydrophobic pockets between NTHs in the full PLN conformation may be unique to Cx36 because they appeared to be caused by the Ala14 insertion, which is exclusively found in Cx36.

Supplementary Fig. 9: Detailed structural changes in Cx36 and comparison with other connexin structures. a-f Structural comparison of Cx36_{Nano}-WT in the FN (a) and PLN (b) states, Cx43 in the GCN (c) and PLN states (d), Cx31.3 in GCN state (e), and Cx50 in PLN state (f). NTH, π -helix in residues 30-33, and π -helices in residues 39-42 of Cx36 (a, b) are colored in magenta, black, and cyan, respectively, and the corresponding regions of Cx43 (c, d), Cx31.3 (e), and Cx50 (f) are represented in the same colors. During the transition from the FN to the PLN state of Cx36, the cytoplasmic half (residues 20-33) of TM1 undergoes $\sim 65^\circ$ helical rotation (b). Each connexin structure was cross-sectioned at two levels indicated by dotted lines (cross-sections 1 and 2) and viewed from the extracellular side. Dramatic changes in the interaction network of TM helices are caused by the α -to- π -helix transition in TM1.

Fig. 3: Unique structural features of the Cx36 PLN state. a-e Detailed structures of NTHs and the NTH-TM1 loops in Cx36 (a), Cx43 (b), Cx46 (c), and Cx50 (d), and their structural alignment (e). f-i Ribbon representation of two facing protomers in Cx36 (f), Cx43 (g), Cx46 (h) and Cx50 (i) hemichannel regions. NTHs and acyl chains in the lipid-binding pockets are represented as magenta ribbons and dark gray ball-and-chain models, respectively. The detailed interactions in box1 (blue dashed line) and box2 (red dashed line) are represented in the bottom panels. Compared with Cx43, Cx46, and Cx50, Cx36 has two unique structural features: hydrophobic pockets between two NTHs (f, box1, green dotted circles) and an intermolecular salt bridge between two neighboring NTHs (f, box2, black dotted line). The Cx36 Glu8, which forms an intermolecular salt bridge by interacting with Arg9, is colored green. Other neighboring protomers and the π -helix are shown as white and black ribbons, respectively.

Reviewer1-Q4

The structure of this cytoplasmic loop used for the MD simulation is forming an assembly on the cytoplasmic entrance and is possibly not well-founded, which is different from the Cx31.1 model used for MD before by the same authors. AlphaFold has generated this portion, but one would expect it to interfere with ion permeation with this CL arrangement. In fact, the single-channel conductance of Cx36 estimated by MD was 3.5 pS, which is, as the authors mentioned, small as ref. 31 shows 5-10 pS, and another study shows 10-15 pS (Srinivas et al. *Neurosci.* 1999 19(22):9848-55). Is this MD reasonable to estimate the conductance? Also, is this analysis consistent with Cx43, Cx31.1 or other connexins?

RESPONSE

We thank the reviewer for valuable comments. Since the CL structure generated by AlphaFold mostly consisted of unstructured loop regions and was highly flexible during simulation, the result would not significantly change depending on the prediction method of the CL structure. We also confirmed that there were large pores/gaps between CLs for ions to freely diffuse through. However, as the reviewer pointed out, large CLs located close to the channel gate could interfere with ion permeation. This may be more significant in Cx36 GJC because Cx36 CL is much longer than those of Cx43 and Cx31.3. Hence, we performed MD simulation for Cx36 GJC with complete deletion of CL and found that the conductance (120 pS) was ~3 times higher than that with CL structures generated by AlphaFold (35 pS). This suggests that the CL may play an important role in the regulation of ion transfer, and thus the experimentally determined conductance (5-15 pS) cannot be well explained by our structures and MD simulations in the current state that the CL structure and dynamics are unknown. It should be noted that we found a crucial error in our calculation of conductance in the original manuscript; the calculated current was correctly written, but we made a mistake in the simple calculation of $G=I/V$. The conductance for the channel with CLs was not 3.5 pS but 35 pS. We have corrected the error and added new data in the revised manuscript.

We reinterpreted our data as followings based on the correction and new data. These interpretations are included in the revised manuscript (Lines 307-326).

- 1) CL may play an important role in the regulation of ion transfer.
- 2) The calculated conductance values were higher than the experimental value (5-15 pS) by a few factors. This discrepancy is usual in MD simulations because the viscosity of the TIP3P model—the standard water model for the CHARMM force field—is about 30% of the experimental viscosity¹⁻³, resulting in diffusion coefficients and currents overestimated by a factor of three. For example, current values computed using the MD simulation method showed a deviation from the experimental value by a factor of two to three^{2, 4, 5}. If this also applies to the case of Cx36, the calculated conductance of Cx36 GJC with CLs changes to ~12 pS, which is within the range of the experimental one.
- 3) However, we should not rely on the MD simulation data because the experimentally determined conductance (5-15 pS) cannot be well explained by our structures and MD simulations in the current state where the CL structure and dynamics are unknown.
- 4) Although the conductance may not have been correctly calculated in MD simulations, the cation selectivity was consistent between our simulations and previous electrophysiology data

Regarding the reliability of MD to estimate the conductance, the same MD simulation method has been used for the studies on many different channels⁶. The calculated conductances seem not to be always similar to the experimental data. For Cx31.3 hemichannel, the experimental unitary conductance was not available. For Cx43 GJC, the full open state is controversial. So we cannot directly compare the experimental and calculated conductance yet. However, the charge selectivity in MD simulation has always matched the experimental data, and seems to be generally reliable.

References

1. Yeh, I., and Gerhard Hummer. Diffusion and electrophoretic mobility of single-stranded RNA from molecular dynamics simulations. *Biophysical journal* **86.2**, 681-689 (2004).
2. Aksimentiev, A. and Klaus S. Imaging α -hemolysin with molecular dynamics: ionic conductance, osmotic permeability, and the electrostatic potential map. *Biophysical journal* **88.6**, 3745-3761 (2005).
3. Vega, C. and Abascal, J. L. Simulating water with rigid non-polarizable models: a general perspective. *Physical Chemistry Chemical Physics* **13**, 19663-19688 (2011).
4. Prajapati, J. D. *et al.* Changes in Salt Concentration Modify the Translocation of Neutral Molecules through a Δ CymA Nanopore in a Non-monotonic Manner. *ACS nano* **16.5**, 7701-7712 (2022).
5. Cao, C. *et al.* Single-molecule sensing of peptides and nucleic acids by engineered aerolysin nanopores. *Nature communications* **10.1**, 1-11 (2019).
6. Maffeo, C. *et al.* Modeling and simulation of ion channels. *Chemical reviews* **112.12**, 6250-6284 (2012).

[Reviewer1-Q5]

The interpretation of lipid diffusion in and out of the pore is similar to those proposed for Innexin-6, Pannexin-1 and CALHMs. As mentioned by the authors, the inter-subunit space of Cx36 is small and insufficient for lipid migration as far as the solved structures are considered. The dynamism of the Cx36 channels (Line 396) is a bit weak to assert the lipid diffusion. Did the author ever find any top-view 2D class averages that lose 6-fold symmetry? Or were there classes showing symmetry breaking during 3D classification even if it did not reach high-resolution?

RESPONSE

We thank the reviewer for valuable comments. We indeed observed heptameric top-views of Cx36 heptamer in 2D classification in all datasets (Authors' Response Fig. 2, right panels of each 2D classes of Cx36_{Nano}-WT, Cx36_{Nano}- Δ N8, Cx36_{LMNG}-BRIL, and Cx36_{Nano}-BRIL- Δ N16). We tried 3D reconstruction of Cx36 heptamer particles, but we could not obtain a reliable 3D density map of the Cx36 heptamer, because of a lack of side-views, even from Cx36-BRIL datasets. Although the lipid diffusion mechanism was not resolved in this study,

this data suggests that the structural dynamics between hexamer and heptamer might be involved in lipid diffusion. Inter-protomer interactions in a heptameric hemichannel might be similar to those of a hexameric hemichannel in which an inter-protomer interaction is broken and has a wide membrane-opening. We have added this data (Supplementary Fig. 1f) and the following discussion (Page 15, Lines 479-484) in the revised manuscript.

Line 479-484:

... our data yet. Interestingly, we have observed top-views of either tetradecameric GJCs or heptameric hemichannels in all the datasets collected in this study. This suggests that Cx36 GJC or hemichannel might be interchangeable between two different assemblies with six- and seven-fold symmetries (Supplementary Fig. 1f). The interchange would require a temporary disruption of the channel assemblies, which may allow the diffusion of lipids into and out of channel pores.

Authors' Response Fig. 2. 2D-Classes of hexameric- (left panels) and heptameric states (right panels) of all constructs used in this study. The schematic arrangement of TM1 to TM4 is shown as circles with the number.

[Reviewer1-Q6]

Line 321,

“PLN: 17%, FN: 0.5%”

This means that ~82% of junction channels are a mixture of PLN and FN. How were most of these particles (~82%) processed in the structure analysis?

RESPONSE

As the reviewer pointed out, the Cx36-WT GJC structure with full FN and PLN hemichannels contains substantial noises. The full FN hemichannel would be reconstructed mainly from hemichannel particles with 4, 5, and 6 FN protomers, while the full PLN hemichannel was done from the other particles. We have added the description of substantial noises included in the 3D map density in the revised manuscript, not to cause the readers' misunderstanding.

Line 360-363:

The unclear NTH densities would be primarily because substantial noises from PLN and FN protomers were included in the final 3D map. In addition, NTHs in this conformation might be still flexible due to weak interaction with TM2 and lie on the intracellular layer of the pore-occluding lipids.

Are the features of mixed PLN and FN not visible in the image processing? For example, one can expect that the volume of the pore occluding densities increases as the number of FN subunits increases.

RESPONSE

With our current dataset, we could not obtain a reliable 3D reconstruction map of any hemichannel with mixed PLN and FN protomers. Much more data with higher resolution information would be required to precisely classify hemichannel populations with mixed PLN and FN protomers. However, we thank the reviewer for the question and hypothesis. It is interesting and important because the analyses may clarify whether the channel is completely closed or not or how big is the channel opening.

In the individual subunit model (Oh et al. JGP 116, 13-32, 2000), Oh et al. point out that only one protomer is sufficient to initiate the voltage gating going toward closure. Is it possible to tell how many PLNs would cause the hemichannel to be open?

RESPONSE

Since the hemichannel model with a 5:1 ratio of PLN/FN forms a bigger pore than the full PLN hemichannel, something must block the pore during V_j-dependent closing. We think that lipids may be involved in V_j-gating, but we cannot explain how yet because it is unclear how lipids behave in GJCs with mixed FN and PLN or during the conformational changes of individual subunits.

[Reviewer1-Q7]

The authors have shown multiple structures of nanodisc reconstituted Cx36, including ΔN8, BRIL-ΔN16, but the density in the pore seen in FN seems to vary. Some of those look like the pore aggregate densities where the ball-and-chain mechanism proposed for Cx26 pH gating (Khan et al. 2020, Cell Rep. 31,107482), and in that model, those were interpreted as the N-termini occluding the pore. Are these contour level issues, or do they all have bilayer properties?

RESPONSE

All pore-occluded structures showed apparently similar bilayer properties. The aggregate densities at the cytoplasmic layer seem to be because of noise densities shown at very low contour levels. We have changed the figure using proper contour levels (Supplementary Fig. 6) in the revised manuscript. New figures clearly show bilayer properties for all presented structures.

In the protomer-focused classifications of all datasets performed during revision, we found that one or two classes showed the substantial density of a different NTH conformation which is similar to gate-covering NTH (GCN) of Cx31.3 or Cx43 (Supplementary Fig. 10). Therefore, we have also described that partially contributed the densities at the cytoplasmic layer in the revised manuscript (Page 12, Lines 352-363). We could not classify and solve the structure of Cx36 GJC/hemichannel in this conformation due to its relatively small portion.

[Reviewer1-Q8]

Line 429,

The final paragraph of the Discussion is quite speculative. The authors mention that the structure of Cx36 with Mg²⁺ did not show a significant difference. For the discussion about the contribution of Mg²⁺ to channel activity inhibition, additional positive and significant data are required. I recommend that this paragraph is deleted.

RESPONSE

As the reviewer recommended, we have deleted the paragraph describing the inhibition of channel activity by Mg²⁺ in the revised manuscript.

Minor points:

[Reviewer1-Q9]

Abstract, last line

“plasticity of electrical synapse.”

This work does not account for the synaptic plasticity directly. This expression seems misleading.

RESPONSE

We agree with the reviewer and have deleted the “directly related to plasticity of electrical synapse” (Line 31).

[Reviewer1-Q10]

Line 77,

“the channel pores were blocked by lipid bilayers”

It is likely, but not conclusive. I recommend milder expression like pore occlusion.

RESPONSE

We agree and have replaced “blocked by lipid bilayers” with “filled with two layers of lipids”.

Lines 76-77:

In the pore-occluded state, the channel pores were filled with two layers of lipids

[Reviewer1-Q11]

Line 81,

“...determine the distinct structural features”

This phrase is uncomfortable because MD in this work does not show distinct structural features but supports the functional properties of Cx36, for instance, cation selectivity and small conductance. This should be revised.

RESPONSE

We agree and have modified the sentence as follows.

Lines 81-82:

... dynamics (MD) simulations were used to investigate the structural dynamics and functional properties of Cx36 as a neuronal gap junction channel.

[Reviewer1-Q12]

Line 103,

“Notably,....”

It is misleading if this sentence means that the Cx36-BRIL function is normal or comparable to that of WT Cx36. Ref. 20 demonstrates that the deletion of the cytoplasmic loop of Cx36 significantly reduced the junctional conductance (WT: 2.54 μ S to mutant: 0.39 μ S) while the response to acidosis is shifted. When discussing open or closed, the absolute value of conductance is more important. As described above, the structures of Cx36LMNG-WT and Cx36LMNG-BRIL look different, and the conductance values in Ref. 20 seem to account for the structural differences. This should be addressed.

RESPONSE

We agree with the reviewer and have removed the sentence (Line 103 in the original manuscript) in the revised manuscript (Page 6, Line 101). As described earlier, it is unclear whether CL affects the NTH conformation. Since no map density of CL was observed in all structures, our structures cannot explain how the deletion of the N-terminal half of CL caused such a large decrease in conductance. The C-terminal halves of six CLs in the mutant channel might interact with each other or with TM helices and NTHs to inhibit ion transfer. To avoid any misleading, We have removed the sentence and revised all descriptions related to the BRIL-fused constructs considering the potential effect of the BRIL-fusion for Cx36 function throughout the revised manuscript.

[Reviewer1-Q13]

Line 118,

The term of “solvent channel” is unclear. Does this mean the pathway of water molecules forming a hydrogen bond network between inside and outside the pore?

RESPONSE

We agree that we did not clearly define the solvent channel. We have modified the sentences as follows.

Lines 111-116:

The two hemichannels docked with each other through intermolecular interaction of ECLs, and water molecules were highly concentrated at the boundary between the TM helices and the ECLs (Fig. 1a middle, red spheres). At the corresponding boundary of Cx31.3 hemichannel, a solvent tunnel was observed²³. However, similar with available Cx26 and Cx46/50 GJC structures, the solvent tunnel was closed in Cx36 GJC by the interactions of Glu49, Arg77, Arg246, and Glu249 residues (Supplementary Fig. 2a).

[Reviewer1-Q14]

Line 140,

Ref. 27 should read ref. 21.

RESPONSE

We have replaced the citation ref. 27 with ref. 21 in this sentence (Line 136).

[Reviewer1-Q15]

Line 145,

“electron density” should be revised because visible densities in cryo-EM maps are not electron density but potential.

RESPONSE

We have removed the “electron” in the sentence (Line 142).

[Reviewer1-Q16]

Line 146,

“acyl chains of LMNG” What if carried-over lipids? The head group of LMNG is large. Something should be mentioned about the position, orientation, and how it binds.

RESPONSE

While the acyl chains bound to the pore inside were clear, their head groups were not seen at all. Although cellular lipids would be mostly removed during the purification process, as the reviewer pointed out, we cannot exclude the possibility that the acyl chains are from cellular

lipids. Therefore, we revised the sentence as “acyl chains of LMNG or cellular lipids”. The position of an acyl chain bound to each protomer was shown in Fig. 1b.

[Reviewer1-Q17]

Line 147,

“This suggests that CHS and/or LMNG may stabilize the free NTH...” sounds strange. FN (free N-terminus) is named after being invisible and not stabilized. “stabilize” should be revised by something like induce or cause.

RESPONSE

We agree with the reviewer and have replaced the “stabilize” with “contribute to” (Line 144).

[Reviewer1-Q18]

Line 151,

The authors added phenylalanine to the sample grid to solve the orientation preference of the Cx36 channels. What kind of effect do the authors expect? Is there any precedent example of using phenylalanine for this problem before?

RESPONSE

Based on our amino acid sequence alignment (Supplementary Fig. 1e), we hypothesized that the hydrophobic interaction between the hydrophobic CL and air-water interface might contribute to the preferred orientation of Cx36 particles. To interfere with the hydrophobic interaction by covering the air-water interface, we tried a number of amphipathic additives. Eventually, the phenylalanine was most successful in case of Cx36. There was no precedent example of using phenylalanine before. We hypothesized that phenylalanine might cover the air-water interface or interact with hydrophobic residues in CL and CTL and block the direct interaction between air-water interface and Cx36. This approach did not work for the Cx43 GJC sample, but we believe that it is worth of trying for other cryo-EM samples with a problem of preferred orientation.

[Reviewer1-Q19]

Line 160,

Why are soybean polar lipids used for making nanodiscs while Cx36 is derived from human? POPC was used for the MD simulation and is there a possibility that the different lipid components could lead to alternative conclusions?

RESPONSE

We chose soybean lipids because they were relatively easy to handle and have been used for the structural studies of Cx26 hemichannel¹ and pannexin-1² by others and Cx43 GJC by us. Lipid components, especially cholesterols, could shift the conformational equilibrium to FN, since we observed the binding CHS to the pore interior of Cx36 GJC. Therefore, we have collected and analyzed the dataset of Cx36-BRIL GJC in brain lipid extract. Although the

analysis was not completed yet, we have found a similar conformational equilibrium with a PLN/FN ratio of 6:4. The equilibrium shifted from 9:1 (in soybean lipids) to 6:4 (in brain lipids), but it is unclear whether the change is caused by cholesterol. More studies are needed to clarify whether cholesterol plays an important role on the channel activity.

However, lipid components might not be very important in the activity of Cx36 GJC in cells. Cx36 maintains its channel function in various cell types such as *Xenopus laevis* oocytes (Ref. 20), N2A (murine neuroblastoma 2A, Ref. 20), RIN (rat β -cell insulinoma Ref. 25), HeLa cells (Ref. 25), suggesting that this channel may not be very sensitive to different lipid components.

MD simulation was performed only for the full PLN conformation during a very short time (1.2 μ sec). Since no lipid was inside the pore initially and diffused into the pore during the simulation, the result would not change depending on the lipid components used in the simulation.

References

1. Khan, A. K. *et al.* Cryo-EM structure of an open conformation of a gap junction hemichannel in lipid bilayer nanodiscs. *Structure* **29**, 1040-1047 (2021).
2. Michalski, K. *et al.* The Cryo-EM structure of pannexin 1 reveals unique motifs for ion selection and inhibition. *Elife* **9**, e54670 (2020).

[Reviewer1-Q20]

Line 173,

“almost identical” is unclear. Is this regarding the built models? The maps look different as one has BRIL.

RESPONSE

We agree and have replaced “structure” with “models including TM helices and ECLs” and revised the sentence as follows.

Lines 161-163:

the protein models including TM helices and ECLs was almost identical to those in the LMNG/CHS environment (Fig. 1a and Fig. 2a).

[Reviewer1-Q21]

Line 174,

“it showed no clear map densities of CHS...”

Does this mean CHS was pushed out during nanodisc reconstitution? Could it be that the binding of CHS in the pore observed in the structure in detergents might be artificial? Because the authors mention the contribution of CHS before, this sounds not consistent. I

am wondering what occupies this space of Cx36 in nanodiscs. In line 176, “data not shown” should be avoided.

RESPONSE

We thought that CHSs were likely excluded from the GJC-nanodisc sample during nanodisc reconstitution. The final sample in detergents always contained a substantial amount of CHS because CHSs (~10% of total detergents) were included in the buffers for all purification steps. In contrast, the Cx36-nanodisc sample would not have much CHS because free CHSs were likely removed by using Bio-beads as an essential step of reconstitution. In this case, phospholipids would interact with the pore interior where CHS molecules are bound in the structure of LMNG/CHS. However, there is another high possibility that the map densities were not clearly shown simply because of low resolution.

The expression “data not shown” has been removed in the revised manuscript. We have modified the sentence as follows

Lines 163-166:

However, unlike the structure in detergents, it showed no clear map densities of CHS molecules in the interior of the pore (Supplementary Fig. 6a) probably because the resolution was much lower and/or CHSs were mostly removed during nanodisc reconstitution.

[Reviewer1-Q22]

Line 221,

“due to the insertion of Ala14,.....”

This reads that only Ala14 causes the N-terminal portion specific to Cx36, but in fact, other factors like Glu17 and lipid acyl chains may contribute to that conformation. This can be deleted.

RESPONSE

As the reviewer suggested, we have deleted the phrase (Line 216).

[Reviewer1-Q23]

Line 259,

“...N state may also be obstructed by lipids”

Is there a possibility that anything other than lipids may comprise the pore density? For example, invisible C-terminus or CL?

RESPONSE

We strongly believe that the pore-occluding densities are phospholipids because of the double-layer properties of the densities, which were shared by the structures of Cx36_{Nano}-WT, Cx36_{Nano}-ΔN8, and Cx36_{Nano}-BRIL-ΔN16 GJCs. The possibility of pore-occlusion by NTH and CL can be excluded based on the structure of Cx36_{Nano}-BRIL-ΔN16, which does not

contain both NTH and CL. CTs would be structurally difficult to block the core, considering their short lengths of ~46 residues (Author's Response Fig. 3). Nevertheless, since we could not clearly identify the lipid heads and tails, we cannot completely exclude the possibility that CL or CT may partially contribute to the densities.

Authors' Response Fig. 3. Possible paths for pore-occlusion by CT of Cx36_{Nano}-BRIL-ΔN16. The map densities of the transmembrane domain, C-terminus (C-term) and BRIL were colored in green, red and white. Because of the relatively low resolution (3.8 Å) of density of BRIL in Cx36_{Nano}-BRIL-ΔN16, each BRIL density was not clearly shown. Therefore, we superimposed densities of Cx36_{LMNG}-BRIL (2.2 Å) and Cx36_{Nano}-BRIL-ΔN16. We displayed the transmembrane domain of Cx36_{Nano}-BRIL-ΔN16 and the BRIL domain of Cx36_{LMNG}-BRIL. There are two expected paths for CT of Cx36_{Nano}-BRIL-ΔN16 (path1; yellow line, and path2; orange line), but no density was observed around the paths in all BRIL-fused constructs (Cx36_{LMNG}-BRIL and Cx36_{Nano}-BRIL-ΔN16).

[Reviewer1-Q24]

Line 261,

“Cx36_{Nano}-ΔN8”

If my understanding is correct, this construct as well as Cx36_{Nano}-WT does not have BRIL. The authors described that BRIL has changed the behavior in terms of the orientation in the hole of grids (Line 107). Is the effect of BRIL not essential for the structure determination or solving the particle orientation problem? Does it depend on nanodisc or detergent? The modification in CL is expected to affect the functionality of the Cx36 channels seriously, and if BRIL is not needed for structure determination, that would be better.

RESPONSE

With Cx36-WT GJC samples, it is extremely difficult to get side views sufficient for further processing. The addition of phenylalanine increased the number of side views, but they were not always sufficient to achieve high resolution. A high resolution above 2.5 Å would be necessary to understand the lipid-protein interaction and precise conformations of flexible NTH, but we never reached such a high resolution with Cx36-WT, regardless of detergents or lipids (3.1 Å was the maximum resolution so far). Since we understand the reviewer's concern about BRIL fusion that can remove important functions of Cx36 GJC, we always try to determine the structures of both Cx36-BRIL and WT at high and low resolutions, respectively.

[Reviewer1-Q25]

Line 289,

In MD simulations, can the movement of the lipids be traced? What if the FN structure is used as an initial model for MD?

RESPONSE

We have investigated the movie of the MD simulation of the ion transfer but have not found lipid molecules moving into the channel in a short time of 1.2 μsec. We have not performed MD simulation of the full FN structure, but we are quite sure that such a short simulation is not enough to make a large conformational change of TM helices necessary to form a membrane opening or a large gap between protomers. In addition, we have done similar simulations for Cx43 GJCs, but have never seen the conformational change of NTH between PLN and gate-covering NTH (GCN) during simulations. Thus, we do not believe any large conformational changes predicted by MD simulations.

[Reviewer1-Q26]

Line 344,

“π-helix is also found in all solved”

Because this may not apply to all connexins in the future, please revise it to the moderate expression.

RESPONSE

We agree with the reviewer and have replaced “all solved connexin structures” with “currently the available connexin structures such as Cx26, Cx31.3, Cx43, and Cx46/50” as follows.

Lines 392-394:

This π-helix is also found in the currently available connexin structures such as Cx26, Cx31.3, Cx43, and Cx46/50 (Supplementary Fig. 9a-f, cyan)^{4,22,23,26}

[Reviewer1-Q27]

Line 387,

Do the N-termini of Cx36 have a voltage sensor? It is interesting if Cx36 can exhibit the gating depending on the transjunctional voltage mediated by the N-terminus. It has been

postulated that the N-terminus, including the voltage sensor, should be located near the entrance of the pore to feel the electric field to respond to the transjunctional voltage (Verselis et al. Nature, 1994). If the N-terminal portion of FN is totally out of the pore pathway, it would be impossible for the N-terminus to sense the voltage field. The description of the relationship between the voltage gating property of Cx36 and the structure is preferable.

RESPONSE

We appreciate the reviewer's thought-provoking point. We agree with the reviewer that it would be better to describe the relationship between the voltage gating property of Cx36 and the structure. As the reviewer depicted, Verselis et al. postulated that the N-terminus, including the voltage sensor, should be located near the entrance of the pore to feel the electric field and Rimkute, Lina, et al. showed that the negatively charged residues (Glu8 and Glu12) of N-terminus functions as sensor for intercellular pH and concentration of Mg^{2+} (Rimkute, Lina, et al. (2018)).

It is unclear where NTH is mainly located in the FN conformation. Since NTH is amphipathic, it is likely on the intracellular layer of the pore occluding lipids. In addition, we also observed gate-covering NTH (GCN) conformation in the protomer-focused analyses during revision. As the reviewer indicated, the N-terminus should be located near the entrance of the pore to feel the electric field. Therefore, in response to the applied voltage on the channel, the NTH conformation may change between PLN and GCN or FN on the intracellular layer of the pore occluding lipids. We have added this discussion on the voltage gating property of Cx36 in the revised manuscript as followings.

Line 489-500:

Forth, it is unclear whether the conformational change of Cx36 GJC discovered in this study is involved in its voltage-gating mechanism. Since the charged residues in NTHs function as voltage sensors^{44,48-50}, the transjunctional voltage likely induces the conformational transition of NTHs. Because two hemichannel regions in a GJC face each other, their responses to the applied transjunctional voltage would be opposite to each other. In the case of Cx36 GJC, the full PLN conformation might be formed in one hemichannel region and the full FN in the other, and vice versa, resulting in the channel closing in both cases. Amphipathic NTHs in the FN state might not be totally out of the pore pathway but lie on the intracellular layer of the pore-occluding lipids so that they can sense the voltage field. To prove this hypothesis, further structural and electrophysiological studies are needed to find molecules or conditions that greatly shift the conformational equilibrium of the channel to the FN and PLN states, respectively, and investigate their effects on the ionic current through the channel.

[Reviewer1-Q28]

Line 503,

The information of ref. 26 should be written appropriately.

RESPONSE

It is a preprint paper posted in Research Square. We have edited the reference according to the guide to formatting articles.

Lines 842-843:

Woo, J. S. *et al.* Structural insights into the gating mechanism of human Cx43/GJA1 gap junction channel. Preprint at <http://doi.org/10.21203/rs.3.rs-730129/v1> (2021)

[Reviewer1-Q29]

Fig.3 legend,

The sub-panel labels should be written in the legend like “Cx36 (a) and Cx50 (b), and superposition of Cx36 and Cx50 (c)”.

RESPONSE

We have added the sub-panel labels as followings.

a-e Detailed structures of NTHs and the NTH-TM1 loops in Cx36 (a), Cx43 (b), Cx46 (c), and Cx50 (d), and their structural alignment (e). **f-i** Ribbon representation of two facing protomers in Cx36 (f), Cx43 (g), Cx46 (h) and Cx50 (i) hemichannel regions.

Reviewer #2

Review Comments:

The authors present a well-written and comprehensive analysis of the previously unexplored molecular architecture of human Cx36 gap junction intercellular channels via the clever application of Cryo-EM techniques. They explore, compare, and contrast Cx36 structures with previously solved gap junction channel structures such as Cx50 and Cx26 and find features that are previously undescribed in Cx channels such as the occlusion of the pore by lipids in what they refer to as the FN state. This is significant as Cx36 is the main neuronal gap junction protein and the authors' findings illuminate new questions about the regulation of electrical synapse conductance and the potential for new studies of Cx36 structure/function.

While the study is well written and comprehensive, several minor revisions are suggested to increase clarity of the manuscript.

RESPONSE

We greatly appreciate the reviewer for reviewing our manuscript and providing valuable suggestions, which helped us in improving the quality of the manuscript. We are grateful for the description “the authors’ findings illuminate new questions about the regulation of electrical synapse conductance and the potential for new studies of Cx36 structure/function”, on which we totally agreed. To address the minor concerns of the reviewer, we have thoroughly revised the manuscript accordingly.

Minor revisions:

[Reviewer2-Q1]

Overall, the paper suffers from an overuse of initialisms that are not always explained in the text. It would be beneficial, especially in the figure legends, to clearly state all initialism definitions at the end of the figure legend. For example, in Figure 1 it would be helpful to define in the figure legend: CHS, ECL, NTH, and EC.

RESPONSE

We appreciate the reviewer's considerate point. We have added the definition of each initial in the legend of Figure 1. Also, we have thoroughly checked the overuse of initialisms and corrected them in the revised manuscript.

Main text:

[Reviewer2-Q2]

Line 43: The 21 connexin genes referenced need to be clarified as 21 human connexin genes.

RESPONSE

We have added the “human” in the sentence in the revised manuscript.

Line 45:

Twenty-one human connexin genes, ...

[Reviewer2-Q3]

Lines 69-71: Innexin and pannexins are discussed without referencing their relationship to Connexins. It would be helpful for the reader to explain these relationships.

RESPONSE

We have added the following sentences explaining the relationship of connexins, innexins, and pannexins in the first paragraph of the introduction as following.

Line 41-44:

Vertebrate GJCs are formed by connexins, while invertebrate GJCs consist of innexins with no sequence homology to connexins. Pannexins are innexin homologs in vertebrates and function as hemichannels connecting cytoplasmic and extracellular space³.

[Reviewer2-Q4]

Line 90: In Extended Data Figure 1a, several forms of Cx36 are compared, yet we have not yet had an introduction to them in the text. This figure was then quite confusing to parse upon first view. We suggest moving 1e to become 1a and offering a short introduction to these forms before moving on to the current 1a. In addition, the SDS-PAGE gel is completely unexplained in the figure legend. The relevance of the bands and the identity of MSP1E1 is unknown until much later in the text. Explanation is needed.

RESPONSE

As the reviewer recommended, we have rearranged the Supplementary Fig. 1, moving 1e to become 1a with a short introduction at the figure legend. We have also added the figure legend for the SDS-PAGE gel in the revised manuscript (Supplementary items Lines 15-17).

[Reviewer2-Q5]

Line 95: The figure legend for Extended Data Figure 1d does not state what is highlighted in red. We assume these to be hydrophobic residues. Please correct the figure legend to explain the red color designation.

RESPONSE

We thank the reviewer for kindly finding errors. We have added the sentence “The hydrophobic residues are shaded in red.” to the figure legend in the revised manuscript (Supplementary items Line 24).

[Reviewer2-Q6]

Line 172: The acronym CTL is unexplained.

RESPONSE

It was a typo for CT (C-terminal loop). We have replaced the “CTL” with “CT” in the revised manuscript (Line 161).

[Reviewer2-Q7]

Line 214: Figures are referred to out of order.

RESPONSE

The supplementary figures were referred to out of order. We corrected them in the revised manuscript (Line 204).

[Reviewer2-Q8]

Line 290: The acronym POPC is unexplained.

RESPONSE

We added the full name “1-palmitoyl-2-oleoylphosphatidylcholine” to the sentence in the revised manuscript (Lines 292-293)

[Reviewer2-Q9]

Line 615: The description of cells “harvested at 500 g for 20 minutes” is an odd phrasing and confusing. Please reword. Same for line 656.

RESPONSE

We have replaced “harvested” with “centrifuged.”

Line 529:

the cells were centrifugated at 500 g for 20 min

Line 567:

the cells were centrifugated at 500 g for 10 min

[Reviewer2-Q10]

Line 975: We believe “meshes” is incorrectly represented as “mashes” here.

RESPONSE

We have replaced the “meshes” with “meshes” (Supplementary items Line 37)

[Reviewer2-Q11]

Line 995: The blue colors are difficult to distinguish/understand. We suggest diversifying the colors here.

RESPONSE

As the reviewer suggested, we have shaded amino acids with diversified colors; yellow (80%), green (90%), or cyan (100%), according to sequence conservation. We also edited the figure legend accordingly. Here, residues 1-96 are shown as a representative (see below).

Supplementary items Lines 54-62:

Supplementary Fig. 4: Amino acid sequence alignment of human connexins. All human connexins except highly diversified Cx23 were included in the sequence alignment. The conserved residues are highlighted by yellow (80%), green (90%), or cyan (100%) colors. Ala14 is shaded by red. Trp4, Ala14, Gln17, residues 30-33 (α - or π -helix), residues 39-42 (π -helix), Asp47, K238, and E239 in Cx36 are indicated by red boxes and labeled. Schematic drawings of the secondary structures in Cx36 are shown above the amino acid sequence. Coils, α -helices, and β -sheets are represented as solid lines, cylinders, and arrows, respectively.

[Reviewer2-Q12]

Extended Data Fig 10 is not referenced in the manuscript.

RESPONSE

We have added the citation of Supplementary Fig. 11 (Extended Data Fig. 10 in the previous manuscript) to line 631.

Reviewer #3

Review for manuscript Cryo-EM structures of human Cx36/GJD2 neuronal gap junction channel

Summary of findings:

The authors of this work aimed to derive the structure of the human gap junction channel (GJC) formed by the human Cx36. To do so, they obtained eight structures of human Cx36 GJC in its pore-occluded and open states using single-particle cryo-electron microscopy (cryo-EM). In the pore-occluded state, the channel pores were blocked by lipid bilayers and the NTHs of Cx36 were excluded from the pore. In comparison, the channel pore was completely open without any obstruction in the pore-lining NTH (PLN) state, suggesting that the binding of NTHs to the channel pore is an essential step for channel opening. By analyzing the set of determined structures, the authors of this work proposed that a dynamic (thermodynamic) equilibrium between the closed and open conformation is present for the case of the Cx39 GJC. Moreover, in the open state of the GJC, the lumen pore is narrower and more acidic than those of other connexins, suggesting an explanation for its low conductance and strong cation selectivity. On top of that, the authors describe a conformational change of the NTH in the open state of the GJC that is accompanied by an alpha-to-pi-helix transition of TM1. As a whole, the work proposed by Lee et al provides high resolution information on the conformational flexibility of the Cx36 GJC, suggesting a novel and potential role of lipids in the regulation of the channel activity that could be consistent with similar mechanism reported for both innexins and pannexins channels as well.

RESPONSE

We greatly appreciate the valuable time that the reviewer spent reviewing our manuscript and providing valuable suggestions, which helped us in improving the quality of the manuscript. To address the reviewer's concerns, we have thoroughly revised the manuscript. We feel that our revised manuscript has been greatly improved thanks to the reviewer's considerate and thought-provoking comments.

Manuscript assessment:

In general terms, is a very well written manuscript, using a consistent and sounding language but relying on unorthodox definitions (see below) that may be confusing for the connexin-aware academic community. This is the case of using the GJIC acronyms (i.e. gap junction intercellular channel) instead of the usual GJC (i.e. gap junction channel) and the term heterotypic channel to define a GJC whose composing hemichannels appear in different structural conformation. The experimental design is straightforward and similar to that of previous works describing the structure of innexins and pannexins channels (see manuscript) solved by single-particle cryo-electron microscopy using lipid nano-discs. Following this approach, the authors of this work produced WT and several mutant Cx39 GJC (see extended

data figure 1) which were structurally analyzed. Among other interesting proposals arising from the authors work, they found that the closed structure of the channel present lipids in the lumen pore (referred as FN). This occurs simultaneously with an NTH which is absent. Therefore, the authors propose a novel regulatory role for lipids in the case of Cx39 GJC. The authors complemented their structural analyses by molecular dynamics simulation of the WT GJC to determine the conductance of the channel under a transjunctional potential difference. In doing so, they propose that the low conductance of this channel is due to both a narrower pore and an acidic charged patch that may in turn hinder ion transport. Of note, despite the relevance for their results, the authors didn't discuss that the pore lumen of the FN state, the one classified as closed, is larger than that of the closed state of the channel. Probably the most appealing but also controversial finding of this work is the existence of a dynamic equilibrium between the closed and open conformation of the Cx39 GJC. If this proposal is to remain valid, it will certainly reconfigure the biophysical landscape of the GJC world.

RESPONSE

We are grateful for the description “If this proposal is to remain valid, it will certainly reconfigure the biophysical landscape of the GJC world”, which we totally agree on. As the reviewer indicated, we removed unorthodox definitions including GJCh and heterotypic channel in the revised manuscript. Detailed responses are included as followings.

After careful evaluation of this manuscript, a series of issues arise that should be thoroughly accessed before publication (see annotated manuscript):

[Reviewer3-Q1]

Line 21 and 38: The authors should consider using a more standard nomenclature denoting this structure as a Gap Junction Channel (GJC). As the GJC is in fact, an intercellular molecular structure, the nomenclature suggested by the authors seems redundant.

RESPONSE

As the reviewer suggested, we have replaced “GJCh” with “GJC” throughout the manuscript.

[Reviewer3-Q2]

Line 28: Which is the actual characteristic making this loop unique?

RESPONSE

The NTH-TM1 loop of Cx36 is longer than those of other connexins, and thus a large hydrophobic pocket is formed between neighboring loops. We could not describe this due to the limited number of words in the Abstract. We have removed the description on the NTH-TM1 loop and added another important observation in the revised manuscript.

Lines 28-29:

... the α -to- π -helix transition of the first transmembrane helix, which weakens the protomer-protomer interaction.

[Reviewer3-Q3]

Line 44: consider replacing homology with identity

RESPONSE

We have replaced the “homology” with “identity” as suggested by the reviewer for an accurate description. (Line 46)

[Reviewer3-Q4]

Line 65: As GJCs perform a series of fundamental biological functions, such as electrical synapse, a clear pharmaceutical strategy that can be followed to use GJC blockers as drugs is currently missing.

RESPONSE

We appreciate the reviewer’s comment and have added the sentences for the explanation of pharmaceutical approaches of blocking GJC.

Lines 63-65:

... treating these pathological situations¹⁷. For example, in ALS, the secondary neuronal death is extended by neuronal GJC, and the progressive neuronal death can be mitigated by blocking Cx36¹⁷.

[Reviewer3-Q5]

Lines 70 – 76: The authors should consider including a clear connection, either structural or phylogenetical, between innexins, pannexins and connexins to better support this paragraph.

RESPONSE

We added the following sentences explaining the relationship of connexins, innexins, and pannexins in the first paragraph of the introduction as following.

Lines 41-44:

Vertebrate GJCs are formed by connexins, while invertebrate GJCs consist of innexins with no sequence homology to connexins. Pannexins are innexin homologs in vertebrates and function as hemichannels connecting cytoplasmic and extracellular space³.

[Reviewer3-Q6]

Line 81: consider adding the electrostatic characteristic to the NTH binding

RESPONSE

We have added the “by the hydrophobic interaction” to the sentence as follows.

Lines 79-80:

... suggesting that the binding of NTHs to TM1 and TM2 of channel pore by the hydrophobic interaction is an essential step for channel opening

[Reviewer3-Q7]

Line 82: consider adding TM1, or the corresponding TMHs to better define the binding region

RESPONSE

We have added the “TM1 and TM2 of” to the sentence as follows.

Lines 79-80:

... suggesting that the binding of NTHs to TM1 and TM2 of channel pore by the hydrophobic interaction is an essential step for channel opening

[Reviewer3-Q8]

Line 117: consider replacing homology with identity

RESPONSE

We have replaced the “homology” with “identity” as suggested by the reviewer for an accurate description. (Line 110)

[Reviewer3-Q9]

Line 147: consider adding the electrostatic characteristic to the NTH binding

RESPONSE

This is related to the comment of “reviewer3-Q6”. We have modified Line 80 in the revised manuscript, and we could not find more descriptions regarding the NTH binding to the channel pore.

[Reviewer3-Q10]

Line 237: Despite the authors use the term heterotypic to refer to a GJC where its constituents hemichannels are in different conformations, this usage could be confusing as in the field of connexins, heterotypic usually refers to channels formed by protomers belonging to different connexins. This is the case of the Cx46/50 GJC structure.

RESPONSE

We agree with the reviewer and have replaced the term “heterotypic” with “structurally hetero-junctional”. We have corrected this terminology throughout the revised manuscript accordingly.

[Reviewer3-Q11]

Line 251 – 252: It is well known in the field that for the case of GJC, maximum conductance arises when the transjunctional voltage difference tends to zero. This evidence seems to be contradictory to the authors statement and it should be considered for further discussion.

RESPONSE

The reviewer pointed out that the structures should show the maximum conductance state in our experimental condition with no transjunctional voltage, and the conformationally dynamic state would not be the state with the maximum conductance. There are two possibilities that we have hypothesized and are trying to prove. 1) The conformationally dynamic state may have the maximum conductance. For example, a hemichannel region with four PLN protomers and two consecutive FN protomers without lipids or with highly mobile lipids has a pore substantially bigger than the full PLN conformation. Therefore, the dynamic conformational change of individual protomers, especially the frequent change in one or two NTHs from the full PLN conformation, could result in an ion transfer rate higher than that in the full PLN conformation. 2) The full PLN state may be the maximum conductance state and require other factors that are abundant in cells but not included in our experimental system. A specific membrane lipid or amphipathic molecule may strongly bind to the pockets between NTHs in the full PLN conformation and stabilize the conformation. We have added this discussion in the revised manuscript as followings.

Lines 501-513:

GJCs generally show the maximum conductance when the transjunctional voltage difference tends to zero, and our experimental conditions for the structural study have no transjunctional voltage. Therefore, although Cx36 GJC in the full PLN conformation showed substantial ionic current and cation selectivity in our MD simulation, the conformationally dynamic state of Cx36 GJC might form a larger pore and show the maximum conductance. For example, a hemichannel region with four PLN protomers and two consecutive FN protomers without lipids or with highly mobile lipids has a pore substantially bigger than that with only PLN protomers. Therefore, the frequent conformational change of one or two NTHs in the full PLN conformation could result in an ion transfer rate higher than no conformational change. Alternatively, the full PLN state might be the maximum conductance state, and its maintenance might require other factors that are abundant in cells but not included in our experimental system. A specific membrane lipid or amphipathic molecule might strongly bind to the pocket between neighboring NTHs in the PLN conformation and stabilize the conformation.

[Reviewer3-Q12]

Line 307: Do the authors refer to C-alpha RMSD? If this the case, as the C-alpha RSMD obtained is close to the crystal structure resolution, the maintenance of the overall structure can be ascertained.

RESPONSE

Yes, it was C-alpha RMSD. We have added “C α ” in front of RMSD in the revised manuscript.

[Reviewer3-Q13]

Line 382: In strict sense, Connexin based-channels are passive transporters. Therefore, they shouldn't be considered as “facilitators”. The authors should consider reconfiguring this sentence to avoid misleading

RESPONSE

We have replaced the “facilitates” with “mediates” as following.

Line 422:

Cx36 GJC mediates ionic transmission through vertebrate electrical synapses

[Reviewer3-Q14]

Lines 386 – 389: A proper set of references is needed here.

RESPONSE

We have added the reference (underlined) to the sentence.

Lines 427-429:

... and closed GJCs or the probability of each GJC being open⁴⁴. Known regulatory mechanisms include inhibition by Mg²⁺ ...

Reference

1. Bargiello, T. A. *et al.* Gating of Connexin Channels by transjunctional-voltage: Conformations and models of open and closed states. *Biochim. Biophys. Acta Biomembr.* **1860**, 22-39 (2018).

[Reviewer3-Q15]

Line 391 – 392: Despite that in ion channels there is plenty of evidence supporting the notion that a certain thermodynamic equilibrium interconnect different states of the channel, to ensure that this phenomenon is also valid for connexin-based channels, in particularly gap junction channels, a larger body of evidence is required. The authors should consider further electrophysiological experiments such as dual-cell voltage clamp protocols.

RESPONSE

We are grateful for the reviewer's comments about the thermodynamic equilibrium issue for gap junction channels. We feel that the reviewer largely supports our hypothesis and totally agree with the reviewer that a larger body of evidence is required to fully validate the model. However, we do not have a good strategy to prove the hypothesis by functional experiments, because it is difficult to define the conformational states of NTHs in electrophysiological experiments, which is necessary to link the functional consequences of specific NTH

conformation. No mutation on NTH could further stabilize the PLN conformation. Mutations on the transmembrane helices may inhibit the conformational dynamics, which would be crucial to expel lipids out of the channel at the initial assembly of the hemichannel (We have preliminary data for another connexin member showing that a mutation in TM1 designed for stabilizing the PLN conformation simply killed the activity). The only possible way is to find an activator molecule, confirm its effect on the GJC structure, and use it in electrophysiology experiments. For these reasons, we remained this question for our following studies and tried to tone down our hypothesis in the revised manuscript.

Reviewer3-Q16

Lines 394 – 395: How the authors can be sure that the lipids blocking the pore lumen are actually a biologically relevant function and not an artefact of the applied protocol. This is particularly appealing when considering that previous works in connexins didn't find these lipids. On the other hand, if these findings are to remain valid, the authors should discuss the role of transjunctional voltage difference regulating the presence of lipids in the channel pore.

RESPONSE

We appreciate the reviewer's thought-provoking point. As the reviewer indicated, no GJC structure with the lipids blocking the pore lumen has been reported so far. However, they were shown for an innexin, a functional homolog of connexin, which may share a common gating mechanism. All structures of Cx26 GJC were solved in detergents, which have big head groups and thus would be difficult to form a bilayer occluding the pore lumen. In addition, their resolutions were not sufficiently high to see the bound lipids/detergents in the pore lumen. The Cx26 hemichannel structures in lipid environments were also solved at low resolutions, and thus it was impossible to see clear lipid densities. The only high resolution structure in a lipid environment is available for heteromeric Cx46/50 GJC, which showed only the full PLN conformation. We determined the structures of Cx43 in POPE lipids, which showed a similar structural equilibrium as well as pore-occluding lipids in the full gate-covering NTH (GCN) conformation (unpublished data). Therefore, pore-occluding lipids have been shown in two different GJCs (Cx36 and Cx43), not shown in Cx46/50 GJC, and not clear in Cx26 GJC. NTHs of Cx46/50 seem to have stronger affinities for the transmembrane domains and thus favor the PLN conformation.

Regarding the possibility of an artifact caused by the applied method, we cannot completely exclude the possibility that the pore-occluding lipids were inserted during nanodisc reconstitution and maintained till cryo-EM imaging. However, there are several reasons why we strongly believe that the pore-occluding lipids are not an artifact.

1) Since the reconstituted GJC-nanodisc sample was further purified by gel filtration, no excess lipid should have been included in the final sample for cryo-EM. Although GJCs were exposed to excess lipids and detergents at the beginning of the reconstitution, most detergents were adsorbed by Bio-beads and excess lipids would mostly form liposomes, which were

removed by gel filtration. More than 12 hr incubation would be enough for the equilibrium of all molecular interactions to be established. The physiological intracellular environment also contains various hydrophobic/amphipathic molecules and lipid vesicles. Therefore, GJCs in the cell membrane are likely exposed to more environmental lipids than those in our simple nanodisc system.

2) Before the channel assembly, connexins should be in the FN or GCN conformation rather than the PLN conformation because NTH is amphipathic. When they gather to assemble into a hemichannel, membrane lipids are likely caged inside the pore. We believe it is reasonable to think that at least some lipids should be expelled from the pore after the initial assembly. It is difficult for us to imagine that, with no energy input, all lipids are completely removed before the assembly is finished. The structures show that the channel assembly with pore-occluding lipids is well maintained and thus energetically quite favorable (see Authors' Response Fig. 4).

Since we have no physiological evidence on the lipid-mediated gating, we have also added the sentence "we cannot completely exclude the possibility that the pore-occluding lipids in the FN conformation were artificially introduced during the cryo-EM sample preparation" in the revised manuscript (Lines 449-450).

The reviewer's comment on the role of transjunctional voltage difference is related to the point of Reviewer #1 Q27. We believe that the NTH, which has been postulated as the voltage sensor (Verselis et al. *Nature*, 1994), would play a key role to feel the electric field from the transjunctional voltage. However, without strong evidence, we would like to be very careful to mention our idea on how the NTH conformation will change in response to the transjunctional voltage. Hence, we have added a short discussion in the revised manuscript as follows.

Lines 489-500:

Forth, it is unclear whether the conformational change of Cx36 GJC discovered in this study is involved in its voltage-gating mechanism. Since the charged residues in NTHs function as voltage sensors^{44,48-50}, the transjunctional voltage likely induces the conformational transition of NTHs. Because two hemichannel regions in a GJC face each other, their responses to the applied transjunctional voltage would be opposite to each other. In case of Cx36 GJC, the full PLN conformation might be formed in one hemichannel region and the full FN in the other, and vice versa, resulting in the channel closing in both cases. Amphipathic NTHs in the FN state might not be totally out of the pore pathway but lie on the intracellular layer of the pore-occluding lipids so that they can sense the voltage field. To prove this hypothesis, further structural and electrophysiological studies are needed to find molecules or conditions that greatly shift the conformational equilibrium of the channel to the FN and PLN states, respectively, and investigate their effects on the ionic current through the channel.

Authors' Response Fig. 4. Two possible scenarios of the channel opening after the initial assembly. Top views are shown in three boxes. Small black circles in the boxes indicate lipids. Green and yellow circles indicate connexins in GCN and PLN conformations, respectively.

Reference

1. Verselis, V. K., Christopher, S. G., and Thaddeus, A. B. Opposite voltage gating polarities of two closely related onnexins. *Nature* **368**, 348-351 (1994).
2. Kraujalis, T. *et al.* The amino terminal domain and modulation of connexin36 gap junction channels by intracellular magnesium ions. *Frontiers in physiology* **247**, (2022).
3. Rimkute, L. *et al.* Modulation of connexin-36 gap junction channels by intracellular pH and magnesium ions. *Frontiers in physiology* **9**, 362 (2018).
4. Bargiello, T. A. *et al.* Gating of Connexin Channels by transjunctional-voltage: Conformations and models of open and closed states. *Biochimica et Biophysica Acta (BBA)-Biomembranes* **1860**, 22-39 (2018).

[Reviewer3-Q17]

Line 401 – 402: Despite a very appealing and highly interesting suggestion, again, a larger body of evidence is required to establish an equilibrium between the open and closed state of the channel. This is particularly relevant when the available evidence suggest that the lumen pore is highly hydrophilic. In such case, a strong free energy gradient should be compensated so to be able to change the conformation of the NTH to allow the insertion of membrane lipids. Up to the same extent, the authors should discuss the accessing pathway that membrane lipids should follow to block the lumen pore. Should they arrive throughout the transmembrane helices or following another pathway?

RESPONSE

This comment is related to the point of the reviewer #1 Q5. We agree with the reviewer that a large body of evidence is required to establish the equilibrium between the open and closed states of the channel. Although we could not fully demonstrate the diffusion pathway of lipids, we have three interesting observations, which are relevant to the lipid diffusion through the transmembrane helices.

1) We observed heptameric top-views of Cx36 heptamer in 2D classification in all datasets (Authors' Response Fig. 2, right panels of each 2D classes of Cx36_{Nano}-WT, Cx36_{Nano}- Δ N8, Cx36_{LMNG}-BRIL, and Cx36_{Nano}-BRIL- Δ N16). We tried 3D reconstruction of Cx36 heptamer particles, but we could not obtain a reliable 3D density map of the Cx36 heptamer, because of a lack of side-views, even from Cx36-BRIL datasets. Although the lipid diffusion mechanism was not resolved in this study, this data suggests that the structural dynamics between hexamer and heptamer might be involved in lipid diffusion. The inter-protomer interactions in a heptameric hemichannel might be similar to those of a hexameric hemichannel in which an inter-protomer interaction is broken and has a wide membrane-opening. Although we cannot exclude the possibility that the heptameric assembly is an artifact in the in vitro system, it can at least tell us that the channel assembly is not strong but easily breakable. We have added this data (Supplementary Fig. 1f) and the following discussion (Page 15, Lines 479-484) in the revised manuscript.

Lines 479-484:

... our data yet. Interestingly, we have observed top-views of either tetradecameric GJCs or heptameric hemichannels in all the datasets collected in this study. This suggests that Cx36 GJC or hemichannel might be interchangeable between two different assemblies with six- and seven-fold symmetries (Supplementary Fig. 1f). The interchange would require a temporary disruption of the channel assemblies, which may allow the diffusion of lipids into and out of channel pores.

Authors' Response Fig. 2. 2D-Class of hexameric- (left panels) and heptameric state (right panels) of all constructs used in this study. Schematic arrangement of TM1 to TM4 is shown as circles with number.

2) We observed that the hydrophobic intermolecular interactions in the cytoplasmic halves of TMDs are mostly broken by the α -to- π -helix transition of TM1 (Supplementary Fig. 9a-d). This was also observed in Cx43 GJC. The conformational transition from FN to PLN may induce partial dissociation of neighboring protomers to expel lipids out of the pore.

3) We observed map densities almost connecting the pore inside and outside at the intermolecular interface in the FN state (Authors' Response Fig. 5). We also observed quite clear map densities of acyl chains near the intermolecular interface in the PLN conformation. Therefore, lipids may be able to go through the transmembrane region even when the intermolecular interaction is very shortly broken.

Authors' Response Fig. 5. Heterogeneous densities at the boundary of Cx36 and lipid membrane. a-f cryo-EM reconstruction map and ribbon representation of Cx36_{Nano}-WT in FN- (left panels) and PLN state (right panels). The density and atomic model of Cx36 in FN-, and PLN state, NTH, acyl chains in hydrophobic pockets and acyl chains outside the channel are colored green, yellow, magenta, cyan, and gray. Gray densities between protomers almost connect the pore inside and outside of Cx36_{Nano}-WT GJC in the FN state (a and b) where the hydrophobic residues are located (c). In Cx36_{Nano}-WT GJC in the PLN state, acyl chains are observed in the channel exterior, near the intermolecular interface, and at the cytoplasmic half of TM helices (d-f).

[Reviewer3-Q18]

Lines 402 – 404: How the authors can prove that the equilibrium described is in fact a biologically relevant mechanism and not an artefact of the applied methodology. Independently of the nature of this phenomenon, either artefactual or biologically relevant, the inclusion of lipids in the lumen pore should be a continuous mechanism guided by a thermodynamic equilibrium. Therefore, the finding of intermediate states is consistent with both scenarios.

RESPONSE

We appreciate the reviewer's considerate comments. As the reviewer kindly mentioned in the beginning, the dynamic equilibrium model will certainly reconfigure the biophysical landscape of the GJC world. It is indeed a highly appealing model, but at the same time requires a large body of additional work to prove its physiological importance. However, we have two plausible hypotheses.

1) The dynamic conformational change may be involved in the transport of large molecules. GJCs have been shown to transfer several fluorescent molecules including sulforhodamine B with a minimum diameter of ~ 12 Å. How such a large molecule can go through the channel pore with a diameter of ~ 11 Å is unclear. In our structural analyses, a larger pore with a diameter of >13 Å can be made during the dynamic conformation change. Quite large cellular metabolites including ATP or dinucleotides may more efficiently pass through the channel in the structural equilibrium state.

2) The affinity of NTH for TMD to form the PLN conformation may be optimized to have the structural equilibrium for ease of the gating regulation. The binding of specific signal molecules may completely shift the equilibrium to full PLN or FN for the channel opening or closing. We are studying the effects of cholesterol and known GJ inhibitors and have made quite a bit of progress, and currently searching for the molecules that can completely shift the equilibrium to PLN. These studies would be able to prove the biological relevance.

We have performed extensive protomer-focused 3D classifications for all datasets during revision and have consistently observed the dynamic equilibrium between the FN and PLN states. We found experimental conditions (either detergent or nanodisc) affected the conformational population of NTHs, suggesting that different lipid compositions in different cell types or different membrane domains would shift the equilibrium. These data have been included in the revised manuscript (Supplementary Fig. 10). To clarify whether GJCs are mainly in the structural equilibrium state as shown in our experimental conditions, we will need to directly observe the actual conformation of NTHs in the native cell membrane by using cryo-electron tomography. The sub-tomogram average method will allow us to find the structural equilibrium in the cell membrane, but the low resolution is a bottleneck in this field. In the future, we hope to solve the 3D structure of Cx36 in the cell membrane.

[Reviewer3-Q19]

Lines 409 - 410: The authors should consider further discussion on the role of detergents altering the lipid equilibrium at the micelles. Up to what extent the alteration of the lipid equilibrium may disrupt important biophysical properties of the membrane such as area per lipid, curvature and viscosity?

RESPONSE

In the original manuscript, we described that Cx36 appeared to exist only in the FN state in detergent micelles containing LMNG and CHS, indicating that the PLN-FN equilibrium completely shifted to the FN state. However, a critical comment of the reviewer #1 led us to perform the protomer-focused 3D classification (see Methods for details) for the Cx36^{LMNG-WT} dataset, which led to a different conclusion. Cx36 GJC in detergents, similar as in lipids, showed the dynamic (thermodynamic) equilibrium between the FN and PLN states of NTHs, with $\sim 29\%$ PLN and $\sim 71\%$ FN (Supplementary Fig. 10). We also observed pore obstructing densities of Cx36 GJC in the full FN conformation in detergents in both Cx36^{LMNG-WT} and Cx36^{LMNG-BRIL} datasets and confirmed that the densities have a double layer property

(Supplementary Fig. 2f). Therefore, we concluded that the dynamic conformational equilibrium of NTHs and the pore occlusion in the full FN state commonly occurred in both detergent and lipid environments.

However, since the PLN/FN ratio slightly differs depending on the environmental condition, different biophysical properties such as area per lipid, curvature and viscosity, as the reviewer pointed out, seem to affect the conformational equilibrium. Although we tried to find a possible reason for the change, we could not find a plausible explanation regarding the biophysical properties of lipids and detergents. Instead, we have added the discussion on the Cx36^{LMNG}-BRIL structure because only this structure showed the complete shift of structural equilibrium to the FN state as follows.

Line 449-458:

Although we cannot completely exclude the possibility that the pore-occluding lipids in the FN conformation were artificially introduced during the cryo-EM sample preparation, we consistently observed the conformational equilibrium of NTHs in the structures of Cx36^{LMNG}-WT, Cx36^{Nano}-WT, and Cx36^{Nano}-BRIL GJCs. However, only the FN conformation was observed in Cx36^{LMNG}-BRIL GJC, indicating that the equilibrium completely shifted to FN. When we examined the inner surface of the channel pore of Cx36^{LMNG}-BRIL GJC, strong map densities were observed for the acyl chains of LMNG, but not the head group, and for the sterol ring of CHS, but not the succinyl group. These data suggest that a specific lipid environment with high cholesterol content might induce the FN conformation of the channel, leading to the complete obstruction of the pore.

While the conformational equilibrium of Cx36 GJC was commonly found in the structures of Cx36^{LMNG}-WT, Cx36^{Nano}-WT, and Cx36^{Nano}-BRIL GJCs, only the FN conformation was observed in Cx36^{LMNG}-BRIL GJC, indicating that the equilibrium completely shifted to FN. One possible reason is high concentration of detergents in the Cx36^{LMNG}-BRIL sample. The expression level of Cx36-BRIL in insect cells was much lower than that of Cx36-WT in human cells. Thus, the purified Cx36-BRIL sample was concentrated approximately 100 times from 0.02 mg/mL to 2 mg/mL, while the Cx36-WT sample was concentrated approximately 10 times from 0.15 mg/mL to 1.5 mg/mL. During the concentration of detergent-solubilized proteins using membrane filters, the overall detergent concentration could increase due to partial concentration of protein-free detergent micelles.

Cx36_{LMNG}-WT

Cx36_{LMNG}-BRIL

Cx36_{Nano}-WT

Cx36_{Nano}-BRIL

Supplementary Fig. 10 Protomer-focused 3D classification of Cx36_{LMNG}-WT, Cx36_{LMNG}-BRIL, Cx36_{Nano}-WT and Cx36_{Nano}-BRIL. All protomer classes that do not show clear map density of pore-lining NTH were classified as FN.

[Reviewer3-Q20]

Lines 414 – 415: This is a very interesting observation. The inclusion of cholesterol will certainly change several properties of the membrane. I wonder why the authors didn't included cholesterol in their protocol? Further exploration in this direction could provide insights on the role of cholesterol during connexin-based channel gating and will certainly contribute to the current knowledge.

RESPONSE

Cholesterol is not solubilized very well by detergents, and thus difficult to use in structural and biochemical studies. CHS is widely used to replace cholesterol because 1) it is structurally very similar to cholesterol, 2) more soluble than cholesterol, therefore easier to use in biochemical studies of proteins, and 3) functionally mimics cholesterol quite well.

We have been developing protocols to include cholesterol in lipid-nanodiscs. We are currently studying the effect of the increased cholesterol level on the structural equilibrium. However, this project will take much time and effort, and we feel that it is beyond the scope of this study. We thank the reviewer for the insightful and supportive comments on the importance of further exploration in this direction, and we hope that the reviewer understands that the data was not included in this study.

[Reviewer3-Q21]

Line 419: The authors should include a discussion comparing their results with that of Khan and collaborators (<https://doi.org/10.1016/j.celrep.2020.03.046>) where they found a ball-and-chain mechanism mediated by the NTH for pH-mediated regulation of Cx26 hemichannels. If the ball and chain mechanism will remain valid for GJCs, could it be relevant in dragging lipids into the lumen pore?

RESPONSE

We think that pore occlusion by lipids may not be the only closing model. In the study of Khan and collaborators, Cx26 GJC was reconstituted in amphipol A8-35. Therefore, the role of detergents or lipids is completely excluded. According to the reviewer's suggestion, we have added the following discussion in the revised manuscript.

Line 462-466:

For example, the channel closing by Ca²⁺-loaded calmodulin has been extensively studied, and the calmodulin-cork model has been proposed⁴⁷, although this model needs to be confirmed by structural studies in the future. In addition, direct pore-plugging by NTHs was recently observed in Cx26 GJC at acidic pH⁵. In this study, Cx26 GJC was reconstituted in amphipol A8-35, and thus the role of detergents or lipids was likely excluded.

[Reviewer3-Q22]

Lines 445 – 447: As the authors suggested, in the FN state, the lumen pore becomes obstructed by the presence of lipids. Therefore, it may be the case that Mg²⁺ binding is an intermediate state of the closing process. If this is the case, could the authors find these bindings in the intermediate states of the protomers?

RESPONSE

We thank the reviewer for the valuable suggestion. Unfortunately, the structures of individual protomers cannot be resolved at high resolution due to technical limitations. The reviewer #1

suggested removing the data and discussion of the structure in the presence of Mg^{2+} and we agree with the reviewer #1. We will pursue the study on the effect of Mg^{2+} once we find the condition that completely shifts the conformational equilibrium to PLN. Then, we may be able to prove our hypothesis that Mg^{2+} may block the PLN state of Cx36 by binding to the acidic bands.

[Reviewer3-Q23]

Line 453: consider replacing inhibitory by regulatory

RESPONSE

The whole paragraph on the role of Mg^{2+} has been deleted in the revised manuscript following the suggestion by the reviewer #1.

[Reviewer3-Q24]

Lines 560 – 561: The authors should consider including a brief discussion comparing the position and orientation of the NTHs of both structures. This could be particularly relevant to shed lights on the controversy arising from the rotation of the Cx26 NTH exposing hydrophobic residues to the lumen pore.

RESPONSE

The electron density for NTH was extremely weak in the structure of Cx26 GJC reported in 2009, and thus the NTH structure was likely difficult to be correctly modeled. Since the currently available structures of Cx36, Cx43, Cx46, and Cx50 in the PLN conformation showed similar binding modes of the conserved tryptophan in NTH, we believe that the NTH structure of Cx26 in the PLN conformation was wrong and high-resolution structure of Cx26 in the PLN conformation has never been solved so far. We have added the figures comparing the position and orientation of NTHs of Cx36 and Cx26 (Supplementary Fig. 9g) and the corresponding descriptions in the revised manuscript.

Supplementary Fig. 9g. Structure comparison of Cx36_{Nano}-WT in PLN state and Cx26. Both atomic models of Cx36_{Nano}-WT in PLN state (yellow ribbon) and Cx26 (PDB ID: 2ZW3, blue ribbon) are superimposed (left panel, RMSD of 2.26 Å). Residues 30-33 of

Cx36_{Nano}-WT in PLN state shows π -helix (black ribbon), while α -helix is observed in corresponding residues of Cx26 (residues 28-31, middle and right panel).

Lines 203-210:

... the overall volume of the pore lumen in Cx36 GJC is much smaller than those of Cx43 and Cx46/50 GJCs^{22, 26} (Supplementary Fig. 5a, b). We also compared the NTH structure of Cx36 with that of Cx26 where the conserved tryptophan (Trp3) is exposed to the pore lumen²¹ (Supplementary Fig. 9g). Since the high-resolution structures of Cx36, Cx43, Cx46, and Cx50 commonly showed the binding of the conserved tryptophan to the groove between two adjacent TM1s, Cx26 would have a similar binding mode of Trp3 in its PLN conformation. Because the electron density of Cx26 NTH was extremely weak, the structure of the N-terminus including Trp3 in Cx26 would have been difficult to be correctly modeled.

[Reviewer3-Q25]

Lines 575 – 577: For clarity, the authors should consider removing the electron density map from panel a.

RESPONSE

As the reviewer recommended, we have removed the density map from Fig. 3a (see below).

[Reviewer3-Q26]

Lines 596 – 597: The position of the NTH in the FN state of the channel is not clearly represented. The authors should consider identifying both the position and orientation of the NTH in this state. This is particularly relevant to obtain a better interpretation of this figure compared to figure 5.

RESPONSE

The position of NTH in the FN state was not defined since it is flexible. We assumed that the reviewer probably meant a possible position of NTH. Therefore, we have added dotted circles that indicate possible positions of NTH in the FN state in Fig. 4.

[Reviewer3-Q27]

Line 607: The authors should consider rephrasing this sentence because, in its current form, could be misleading. Despite the lumen pore in the FLN state could be empty of lipids, it is certainly filled with other molecules such as water, ions and gases.

RESPONSE

We replaced the “empty” with “open” as following to avoid any misleading.

Line 1013:

... whereas that of Cx36 in the PLN state is open.

[Reviewer3-Q28]

Line 803: Do the authors refer to heterogenous instead of heterotypic? In the usual language of the field, heterotypic refers to channels composed of protomers belonging to different connexins.

RESPONSE

We have changed “heterotypic” to “hetero-junctional” throughout the manuscript.

[Reviewer3-Q29]

Lines 855 – 857: This is certainly an unorthodox protocol. I wonder why the authors decided not to use a comparative modelling approach such as modeler to generate the coordinates of the CL, using as template the CL region predicted by AlphaFold.

RESPONSE

This comment is related to the point Q4 of the reviewer #1. Please see our response to [reviewer1-Q4]. Briefly, the CL region predicted by Alphafold was flexible, and the conformation was not maintained during the simulation. Thus, we expected that the result would not change much by using a conventional modeling method. We have performed MD simulation for the Cx36 GJC model with complete deletion of CL to see whether flexible CLs affect the ion conductance. We have included new data and interpretation in the revised manuscript (Supplementary Fig. 7 and Lines 314-326)

Reviewers' Comments:

Reviewer #1:

Remarks to the Author:

The manuscript by Lee et al. has been revised and greatly improved by the authors. The newly added data are complementary and compelling. Specifically, the heptameric feature found in the 2D class averages is intriguing, providing a possible and reasonable explanation for lipid migration. This new finding will be an excellent contribution to the field of connexin gap junction research. The revisions and descriptions are fair, careful, and based on an objective point of view. The authors have addressed all my concerns appropriately. I can recommend this manuscript for publication.

Reviewer #2:

Remarks to the Author:

The response is thorough and convincing - the recommendation is to publish.

Reviewer #3:

Remarks to the Author:

Review for manuscript Cryo-EM structures of human Cx36/GJD2 neuronal gap junction channel V2

Rebuttal assessment

The authors of this manuscript have presented a thorough and thoughtful response to the issues raised during the initial revision process. All responses were considered and assessed in detail, as outlined below:

Q1:

1. As suggested during the reviewing process, the authors replaced "GJICH" with "GJC". This change has improved the manuscript by making the terminology usage more appropriate and consistent with current literature.

Q2:

1. Due to space limitations in the abstract, instead of highlighting the unique nature of the NTH-TM1 loop of Cx36, the authors decided to replace this sentence, denoting instead the α -to-n-helix transition of the first transmembrane helix, which weakens the protomer-protomer interaction. Despite this conformational change can be of interest to the structurally aware audience, the current text is not clear about which state (open or obstructed) the conformational change is referring to. Please rephrase it to clarify this point.

Q3:

1. As suggested during the reviewing process, the authors replaced "homology" with "identity". This change has improved the manuscript by making the terminology usage more appropriate and consistent with current literature.

Q4:

1. To highlight the importance behind pharmaceutical approaches aimed to blocking GJCs, the authors included a sentence denoting that in ALS, the secondary neuronal death is extended by neuronal GJCs. Therefore, they suggest that progressive neuronal death can be mitigated by blocking Cx36. Despite sounding, the authors should consider discussing, at least briefly, that Cx36 is widely expressed in the nervous system, including in areas that are not affected by neurodegenerative disorders such as ALS. Therefore, blocking Cx36 GJCs would likely have unwanted side effects in healthy regions of the brain. Moreover, Cx36 channels have been shown to play a role in the regulation of neuronal activity, including synaptic transmission and plasticity. Blocking these channels

could disrupt normal brain function and potentially worsen symptoms of neurodegenerative disorders.
Q5:

1. To state a connection between connexins, innexins and pannexins, the authors included a sentence denoting that "vertebrate GJCs are formed by connexins while invertebrate GJCs consist of innexins with no sequence homology to connexins. Pannexins are innexin homologs in vertebrates and function as hemichannels connecting cytoplasmic and extracellular space." It correctly states that vertebrate GJCs are formed by connexins, and that invertebrate GJCs consist of innexins, which have no sequence homology to connexins. Are the authors referring to sequence identity instead of homology?
2. In consequence with this apparent confusion, the last sentence is not entirely accurate. Pannexins are a family of proteins that are functional homologous to innexins with low sequence identity, therefore, the existence of shared a common ancestor between them is not clearly demonstrated. Hence, they cannot be considered as homolog proteins. Please clarify these concepts.

Q6-Q7:

1. To clarify the nature of the electrostatic binding of the NTH, the authors of this manuscript added the following sentence: "suggesting that the binding of NTHs to TM1 and TM2 of channel pore by the hydrophobic interaction is an essential step for channel opening". Please consider the following correction: "suggesting that the binding of NTHs to TM1 and TM2 of the channel pore through hydrophobic interactions is a crucial step for channel opening"

Q8:

1. As requested, the authors have replaced the use of the term "homology" with "identity." To provide further clarity, it would be beneficial to include the actual identity value in the text.

Q9:

1. Properly solved in Q6.

Q10:

1. As requested, the authors of this manuscript replaced the unorthodox usage of "heterotypic" channel referring to GJC where its constituents hemichannels are in different conformations with "structurally hetero-junctional". This change was applied to the whole manuscript.

Q11:

1. During the previous reviewing process, a contradiction between the author's proposal and the available evidence was identified and pointed out. In lines 251-252 of the original manuscript, the authors stated that "These analyses suggest that the conformations of the two opposing hemichannels are not strongly dependent on each other, but rather individually regulated". Therefore, referring to the "individually regulated hypothesis", I recalled the evidence denoting that maximum conductance arises when the transjunctional voltage difference tends to zero. Therefore, the evidence seems to contradict the aforementioned hypothesis denoting that the regulation of the transjunctional gating depends on a voltage sensing process depending on both opposed hemichannels. This issue was not properly addressed and now the controversial sentence appears in lines 238-239, in the corrected manuscript.

2. On an effort to address the raised issue considering the results of the MD simulation, the authors now included a new paragraph in lines 501-513 in which they propose two alternative hypotheses to explain the obtained conductance differences: 1) The conformationally dynamic state may have the maximum conductance; 2) The full PLN state may be the maximum conductance state and require other factors that are abundant in cells but not included in our experimental system. Despite in the light of the presented evidence, both hypotheses can be sustained, further MD simulations considering the states of the channel should be considered.

Q12:

1. As suggested, the authors were referring to the C-alpha RMSD. Hence, they added "Ca" in front of RMSD.

Q13:

1. As suggested, the authors changed the word "facilitates". Instead, they decided to use "mediates".

Q14:

1. As suggested, the authors included a new reference in lines 427-429.

Q15:

1. As discussed during the reviewing process, the hypothesis of thermodynamic equilibrium

interconnecting different states of the GJC channel, particularly considering the role of lipids as structural mediators of channel blocking, remains as a very appealing yet unsupported proposition. The authors agreed that larger body of evidence is required to fully validate the model, providing a suitable discussion in the rebuttal letter to support their decision to tone down their hypothesis. This discussion should be included in the main text and combined with that of lines 497-500 of the revised manuscript which also addresses the need for further research.

Q16:

1. As discussed during the reviewing process, this is the first time that a GJC structure having lipids blocking the pore lumen has been reported. Therefore, the authors should discuss if the presence of pore-blocking lipids are related to a biologically relevant function, or an artefact of the applied protocol. As mentioned, if these findings are to remain valid, the authors should discuss the role of transjunctional voltage difference regulating the presence of lipids in the channel pore.
2. The extensive and proper response from the authors is very much appreciated.
3. To properly address this issue, the authors now included a sentence in lines 449-450, indicating that they "cannot completely exclude the possibility that the pore-occluding lipids in the FN conformation were artificially introduced during the cryo-EM sample preparation". The authors should consider including an improved version of the Author's response figure 4 in the supplementary material with a proper legend.
4. The authors have included a new paragraph between lines 489-500, in which they discuss the role of transjunctional voltage difference, an issue that was also raised by Reviewer 1 in Q27. Of note, as the authors mention in this paragraph "Because two hemichannel regions in a GJC face each other, their responses to the applied transjunctional voltage would be opposite to each other". This is an argument defying the controversial sentence appearing in lines 238-239 in the revised manuscript (see Q11)

Q17:

1. As raised during the reviewing process and previously mentioned Q15, a larger body of evidence is required to establish an equilibrium between the open and closed state of the channel. This is particularly relevant when the available evidence suggests that the lumen pore is highly hydrophilic. In such case, a strong free energy gradient should be compensated so to be able to change the conformation of the NTH to allow the insertion of membrane lipids. According to the response to Q15, the authors agree that a larger body of evidence is needed, denoting that this issue is related to Reviewer 1, Q5. Moreover, the authors should discuss the accessing pathway that membrane lipids should follow to block the lumen pore.
2. The authors included a new panel in supplementary figure 1f and a new paragraph between lines 479-484 denoting that "Interestingly, we have observed top-views of either tetradecameric GJCs or heptameric hemichannels in all the datasets collected in this study. This suggests that Cx36 GJC or hemichannel might be interchangeable between two different assemblies with six- and seven-fold symmetries (Supplementary Fig. 1f)."
3. If the existence of an actual seven-fold symmetry channel is to remain valid, a larger body of evidence is needed. Particularly, in the light of the role of the transition between six-fold and seven-fold symmetries of the GJC during the inclusion of lipids in the channel lumen, as mentioned by the authors "The interchange would require a temporary disruption of the channel assemblies, which may allow the diffusion of lipids into and out of channel pores."
4. A proper discussion is needed to better support the existence of the equilibrium between the seven-fold and six-fold symmetries, particularly from the biological point of view as this transition may be the result of an artifact produced by the nano-disc experimental protocol. Consider the suitability for this discussion of an improved version of the Author's response figure 4 in the supplementary material together with a proper legend.

Q18:

1. During the reviewing process, an issue was raised related to the inclusion of lipids in the lumen pore as evidence supporting the existence of a thermodynamic equilibrium. This issue is particularly relevant considering that the finding of intermediate states is consistent with both an artifact of the experimental protocol and a biologically relevant process. It is also consistent with the analysis to the authors response denoted in Q17.4, above.

2. Despite the hypotheses mentioned by the authors in their response should be carefully evaluated, without further evidence it should be considered as part of the discussion arising from their results. Please consider incorporating this interesting discussion to that of lines 479-484, as denoted in Q17.2, above.

Q19:

1. Another issue was raised on the role of detergents altering the lipid equilibrium at the micelles used during the experimental protocol related to determine up to what extent the alteration of the lipid equilibrium may disrupt important biophysical properties of the membrane such as area per lipid, curvature and viscosity?

2. A critical issue raised by reviewer 1 led the authors to perform the protomer-focused 3D classification arriving to a different conclusion. The authors now concluded that "the dynamic conformational equilibrium of NTHs and the pore occlusion in the full FN state commonly occurred in both detergent and lipid environments".

3. The authors mention in their rebuttal letter that "However, since the PLN/FN ratio slightly differs depending on the environmental condition, different biophysical properties such as area per lipid, curvature and viscosity, as the reviewer pointed out, seem to affect the conformational equilibrium. Although we tried to find a possible reason for the change, we could not find a plausible explanation regarding the biophysical properties of lipids and detergents. Instead, we have added the discussion on the Cx36LMNG-BRIL structure because only this structure showed the complete shift of structural equilibrium to the FN state..."

4. In consequence, the authors included a new paragraph between lines 449-458, denoting that "Although we cannot completely exclude the possibility that the pore-occluding lipids in the FN conformation were artificially introduced during the cryo-EM sample preparation, we consistently observed the conformational equilibrium of NTHs in the structures of Cx36LMNG-WT, Cx36Nano-WT, and Cx36Nano-BRIL GJCs." Moreover, they also mention that "These data suggest that a specific lipid environment with high cholesterol content might induce the FN conformation of the channel, leading to the complete obstruction of the pore. "

5. Despite the importance of other biophysical membrane properties was not thoroughly addressed, the general issue is now discussed.

Q20:

1. In this question, the role of cholesterol during connexin-based channel gating is highlighted.

2. In their rebuttal letter, the authors mention that they "have been developing protocols to include cholesterol in lipid-nanodiscs. We are currently studying the effect of the increased cholesterol level on the structural equilibrium."

3. The authors also "thank the reviewer for the insightful and supportive comments on the importance of further exploration in this direction, and we hope that the reviewer understands that the data was not included in this study."

4. Considering the complexity behind the development of this protocol and the authors awareness on the role of cholesterol, the issue is solved.

Q21:

1. This question raised the need for the comparison of the authors' result with that of Khan and collaborators (<https://doi.org/10.1016/j.celrep.2020.03.046>) where they found a ball-and-chain mechanism mediated by the NTH for pH-mediated regulation of Cx26 hemichannels.

2. A brief but proper discussion is now included between lines 462-466, including this reference.

Q22:

1. This question is related to the role of Mg²⁺ binding as an intermediate state of the closing process. If this is the case, could the authors find these bindings in the intermediate states of the protomers?

2. Following a suggestion from Reviewer 1, the data and discussion of the structure in the presence of Mg²⁺ was removed.

3. The authors mention that they "will pursue the study on the effect of Mg²⁺ once we find the condition that completely shifts the conformational equilibrium to PLN. Then, we may be able to prove our hypothesis that Mg²⁺ may block the PLN state of Cx36 by binding to the acidic bands. "

4. The issue is solved.

Q23:

1. Solved according to Q22.

Q24:

1. During the reviewing process, an issue was raised according to need for a brief discussion comparing the position and orientation of the NTHs.
2. As suggested, the authors included a new panel (g) in supplementary figure 9, and a proper discussion between lines 203-210.

Q25:

1. As suggested, the authors removed the density map from Fig. 3a

Q26:

1. As suggested, the authors included dotted circles indicating possible positions of NTH in the FN state in Fig. 4

Q27:

1. As suggested, the authors replaced the "empty" with "open" to avoid any misleading.

Q28:

1. During the reviewing process, an issue was raised regarding the use of modelling protocol for the CL. A similar issue was raised by Q4 of the reviewer #1.
2. The authors decided to perform MD simulation for the Cx36 GJC model with complete deletion of CL to see whether flexible CLs affect the ion conductance.
3. They now included new data and interpretation in the revised manuscript (Supplementary Fig. 7 and Lines 314-326)
4. The issue is solved.

Authors' Response to Reviewers' Comments

Reviewer #1 (Remarks to the Author):

The manuscript by Lee et al. has been revised and greatly improved by the authors. The newly added data are complementary and compelling. Specifically, the heptameric feature found in the 2D class averages is intriguing, providing a possible and reasonable explanation for lipid migration. This new finding will be an excellent contribution to the field of connexin gap junction research. The revisions and descriptions are fair, careful, and based on an objective point of view. The authors have addressed all my concerns appropriately. I can recommend this manuscript for publication.

RESPONSE

We appreciate the reviewer's comments.

Reviewer #2 (Remarks to the Author):

The response is thorough and convincing - the recommendation is to publish.

RESPONSE

We appreciate the reviewer's comments.

Reviewer #3 (Remarks to the Author):

Review for manuscript Cryo-EM structures of human Cx36/GJD2 neuronal gap junction channel V2

Rebuttal assessment

The authors of this manuscript have presented a thorough and thoughtful response to the issues raised during the initial revision process. All responses were considered and assessed in detail, as outlined below:

RESPONSE

We appreciate the reviewer's constructive comments and provide point-by-point responses.

[Reviewer3-Q1]

1. As suggested during the reviewing process, the authors replaced “GJCh” with “GJC”. This change has improved the manuscript by making the terminology usage more appropriate and consistent with current literature.

RESPONSE

We appreciate the reviewer’s comments.

[Reviewer3-Q2]

1. Due to space limitations in the abstract, instead of highlighting the unique nature of the NTH-TM1 loop of Cx36, the authors decided to replace this sentence, denoting instead the α -to- π -helix transition of the first transmembrane helix, which weakens the protomer-protomer interaction. Despite this conformational change can be of interest to the structurally aware audience, the current text is not clear about which state (open or obstructed) the conformational change is referring to. Please rephrase it to clarify this point.

RESPONSE

The alpha-to-phi conformational change in TM2 is related to the channel opening. As suggested by the reviewer, we have revised the sentence as following.

Lines 26-29:

... In the open state with pore-lining NTHs, the pore is more acidic than those in Cx26 and Cx46/50 GJCs, explaining its strong cation selectivity. **The conformational change during channel opening also includes** the α -to- π -helix transition of the first transmembrane helix, which weakens the protomer-protomer interaction.

[Reviewer3-Q3]

1.As suggested during the reviewing process, the authors replaced “homology” with “identity”. This change has improved the manuscript by making the terminology usage more appropriate and consistent with current literature.

RESPONSE

We appreciate the reviewer’s comments.

[Reviewer3-Q4]

1.To highlight the importance behind pharmaceutical approaches aimed to blocking GJCs, the authors included a sentence denoting that in ALS, the secondary neuronal death is extended by neuronal GJCs. Therefore, they suggest that progressive neuronal death can be mitigated by blocking Cx36. Despite sounding, the authors should consider discussing, at least briefly, that Cx36 is widely expressed in the nervous system, including in areas that are not affected by neurodegenerative disorders such as ALS. Therefore, blocking Cx36 GJCs would likely have unwanted side effects in healthy regions of the brain. Moreover, Cx36 channels have been shown to play a role in the regulation of neuronal activity, including synaptic transmission and

plasticity. Blocking these channels could disrupt normal brain function and potentially worsen symptoms of neurodegenerative disorders.

RESPONSE

As the reviewer suggested, we have added the following sentence.

Lines 67-69:

However, since Cx36 is widely expressed in the nervous system and plays a role in the regulation of neuronal activity, inhibiting Cx36 GJCs could disrupt normal brain function and potentially worsen the symptoms of neurodegenerative disorders.

[Reviewer3-Q5]

1.To state a connection between connexins, innexins and pannexins, the authors included a sentence denoting that “vertebrate GJCs are formed by connexins while invertebrate GJCs consist of innexins with no sequence homology to connexins. Pannexins are innexin homologs in vertebrates and function as hemichannels connecting cytoplasmic and extracellular space.” It correctly states that vertebrate GJCs are formed by connexins, and that invertebrate GJCs consist of innexins, which have no sequence homology to connexins. Are the authors referring to sequence identity instead of homology?

2.In consequence with this apparent confusion, the last sentence is not entirely accurate. Pannexins are a family of proteins that are functional homologous to innexins with low sequence identity, therefore, the existence of shared a common ancestor between them is not clearly demonstrated. Hence, they cannot be considered as homolog proteins. Please clarify these concepts.

RESPONSE

1. According to the reviewer's comment, we have replaced "homology" with "identity".

Line 42:

GJCs consist of innexins with no sequence **identity** to connexins.

2. We agree and have modified the sentence as following.

Line 42-44:

... Vertebrate Pannexins have **detectable sequence identity and structural similarity with innexins, but** function as hemichannels connecting cytoplasmic and extracellular space³.

[Reviewer3-Q6-7]

1.To clarify the nature of the electrostatic binding of the NTH, the authors of this manuscript added the following sentence: “suggesting that the binding of NTHs to TM1 and TM2 of channel pore by the hydrophobic interaction is an essential step for channel opening”. Please consider the following correction: “suggesting that the binding of NTHs to TM1 and TM2 of the channel pore through hydrophobic interactions is a crucial step for channel opening”

RESPONSE

As the reviewer suggested, we have replaced "by" with "through".

Lines 82-83:

... binding of NTHs to TM1 and TM2 of the channel pore **through** the hydrophobic interaction is an essential step ...

[Reviewer3-Q8]

1.As requested, the authors have replaced the use of the term "homology" with "identity." To provide further clarity, it would be beneficial to include the actual identity value in the text.

RESPONSE

We have added the sequence identity value of 52-54% in the revised manuscript.

Lines 113-114:

... as expected from the high sequence identity (**52-54%**) between these connexins^{21,22}.

[Reviewer3-Q9]

1.Properly solved in Q6.

RESPONSE

We appreciate the reviewer's comments.

[Reviewer3-Q10]

1.As requested, the authors of this manuscript replaced the unorthodox usage of "heterotypic" channel referring to GJC where its constituents hemichannels are in different conformations with "structurally hetero-junctional". This change was applied to the whole manuscript.

RESPONSE

We appreciate the reviewer's comments.

[Reviewer3-Q11]

1.During the previous reviewing process, a contradiction between the author's proposal and the available evidence was identified and pointed out. In lines 251-252 of the original manuscript, the authors stated that "These analyses suggest that the conformations of the two opposing hemichannels are not strongly dependent on each other, but rather individually regulated". Therefore, referring to the "individually regulated hypothesis", I recalled the evidence denoting that maximum conductance arises when the transjunctional voltage difference tends to zero. Therefore, the evidence seems to contradict the aforementioned hypothesis denoting that the regulation of the transjunctional gating depends on a voltage sensing process depending on both

opposed hemichannels. This issue was not properly addressed and now the controversial sentence appears in lines 238-239, in the corrected manuscript.

2. On an effort to address the raised issue considering the results of the MD simulation, the authors now included a new paragraph in lines 501-513 in which they propose two alternative hypotheses to explain the obtained conductance differences: 1) The conformationally dynamic state may have the maximum conductance; 2) The full PLN state may be the maximum conductance state and require other factors that are abundant in cells but not included in our experimental system. Despite in the light of the presented evidence, both hypotheses can be sustained, further MD simulations considering the states of the channel should be considered.

RESPONSE

1. It seems that we did not completely understand the reviewer's comment in the previous revision. We now realized that the phrase "individually regulated" could mislead the reader, and thus have removed in the revised manuscript. As the reviewer pointed out, the transjunctional voltage apply on both hemichannels, they cannot be regulated separately. We have modified the sentence not to explain the gating regulation but to focus on the structural interdependency between two hemichannel regions as following.

Lines 242-243:

These analyses suggest that two opposing hemichannel regions are not structurally interdependent and may have different conformations.

2. Cx36 GJC with a mixture of PLN and FN (4:2 to 0:6 ratios) shows a pore with a solvent accessible diameter of ~12 angstrom (Supplementary Fig. 12), which is much larger than that of the full PLN GJC (8.5 angstrom). Therefore, it is reasonable to think that the mixed conformation would show a higher current than the full PLN conformation. However, the ionic transfer would be largely affected by lipids inside the pore, and we do not know how many and what lipid molecules are in the pore of each GJC conformation in cell membrane. Because of the limited information on lipids in the pore, MD simulation of ionic current through each GJC conformation would not be correctly calculated. We will pursue the structural study on the mixed conformations to obtain the information on lipids in the pore. And then, we will begin further MD simulation based on the structural information.

[Reviewer3-Q12]

1. As suggested, the authors were referring to the C-alpha RMSD. Hence, they added "C α " in front of RMSD.

RESPONSE

We appreciate the reviewer's comments.

[Reviewer3-Q13]

1.As suggested, the authors changed the word “facilitates”. Instead, they decided to use “mediates”.

RESPONSE

We appreciate the reviewer’s comments.

[Reviewer3-Q14]

1.As suggested, the authors included a new reference in lines 427-429.

RESPONSE

We appreciate the reviewer’s comments.

[Reviewer3Q15]

1. As discussed during the reviewing process, the hypothesis of thermodynamic equilibrium interconnecting different states of the GJC channel, particularly considering the role of lipids as structural mediators of channel blocking, remains as a very appealing yet unsupported proposition. The authors agreed that larger body of evidence is required to fully validate the model, providing a suitable discussion in the rebuttal letter to support their decision to tone down their hypothesis. This discussion should be included in the main text and combined with that of lines 497-500 of the revised manuscript which also addresses the need for further research.

RESPONSE

We appreciate the reviewer's comment. We have added sentences describing need for definition of conformational states of NTHs in electrophysiological experiments.

Lines 527-537:

However, the hypothesis of thermodynamic equilibrium interconnecting different states of Cx36 GJC, particularly considering the role of lipids as structural mediators of channel blocking, remains to be validated by functional experiments. Unfortunately, there is no method to measure the conformational state of GJCs during a patch clamp recording. In addition, it is difficult to design mutational studies for the measurement of ionic current in a specific channel conformation. Any mutation to fix NTH or TM1 in a specific conformation would inhibit the conformational dynamics of the channel, which may be crucial for lipid exclusion at its initial assembly. To prove this hypothesis, further structural and electrophysiological studies are needed to find molecules or conditions that greatly shift the conformational equilibrium of the channel to the FN and PLN states, respectively, and investigate their effects on the ionic current through the channel.

[Reviewer3-Q16]

1.As discussed during the reviewing process, this is the first time that a GJC structure having lipids blocking the pore lumen has been reported. Therefore, the authors should discuss if the presence of pore-blocking lipids are related to a biologically relevant function, or an artefact of the applied protocol. As mentioned, if these finding are to remain valid, the authors should

discuss the role of transjunctional voltage difference regulating the presence of lipids in the channel pore.

2. The extensive and proper response from the authors is very much appreciated.

3. To properly address this issue, the authors now included a sentence in lines 449-450, indicating that the “cannot completely exclude the possibility that the pore-occluding lipids in the FN conformation were artificially introduced during the cryo-EM sample preparation”. The authors should consider including an improved version of the Author’s response figure 4 in the supplementary material with a proper legend.

4. The authors have included a new paragraph between lines 489-500, in which they discuss the role of transjunctional voltage difference, an issue that was also raised by Reviewer 1 in Q27. Of note, as the authors mention in this paragraph “Because two hemichannel regions in a GJC face each other, their responses to the applied transjunctional voltage would be opposite to each other”. This is an argument defying the controversial sentence appearing in lines 238-239 in the revised manuscript (see Q11)

RESPONSE

3. As the reviewer suggested, we added the Author’s response figure 4 to Supplementary Fig. 11 with figure legend.

4. As we answered in Q11.1, we have modified the sentence (lines 241-242 in this revised version) to focus on the structural interdependency between two hemichannels rather than explaining the gating regulation.

Supplementary Fig. 11: Possible scenarios of lipid exclusion during the channel assembly process. A hypothetical model of stepwise lipid exclusion after the completion of channel assembly is compared with an unlikely model of immediate lipid exclusion before the completion

of channel assembly. The α -helices and β -sheets are represented as cylinders and arrows, respectively. The amphipathic NTH is represented as a magenta cylinder. In the boxes that display the snapshots of the channel assembly process viewed from the top, FN protomers, PLN protomers, and membrane lipids are represented as green, yellow, and black circles. Red arrows indicate movement of lipids.

[Reviewer3-Q17]

1.As raised during the reviewing process and previously mentioned Q15, a larger body of evidence is required to establish an equilibrium between the open and closed state of the channel. This is particularly relevant when the available evidence suggest that the lumen pore is highly hydrophilic. In such case, a strong free energy gradient should be compensated so to be able to change the conformation of the NTH to allow the insertion of membrane lipids. According to the response to Q15, the authors agree that a larger body of evidence is needed, denoting that this issue is related to Reviewer 1, Q5. Moreover, the authors should discuss the accessing pathway that membrane lipids should follow to block the lumen pore.

2.The authors included a new panel in supplementary figure 1f and a new paragraph between lines 479-484 denoting that “Interestingly, we have observed top-views of either tetradecameric GJCs or heptameric hemichannels in all the datasets collected in this study. This suggests that Cx36 GJC or hemichannel might be interchangeable between two different assemblies with six- and seven-fold symmetries (Supplementary Fig. 1f).”

3.If the existence of an actual seven-fold symmetry channel is to remain valid, a larger body of evidence is needed. Particularly, in the light of the role of the transition between six-fold and seven-fold symmetries of the GJC during the inclusion of lipids in the channel lumen, as mentioned by the authors “The interchange would require a temporary disruption of the channel assemblies, which may allow the diffusion of lipids into and out of channel pores.”

4.A proper discussion is needed to better support the existence of the equilibrium between the seven-fold and six-fold symmetries, particularly from the biological point of view as this transition may be the result of an artifact produced by the nano-disc experimental protocol. Consider the suitability for this discussion of an improved version of the Author’s response figure 4 in the supplementary material together with a proper legend.

RESPONSE

Despite of high topological similarity of GJCs including connexins, pannexins, innexins, their structures show various oligomeric states such as hexameric-, heptameric- and octameric state, respectively. Although it is not yet confirmed in Cxs, the symmetry disruption was already observed in pannexin-1. It was hexameric in a negative staining EM experiment (Chiu et al., 2017), but heptameric assembly in several cryo-EM structures (Qu et al., 2020). As the reviewer suggested in Q17.4, we have added following figures as Supplementary Fig. 11b-c and sentences to lines 481-486, to mention the possibility of an artifact and explain why we believe the possibility is not high.

We did not include the discussion on the potential functional role of the heptameric state in cells because we feel it is too much speculation without evidence. Based on the predicted structural model of the 14-meric GJC, the channel may form a bigger pore in its open state compared with

the 12-meric channel, but it may be more difficult to open since more lipids should be inside the pore in its FN state. In contrast, the 14-meric GJC might not exist at all (we don't know yet whether the top view of the channel particle with 7-fold symmetry is from hemichannel or GJC). The heptameric hemichannel might be also formed only shortly in the ER membrane, and thus it may exist in cells but have no functional role as a channel. Nevertheless, this does not exclude a possible role of the heptameric state in channel assembly or lipid exclusion.

Lines 485-493:

... hemichannels in all the datasets collected in this study (Supplementary Fig. 1f). Although we cannot exclude the possibility that these channels are artificially created during protein purification or nanodisc reconstitution, the fact that they were commonly observed in both detergent and lipid environments suggests that they might be formed in various detergent/lipid environments including cell membranes. This observation also suggests that Cx36 GJC or hemichannel might be interchangeable between two different assemblies with six- and seven-fold symmetries. The interchange would require a temporary disruption of the channel assemblies, which may allow the diffusion of lipids into and out of channel pores (Supplementary Fig. 11). This idea needs to be validated by cryo-ET or single molecule experiments.

References

1. Chiu, Yu-Hsin, *et al.* A quantized mechanism for activation of pannexin channels. *Nat. Commun.* **8.1**, 14324 (2017).
2. Qu, Ronggui, *et al.* Cryo-EM structure of human heptameric Pannexin 1 channel. *Cell Res.* **30.5**, 446-448 (2020).

[Reviewer3-Q18]

1. During the reviewing process, an issue was raised related to the inclusion of lipids in the lumen pore as evidence supporting the existence of a thermodynamic equilibrium. This issue is particularly relevant considering that the finding of intermediate states is consistent with both an artifact of the experimental protocol and a biological relevant process. It is also consistent with the analysis to the authors response denoted in Q17.4, above.

2. Despite the hypotheses mentioned by the authors in their response should be carefully evaluated, without further evidence it should be considered as part of the discussion arising from their results. Please consider incorporating this interesting discussion to that of lines 479-484, as denoted in Q17.2, above.

RESPONSE

We have added following discussion in the revised manuscript.

Lines 520-527:

The dynamic conformational change observed in Cx36 GJC, as previously shown in Cx43 GJC, may help large molecules pass through the channels. Since a pore with a diameter of ~12 Å can be formed during the dynamic conformation change (Supplementary Fig. 12), large cellular

metabolites such as ATP or dinucleotides might pass through the channel more efficiently in the structural equilibrium state than in the full PLN state. The binding affinity of NTH for TMD to form the PLN conformation may have been optimized to maintain the structural equilibrium in the cell membrane for ease of the gating regulation. Therefore, the binding of specific signal molecules could shift the equilibrium to the full PLN or FN state for the channel opening or closing.

Supplementary Fig. 12: Pore sizes of the Cx36 hemichannel region in different compositions of PLN and FN. Each hemichannel region with a mixture of PLN and FN (4:2 to 0:6 ratios) shows a pore with a solvent accessible diameter of ~12 Å, which is much larger than that of the full PLN GJC (8.5 Å). The solvent-accessible pore diameters were calculated using HOLE program¹. Yellow and green circles indicate PLN and FN protomers, respectively. Source data for solvent-accessible pore diameters are provided as a Source Data file.

[Reviewer3-Q19]

1. Another issue was raised on the role of detergents altering the lipid equilibrium at the micelles used during the experimental protocol related to determine up to what extent the alteration of the lipid equilibrium may disrupt important biophysical properties of the membrane such as area per lipid, curvature and viscosity?

2. A critical issue raised by reviewer 1 led the authors to perform the protomer-focused 3D classification arriving to a different conclusion. The authors now concluded that “the dynamic conformational equilibrium of NTHs and the pore occlusion in the full FN state commonly occurred in both detergent and lipid environments”.

3.The authors mention in their rebuttal letter that “However, since the PLN/FN ratio slightly differs depending on the environmental condition, different biophysical properties such as area per lipid, curvature and viscosity, as the reviewer pointed out, seem to affect the conformational equilibrium. Although we tried to find a possible reason for the change, we could not find a plausible explanation regarding the biophysical properties of lipids and detergents. Instead, we have added the discussion on the Cx36LMNG-BRIL structure because only this structure showed the complete shift of structural equilibrium to the FN state...”

4.In consequence, the authors included a new paragraph between lines 449-458, denoting that “Although we cannot completely exclude the possibility that the pore-occluding lipids in the FN conformation were artificially introduced during the cryo-EM sample preparation, we consistently observed the conformational equilibrium of NTHs in the structures of Cx36LMNG-WT, Cx36Nano-WT, and Cx36Nano-BRIL GJCs.” Moreover, they also mention that “These data suggest that a specific lipid environment with high cholesterol content might induce the FN conformation of the channel, leading to the complete obstruction of the pore. “

5.Despite the importance of other biophysical membrane properties was not thoroughly addressed, the general issue is now discussed.

RESPONSE

We appreciate the reviewer’s comments.

[Reviewer3-Q20]

1.In this question, the role of cholesterol during connexin-based channel gating is highlighted.

2.In their rebuttal letter, the authors mention that they “have been developing protocols to include cholesterol in lipid-nanodiscs. We are currently studying the effect of the increased cholesterol level on the structural equilibrium.”

3.The authors also “thank the reviewer for the insightful and supportive comments on the importance of further exploration in this direction, and we hope that the reviewer understands that the data was not included in this study.”

4.Considering the complexity behind the development of this protocol and the authors awareness on the role of cholesterol, the issue is solved.

RESPONSE

We appreciate the reviewer’s comments.

[Reviewer3-Q21]

1.This question raised the need for the comparison of the authors’ result with that of Khan and collaborators (<https://doi.org/10.1016/j.celrep.2020.03.046>) where they found a ball-and-chain mechanism mediated by the NTH for pH-mediated regulation of Cx26 hemichannels.

2.A brief but proper discussion is now included between lines 462-466, including this reference.

RESPONSE

We appreciate the reviewer's comments.

[Reviewer3-Q22]

1.This question is related to the role of Mg²⁺ binding as an intermediate state of the closing process. If this is the case, could the authors find these bindings in the intermediate states of the protomers?

2.Following a suggestion from Reviewer 1, the data and discussion of the structure in the presence of Mg²⁺ was removed.

3.The authors mention that they “will pursue the study on the effect of Mg²⁺ once we find the condition that completely shifts the conformational equilibrium to PLN. Then, we may be able to prove our hypothesis that Mg²⁺ may block the PLN state of Cx36 by binding to the acidic bands. “

4.The issue is solved.

RESPONSE

We appreciate the reviewer's comments.

[Reviewer3-Q23]

1.Solved according to Q22.

Q24:

1.During the reviewing process, an issue was raised according to need for a brief discussion comparing the position and orientation of the NTHs.

2.As suggested, the authors included a new panel (g) in supplementary figure 9, and a proper discussion between lines 203-210.

RESPONSE

We appreciate the reviewer's comments.

[Reviewer3-Q25]

1.As suggested, the authors removed the density map from Fig. 3a

RESPONSE

We appreciate the reviewer's comments.

[Reviewer3-Q26]

1.As suggested, the authors included dotted circles indicating possible positions of NTH in the FN state in Fig. 4

RESPONSE

We appreciate the reviewer's comments.

[Reviewer3-Q27]

1.As suggested, the authors replaced the “empty” with “open” to avoid any misleading.

RESPONSE

We appreciate the reviewer's comments.

[Reviewer3-Q28]

1.During the reviewing process, an issue was raised regarding the use of modelling protocol for the CL. A similar issue was raised by Q4 of the reviewer #1.

2.The authors decided to perform MD simulation for the Cx36 GJC model with complete deletion of CL to see whether flexible CLs affect the ion conductance.

3.They now included new data and interpretation in the revised manuscript (Supplementary Fig. 7 and Lines 314-326)

4.The issue is solved.

RESPONSE

We appreciate the reviewer's comments.